# SOFT INJECTION OF TASK EMBEDDINGS OUTPERFORMS PROMPT-BASED IN-CONTEXT LEARNING

## ABSTRACT

In-Context Learning (ICL) enables Large Language Models (LLMs) to perform tasks by conditioning on input-output examples in the prompt, without requiring any update in model parameters. While widely adopted, it remains unclear whether prompting with multiple examples is the most effective and efficient way to convey task information. In this work, we propose **Soft Injection** of task embeddings at attention heads. The task embeddings are constructed only once using few-shot ICL prompts and repeatedly used during inference. Soft injection is performed by *softly mixing* pre-computed task embeddings with attention head activations using pre-optimized mixing parameters, referred to as *soft head-selection parameters*. This method not only allows a desired task to be performed without in-prompt demonstrations but also significantly outperforms few-shot ICL while reducing memory usage and compute cost at inference time. An extensive evaluation is performed across 57 tasks and 12 LLMs, spanning four model families of sizes from 4B to 70B. Averaged across 57 tasks, our method outperforms 10-shot ICL by 10.2%–14.3% across 12 LLMs. A series of analyses show that our method also serves as an insightful tool for analyzing task-relevant roles of attention heads, revealing that task-relevant head positions identified by our method transfer across similar tasks but not across dissimilar ones–uncovering the task-specific nature of head functionality. *Our soft injection method significantly improves task performance and reveals task-specific attention heads, deepening the mechanistic understanding of the roles of attention heads in LLMs.*

## 1 INTRODUCTION

In-context learning (ICL) (Brown et al., 2020) has emerged as a key mechanism for enabling general-purpose use of LLMs, allowing models to perform tasks using only a few input-output examples in the prompt, without model finetuning. A substantial body of research has focused on explaining ICL's effectiveness or developing prompt engineering to improve its performance (Xie et al., 2021; Olsson et al., 2022; Min et al., 2022; Dong et al., 2022; Ye et al., 2022; Wies et al., 2023; Agarwal et al., 2024; Zhang et al., 2025). However, it remains unclear whether prompting with multiple demonstrations is truly the most effective and efficient way to convey task information. In this work, we challenge this assumption by proposing an alternative approach that achieves substantially higher performance while reducing inference-time memory usage and compute cost.

Our approach draws inspiration from two prior works: Function Vector (FV) (Todd et al., 2023) and DARTS (Liu et al., 2018). Although FV has limited performance, it demonstrates that task information can be directly injected into model activations to guide behavior without input-output demonstrations in the prompt. DARTS, a well-known method for neural architecture search (Zoph & Le, 2016), exemplifies how a continuous relaxation of a discrete search space enables efficient gradient-based optimization to solve challenging deep learning problems. Building on these insights, we propose a task embedding injection method that reformulates head selection as a continuous optimization problem, enabling gradient descent to learn task-specific soft injection strategies.

Our method, called SITE (Soft Injection of Task Embeddings), constructs task embeddings and optimizes soft head-selection parameters only once per task. During inference with zero-shot prompts, the task embeddings–which encode task-relevant information–are mixed with the original attention head activations, using the soft head-selection parameters as interpolation weights. This allows

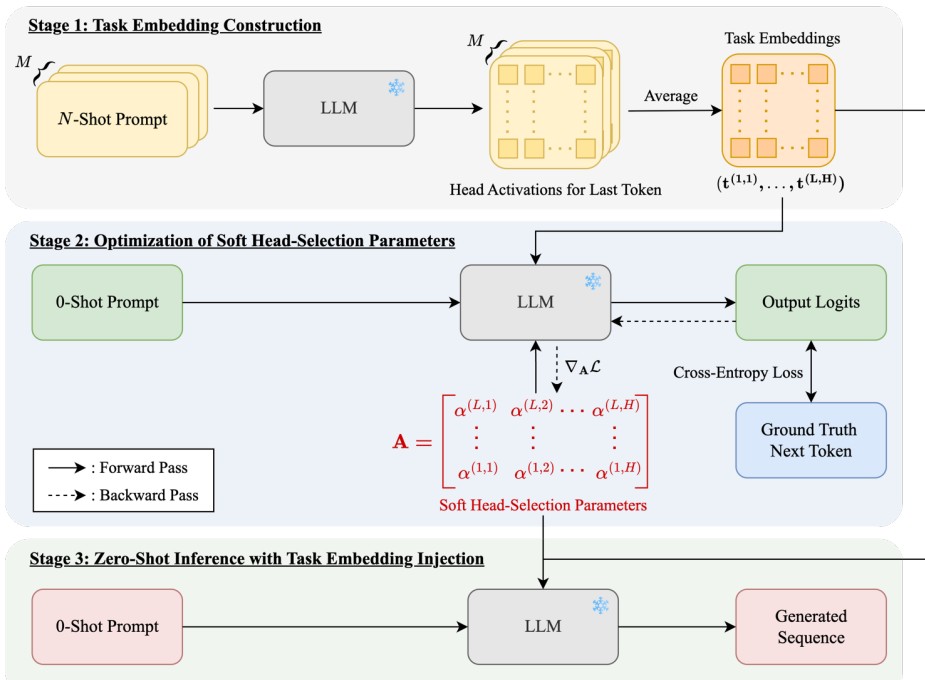

Figure 1: **Method Overview.** Our method consists of three stages. (1) A set of task embeddings is constructed by averaging attention head activations for the last token across few-shot ICL prompts, using $M{=}50$ prompts each with $N{=}10$ input–output pairs. (2) Soft head-selection parameters are optimized to determine how the task embeddings should be injected into the model during zero-shot inference. (3) At inference time, the set of task embeddings and the learned selection parameters are used to guide the model in performing tasks without any in-prompt examples. $L$ and $H$ denote the number of attention layers and the number of attention heads per layer, respectively, in the LLM.

LLMs to perform tasks without input-output examples in the prompt, while achieving significant performance gains over few-shot ICL baselines. We evaluate SITE on 57 ICL tasks (29 abstractive and 28 extractive) across 12 LLMs spanning 4 model families, 3 model variants, and model sizes ranging from 4B to 70B parameters. SITE achieves average performance gains of 10.2%–14.3% over 10-shot ICL, demonstrating strong task performance across diverse LLMs.

Since SITE reformulates head selection as a continuous optimization problem, it also serves as an insightful tool for attributional analysis of task-relevant attention heads. A series of analyses show that SITE not only identifies task-relevant heads, but also reveals that the selected head positions transfer across similar tasks but not across dissimilar ones–uncovering the task-specific nature of head functionality. Additionally, we present four empirical findings: (1) validation loss drops significantly during training, indicating effective optimization of head-selection parameters (i.e., injection locations), (2) increasing the number of examples (or shots) in prompt-based ICL from a few to many does not always improve performance–some tasks remain challenging even with many-shot prompting, while SITE addresses them effectively, (3) a single 10-shot prompt is sufficient to construct task embeddings that significantly outperform 10-shot ICL, highlighting the importance of precise injection over the number of demonstrations, and (4) SITE scales efficiently, matching the runtime and memory cost of zero-shot inference while requiring only minimal computation for the one-time construction of task embeddings and optimization of soft head-selection parameters.

## 2 METHOD

Our method consists of three main stages. (1) First, we construct a set of task embeddings using few-shot ICL prompts. (2) Next, we optimize soft head-selection parameters via gradient descent. (3) Finally, we apply the pre-computed task embeddings and the learned head-selection parameters

to the model, enabling it to perform tasks without any in-prompt examples. Figure 1 provides an overview of our method.

**Task embedding construction.** FV (Todd et al., 2023) showed that averaging the activations of individual attention heads at the last token across few-shot ICL prompts can encode task information. MTV (Huang et al., 2024) adopted these averaged activations as *task embeddings*, which we also use in our method[1]. Let $P_1, P_2, \ldots, P_M$ denote the $M$ few-shot ICL prompts, each containing $N$ input-output examples sampled from the training set. For each prompt $P_m$, we pass it through the model and extract the output of every attention head at every layer. Specifically, for head $h \in \{1, 2, \ldots, H\}$ in layer $l \in \{1, 2, \ldots, L\}$, the output of head $h$ at layer $l$ for prompt $P_m$ is given by:

$$\mathbf{t}_m^{(l,h)} = \text{softmax}\left(\frac{\mathbf{Q}_m^{(l,h)}(\mathbf{K}_m^{(l,h)})^T}{\sqrt{d_k}}\right)\mathbf{V}_m^{(l,h)} \in \mathbb{R}^{S_m \times d_v}, \tag{1}$$

where $\mathbf{Q}_m^{(l,h)}, \mathbf{K}_m^{(l,h)}, \mathbf{V}_m^{(l,h)}$ are the query, key, and value matrices, $d_k$ is the key/query dimension, $d_v$ is the value/head dimension, and $S_m$ is the number of tokens in the tokenized sequence of prompt $P_m$. We then extract the activations at the last token and average them across all $M$ prompts to obtain the task embedding for each head:

$$\mathbf{t}^{(l,h)} = \frac{1}{M} \sum_{m=1}^{M} \mathbf{t}_m^{(l,h)}[-1, :] \in \mathbb{R}^{d_v} \tag{2}$$

The set $\{\mathbf{t}^{(l,h)}\}_{l=1,\ldots,L;\, h=1,\ldots,H}$ constitutes the task embeddings for the given task.

**Optimization of soft head-selection parameters.** To reformulate discrete head selection as a continuous optimization problem, we introduce a learnable matrix $\mathbf{A}$ for each task, where each entry $\alpha^{(l,h)}$ serves as a *soft head-selection parameter* for attention head $h \in \{1, 2, \ldots, H\}$ in layer $l \in \{1, 2, \ldots, L\}$:

$$\mathbf{A} = \begin{bmatrix} \alpha^{(L,1)} & \cdots & \alpha^{(L,H)} \\ \vdots & \ddots & \vdots \\ \alpha^{(1,1)} & \cdots & \alpha^{(1,H)} \end{bmatrix} \in [0, 1]^{L \times H} \tag{3}$$

Each $\alpha^{(l,h)}$ controls the degree to which task-specific information is injected into the corresponding attention head. Let $\mathbf{o}^{(l,h)} \in \mathbb{R}^{d_v}$ denote the original activation of head $h$ in layer $l$ corresponding to the last token of an input prompt during optimization. We apply soft injection by interpolating between the original head activation $\mathbf{o}^{(l,h)}$ and the task embedding $\mathbf{t}^{(l,h)}$:

$$\mathbf{o}^{(l,h)} \leftarrow (1 - \alpha^{(l,h)}) \cdot \mathbf{o}^{(l,h)} + \alpha^{(l,h)} \cdot \mathbf{t}^{(l,h)}, \tag{4}$$

for all $l \in \{1, 2, \ldots, L\}$ and $h \in \{1, 2, \ldots, H\}$. During training, the underlying LLM is kept frozen and only $\mathbf{A}$ is optimized. We update $\mathbf{A}$ over a small number of gradient descent steps by minimizing the cross-entropy loss on next-token prediction, using the output logits of the intervened model. At each iteration, inference is performed on a zero-shot prompt, with the task embeddings injected according to the current values of $\mathbf{A}$. Each $\alpha^{(l,h)}$ is parameterized as the sigmoid of a trainable variable initialized to zero, yielding an initial value of 0.5–corresponding to a neutral starting point with no initial bias toward either the original activations or the task embeddings. A detailed pseudocode of this procedure is provided in Algorithm 1 of Appendix B.2.

**Zero-shot inference with task embedding injection** After learning the soft head-selection parameters $\mathbf{A}$, we use them together with the set of task embeddings $\{\mathbf{t}^{(l,h)}\}_{l,h}$ to guide the LLM in performing tasks without any input-output demonstrations in the prompt. The soft injection is applied in the same manner as during the optimization stage (Equation 4) but only once–at the last token of the initial input prompt–under the assumption that KV cache (Pope et al., 2023) is enabled during autoregressive decoding. No further injection is applied in subsequent decoding steps. This one-time injection embeds task-relevant information into the KV cache at the start of generation, enabling the model to produce the remaining tokens without additional intervention. If KV cache is disabled, the injection must be reapplied at each decoding step to the last token of the initial prompt.

---

[1] A detailed discussion of related work is provided in Appendix A.

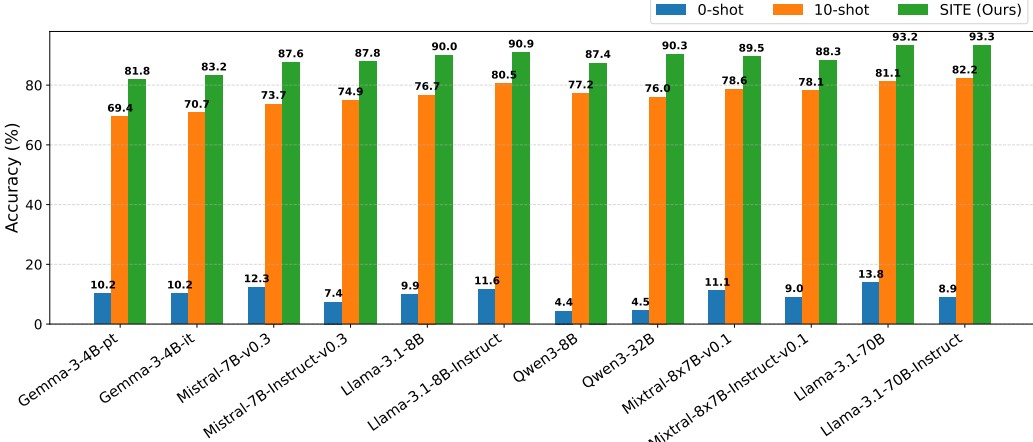

Figure 2: **Average performance across 57 ICL tasks for 12 backbone LLMs.** For each backbone model, the performance of our method is presented along with 0-shot and 10-shot baselines. Average accuracies are annotated above each bar. Task-wise results for all 57 tasks–comparing our method with the 0-shot and 10-shot baselines–are provided in Table 1 for Llama-3.1-8B, and in Tables 13-23 of Appendix C for the remaining 11 LLMs.

## 3 EXPERIMENTS

### 3.1 EXPERIMENTAL SETUP

**Implementation details.** For each task, the dataset is split into training, validation, and test sets following the split ratio used in FV (Todd et al., 2023); only the training and validation sets are used to construct task embeddings and optimize the soft head-selection parameters. Task embeddings are constructed by averaging head activations from $M = 50$ prompts, each containing $N = 10$ input–output pairs. The soft head-selection parameters are optimized with a learning rate of 0.2 for 400 iterations across all 12 LLMs, without hyperparameter tuning for individual models. During training, the LLM is kept frozen, and only the soft head-selection parameters are updated using the Adam (Kingma, 2014) optimizer without any regularization. Checkpoints are selected based on validation loss, evaluated every 50 iterations. To minimize randomness, we use greedy decoding for all models and methods.

**Models.** In this study, we apply our method to 12 LLMs spanning four model families–Llama 3.1 (Grattafiori et al., 2024), Mistral (Jiang et al., 2023; 2024), Qwen3 (Yang et al., 2025), Gemma-3 (Team et al., 2025)–covering three variation types and model sizes ranging from 4B to 70B parameters. The full list of models is provided in Table 6 of Appendix B.1. In comparison with prior task embedding injection methods, FV and MTV (Huang et al., 2024), we restrict evaluation to Llama-3.1-8B due to the high computational cost of FV. For both baselines, we follow the default configurations from the official code and papers, with one minor adjustment to MTV to match our setting: task embeddings are constructed from $M = 50$ prompts with $N = 10$ shots each, instead of the original 100 prompts with 4 shots. Additional implementation details are provided in Appendix B.2.

**Tasks and prompt templates.** We evaluate our method on all 57 ICL tasks provided in the official code repository of FV, comprising 29 abstractive and 28 extractive tasks. Abstractive tasks require the LLM to generate information that is not explicitly present in the prompt, while extractive tasks involve retrieving the answer directly from it. For all experiments, we adopt FV's default ICL prompt template: `'Q:{$x_{i1}$}\nA:{$y_{i1}$}\n\n ... Q:{$x_{iN}$}\nA:{$y_{iN}$}\n\nQ:{$x_{iq}$}\nA:'`, where each {$x_{ik}$} and {$y_{ik}$} is replaced with the corresponding input-output pair. To assess the robustness of our method to prompt formatting, we also evaluate it using four alternative prompt templates, with results reported in Table 12 of Appendix B.4. Detailed descriptions of all 57 tasks and the prompt templates are also provided in Appendix B.

## 3.2 EXPERIMENTAL RESULTS

We applied our soft injection method to 12 LLMs and report the average performance across all 57 ICL tasks in Figure 2. Despite using 0-shot prompts at inference time, our method consistently outperforms both 0-shot and 10-shot baselines across all 12 models. It achieves an average performance gain of 10.2%-14.3% over the 10-shot baseline, demonstrating strong performance across different model families, variation types, and scales. As explained earlier, we intentionally avoided hyperparameter tuning for each model and focused on validating the general potential of our soft injection method. These results suggest that our method, SITE, may offer a promising alternative to few-shot in-context learning, eliminating the need for prompt-based demonstrations at inference time. For a detailed breakdown, task-wise results for all 57 ICL tasks and 12 LLMs–comparing our method with the 0-shot and 10-shot baselines–are provided in Table 1 for Llama-3.1-8B, and in Tables 13-23 of Appendix C for the remaining 11 LLMs.

Table 1 also compares our method with other task embedding injection approaches, with tasks grouped into abstractive and extractive categories. For both categories, FV and MTV significantly outperform the 0-shot baseline, demonstrating the effectiveness of task embedding injection. However, neither method exceeds the 10-shot baseline, which is a fair point of comparison given that their task embeddings are constructed using 10-shot prompts. In contrast, our method achieves substantially higher performance than the 10-shot baseline in both categories. On abstractive tasks, it achieves either the best or second-best performance on all 29 tasks, with an average performance gain of 11.9%. On extractive tasks, it achieves the best performance on all 28 tasks, with an average gain of 14.8%. To further assess generality beyond the 57 ICL tasks, we extend the evaluation to the challenging MMLU-Pro (Wang et al., 2024) benchmark, where results in Appendix D show that our method achieves higher accuracies than the 10-shot baseline, reinforcing its robustness.

Table 1: **Comparison of injection-based methods on 57 ICL tasks using Llama-3.1-8B.** Three injection methods (FV, MTV, Ours) are evaluated along with 0-shot and 10-shot baselines. (a) Results on 29 abstractive tasks. (b) Results on 28 extractive tasks. The best results are shown in **bold**, and the second-best results are underlined.

(a) Abstractive task results

| Task Name | Vanilla | | 0-shot + Injection | | |
|---|---|---|---|---|---|
| | 0-shot | 10-shot | FV | MTV | Ours |
| AG_News | 0.4 | 79.1 | 0.0 | 78.4 | **88.7** |
| Antonym | 0.0 | 69.6 | 50.6 | 64.5 | **71.2** |
| Capitalize | 5.3 | 99.4 | 92.9 | 99.4 | **100.0** |
| Capitalize_First_Letter | 10.0 | **100.0** | 53.5 | 99.4 | **100.0** |
| Capitalize_Last_Letter | 1.2 | 24.0 | 0.0 | 50.3 | **93.0** |
| Capitalize_Second_Letter | 1.2 | 28.5 | 0.0 | 46.7 | **97.0** |
| Commonsense_QA | 40.3 | 72.3 | 23.6 | 60.7 | **62.7** |
| Country-Capital | 4.8 | 92.9 | 35.7 | 92.9 | **95.2** |
| Country-Currency | 0.0 | 78.6 | 0.0 | 71.4 | **81.0** |
| English-French | 0.5 | **81.7** | 2.2 | 75.6 | **81.7** |
| English-German | 1.2 | **75.5** | 2.5 | 63.6 | 69.3 |
| English-Spanish | 0.2 | **84.1** | 2.3 | 70.0 | 83.1 |
| Landmark-Country | 0.0 | **92.6** | 24.0 | 85.1 | 86.9 |
| Lowercase_First_Letter | 0.0 | 99.4 | 50.3 | 99.4 | **100.0** |
| Lowercase_Last_Letter | 0.0 | 39.8 | 0.0 | 53.8 | **96.5** |
| National_Parks | 0.0 | **86.3** | 46.3 | 81.1 | 81.1 |
| Next_Capital_Letter | 0.6 | 2.9 | 0.0 | 2.3 | **98.8** |
| Next_Item | 2.1 | **97.9** | 63.8 | 95.7 | 97.9 |
| Park-Country | 0.0 | **89.8** | 63.7 | 82.2 | 84.7 |
| Person-Instrument | 0.0 | 83.2 | 2.8 | 82.2 | **87.9** |
| Person-Occupation | 0.0 | 64.5 | 7.0 | 73.8 | **80.8** |
| Person-Sport | 0.0 | 95.5 | 0.0 | 94.0 | **97.0** |
| Present-Past | 3.3 | **100.0** | 75.4 | 98.4 | **100.0** |
| Prev_Item | 2.1 | **97.9** | 61.7 | 93.6 | 97.9 |
| Product-Company | 0.0 | 87.2 | 11.0 | **89.9** | 88.1 |
| Sentiment | 0.0 | 95.1 | 0.0 | 92.7 | **96.3** |
| Singular-Plural | 2.3 | **100.0** | 86.1 | 90.7 | **100.0** |
| Synonym | 1.8 | 50.3 | 21.7 | 44.9 | **53.0** |
| Word_Length | 0.0 | 38.6 | 0.0 | 26.3 | **82.5** |
| Average | 2.7 | 76.1 | 26.8 | 74.4 | **88.0** |

(b) Extractive task results

| Task Name | Vanilla | | 0-shot + Injection | | |
|---|---|---|---|---|---|
| | 0-shot | 10-shot | FV | MTV | Ours |
| Adjective_V_Verb_3 | 14.3 | 87.6 | 36.7 | 90.5 | **99.5** |
| Adjective_V_Verb_5 | 9.1 | 84.8 | 27.6 | 71.4 | **98.1** |
| Alphabetically_First_3 | 21.9 | 42.4 | 43.3 | 34.3 | **57.1** |
| Alphabetically_First_5 | 16.7 | 18.6 | 31.0 | 25.2 | **90.5** |
| Alphabetically_Last_3 | 16.2 | 36.2 | 29.5 | 30.5 | **47.6** |
| Alphabetically_Last_5 | 10.5 | 23.3 | 14.8 | 21.9 | **44.3** |
| Animal_V_Object_3 | 12.4 | 79.5 | 55.2 | 86.2 | **99.0** |
| Animal_V_Object_5 | 19.1 | 81.0 | 52.4 | 73.8 | **98.1** |
| Choose_First_Of_3 | 52.9 | 98.6 | 99.1 | 99.1 | **100.0** |
| Choose_First_Of_5 | 52.4 | 97.6 | 97.6 | 98.6 | **100.0** |
| Choose_Last_Of_3 | 1.0 | 97.6 | 61.4 | 92.9 | **100.0** |
| Choose_Last_Of_5 | 3.8 | 94.3 | 53.8 | 85.7 | **100.0** |
| Choose_Middle_Of_3 | 2.9 | 52.4 | 18.1 | 47.6 | **99.0** |
| Choose_Middle_Of_5 | 4.3 | 33.3 | 7.6 | 22.4 | **90.0** |
| Color_V_Animal_3 | 16.7 | 95.7 | 33.8 | 97.6 | **99.5** |
| Color_V_Animal_5 | 15.7 | 91.0 | 29.5 | 94.8 | **99.5** |
| Concept_V_Object_3 | 14.3 | 83.8 | 48.6 | 78.1 | **99.5** |
| Concept_V_Object_5 | 17.6 | 83.3 | 25.7 | 69.5 | **93.8** |
| Conll2003_Location | 21.8 | 87.7 | 45.3 | 89.0 | **94.3** |
| Conll2003_Organization | 39.3 | 77.5 | 58.7 | 70.1 | **91.4** |
| Conll2003_Person | 12.4 | 91.7 | 40.2 | 92.9 | **97.3** |
| Fruit_V_Animal_3 | 23.3 | 82.9 | 37.1 | 88.1 | **99.0** |
| Fruit_V_Animal_5 | 10.0 | 79.1 | 22.9 | 71.9 | **99.5** |
| Object_V_Concept_3 | 17.6 | 97.6 | 38.1 | 96.2 | **100.0** |
| Object_V_Concept_5 | 5.7 | 92.4 | 29.5 | 90.0 | **98.1** |
| Squad_Val | 39.4 | 85.4 | 5.0 | 65.5 | **86.7** |
| Verb_V_Adjective_3 | 11.4 | 94.3 | 41.4 | 89.1 | **97.6** |
| Verb_V_Adjective_5 | 1.9 | 93.8 | 24.8 | 85.2 | **99.0** |
| Average | 17.3 | 77.3 | 39.6 | 73.5 | **92.1** |

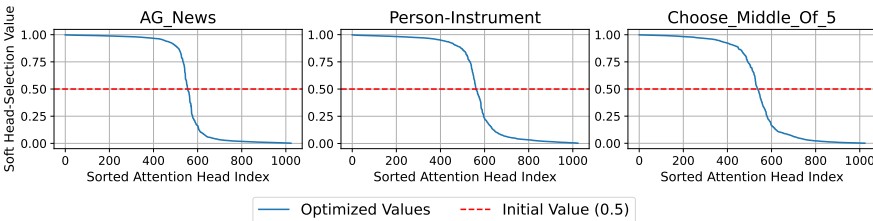

Figure 3: **Optimized values of soft head-selection parameters for three ICL tasks.** Each plot shows the optimized soft head-selection values for all 1024 attention heads in Llama-3.1-8B, sorted in descending order. Dashed lines indicate the initial value of 0.5 assigned to all selection parameters at the start of training. Results for all 57 tasks are provided in Figures 6-8 of Appendix E.1.

## 4 ANALYSIS OF TASK-RELEVANT ATTENTION HEADS THROUGH SITE

Attention head attribution, a branch of mechanistic interpretability research, investigates the functional roles of attention heads in deep neural networks (Hao et al., 2021; Olsson et al., 2022; Todd et al., 2023; Gandelsman et al., 2023; Park et al., 2024; Zhou et al., 2024; Wu et al., 2024; Elhelo & Geva, 2024). In this section, we show that since SITE reformulates head selection as a continuous optimization problem and demonstrates strong task performance, it not only identifies task-relevant attention heads but also clarifies what it means for a head to be *task-relevant*. In Section 4.1, we show that the learned soft head-selection parameters tend to converge toward binary values (0 or 1), effectively identifying task-relevant attention heads. In Section 4.2, we compare task-specific and task-agnostic perspectives and find that the task-relevant roles of attention heads are better explained from the task-specific viewpoint. While the analyses in this section are based on Llama-3.1-8B, the findings generalize to other larger LLMs, as demonstrated in Appendix E.2 and Appendix F.2.

### 4.1 IDENTIFICATION OF TASK-RELEVANT ATTENTION HEADS

Figure 3 shows the optimized values of the soft head-selection parameters for all 1024 attention heads in Llama-3.1-8B, sorted in descending order. Results are presented for three selected ICL tasks, with the dashed horizontal lines indicating the initial value of 0.5 assigned to all parameters at the start of training. Although the Adam optimizer without regularization does not explicitly encourage extreme values, we observe that approximately 80–90% of the parameters converge close to either 0 or 1 across all tasks. This pattern likely arises from two factors: (1) each parameter is defined as the sigmoid of a trainable scalar, smoothly mapping values to $[0, 1]$; and (2) consistent benefits from injecting–or not injecting–into specific heads lead gradient descent to push the parameters toward extremes. A similar pattern is observed across all 57 tasks (see Appendix E.1).

To assess whether heads with optimized values near 1 are indeed task-relevant, we evaluate the *hard injection* variant by thresholding the soft head-selection parameters at 0.5 and using the resulting binary values (0 or 1) to inject task embeddings during inference. Hard injection yields an average accuracy of 88.9% across the 57 ICL tasks, slightly below the 90.0% achieved by our soft injection approach (see Table 25 of Appendix E.3 for task-wise results of hard injection; Table 26 of Appendix E.4 provides further analysis comparing against its complementary variant). These results provide two key insights: (1) the learned soft head-selection parameters effectively identify *task-relevant attention heads*; and (2) the 10–20% of heads with intermediate values between 0 and 1 also contribute to task performance in some cases, explaining the 1.1% average performance gap.

### 4.2 TASK-SPECIFIC VS. TASK-AGNOSTIC: WHICH BETTER EXPLAINS HEAD ROLES?

Our experiments show that our method effectively identifies task-relevant attention heads in LLMs, yielding substantial gains without in-prompt demonstrations. However, an important question remains: *What does it truly mean for an attention head to be 'task-relevant'?* Specifically, *do heads selected for one task generalize across a broad range of tasks, or only to those with similar characteristics?* Prior work has primarily explored *task-agnostic* heads–those that generalize well across a broad range of tasks–which supports interpretability but often underperforms in practice (Olsson et al., 2022; Todd et al., 2023; Singh et al., 2024). In contrast, recent approaches such as

MTV (Huang et al., 2024) employ *task-specific* heads to improve performance, but offer limited insight into how broadly these heads generalize. As a result, the field still lacks a clear understanding of whether task-specific or task-agnostic perspectives more accurately reflect the functional roles of attention heads. To address this gap, we conduct two analyses: (1) a *cross-task analysis*, which tests generalization by applying soft head-selection parameters derived from one task to another (changing only *where* task information is injected), and (2) a *task-agnostic analysis*, which evaluates performance using a single, shared set of head-selection parameters trained across all 57 tasks.

Table 2: **Cross-task analysis for seven evaluation tasks.** For each evaluation task, soft head-selection parameters are replaced with those from other tasks–changing *where* task information is injected, but not *what* is injected. Among the 57 tasks, the table reports the top-3 and bottom-3 head-selection tasks ranked by accuracy. More results are provided in Tables 27-30 of Appendix F.

| Evaluation Task | Task Description | Top-3 Head-Selection Tasks (Accuracy, %) | Bottom-3 Head-Selection Tasks (Accuracy, %) |
|---|---|---|---|
| Adjective_V_Verb_5 | Select the adjective from a list of 5 words (1 adjective, 4 verbs) | Adjective_V_Verb_5 (98.1) Adjective_V_Verb_3 (96.7) Animal_V_Object_5 (78.6) | Verb_V_Adjective_3 (1.4) Verb_V_Adjective_5 (4.8) English-French (5.7) |
| Verb_V_Adjective_5 | Select the verb from a list of 5 words (1 verb, 4 adjectives) | Verb_V_Adjective_5 (99.0) Verb_V_Adjective_3 (99.0) Color_V_Animal_5 (81.4) | Adjective_V_Verb_3 (1.0) Antonym (7.6) Adjective_V_Verb_5 (9.5) |
| Alphabetically_First_5 | Select the word that comes first in alphabetical order from a list of 5 words | Alphabetically_First_5 (90.5) Alphabetically_First_3 (43.8) Commonsense_QA (29.5) | Alphabetically_Last_5 (5.7) Alphabetically_Last_3 (8.1) Park-Country (10.5) |
| Alphabetically_Last_5 | Select the word that comes last in alphabetical order from a list of 5 words | Alphabetically_Last_5 (44.3) Alphabetically_Last_3 (40.5) Commonsense_QA (25.7) | Alphabetically_First_5 (0.0) Alphabetically_First_3 (8.6) English-German (12.9) |
| English-French | Translate the given English word into French | English-French (81.7) English-German (80.3) English-Spanish (80.0) | Person-Occupation (19.4) Person-Instrument (34.0) Next_Capital_Letter (38.9) |
| English-German | Translate the given English word into German | English-French (71.4) English-German (69.3) English-Spanish (68.3) | Person-Occupation (3.4) Prev_Item (23.6) Next_Capital_Letter (26.2) |
| English-Spanish | Translate the given English word into Spanish | English-French (84.4) English-Spanish (83.1) English-German (82.1) | Person-Instrument (39.1) Person-Occupation (39.7) Next_Capital_Letter (40.8) |

**Cross-task analysis.** We define the *evaluation task* as the task being solved (i.e., the task that provides the input query), and the *head-selection task* as the task providing the soft head-selection parameters. For a given evaluation task, we use its own task embedding but apply head-selection parameters optimized for a different head-selection task. This setup keeps the injected content fixed while *changing only the injection locations*. Table 2 presents the results of the cross-task analysis for seven evaluation tasks, along with brief task descriptions to help interpretation.

Table 3: **Comparison of task-specific vs. task-agnostic head-selection on 57 ICL tasks.** Average accuracy is shown for 29 abstractive, 28 extractive, and all 57 tasks. Results include 0-shot and 10-shot baselines, and our method with task-specific and task-agnostic head-selection.

| Task Type | 0-shot | 10-shot | Ours | Ours (Task-Agnostic) |
|---|---|---|---|---|
| Abstractive | 2.7 | 76.1 | **88.0** | 73.4 |
| Extractive | 17.3 | 77.3 | **92.1** | 77.4 |
| All (57 tasks) | 9.9 | 76.7 | **90.0** | 75.3 |

Each evaluation task is assessed using head-selection parameters from all 57 head-selection tasks, and the top-3 and bottom-3 are reported based on accuracy. Notably, for the evaluation task Adjective_V_Verb_5, the top-3 head-selection tasks include semantically similar tasks such as Adjective_V_Verb_5, Adjective_V_Verb_3, while the bottom-3 include semantically dissimilar tasks such as Verb_V_Adjective_3, Verb_V_Adjective_5. Similar trends are observed for Verb_V_Adjective_5, Alphabetically_First_5, and Alphabetically_Last_5. For translation tasks–English-French, English-German, English-Spanish–the top and bottom head-selection tasks are largely consistent across the board. Interestingly, for these translation tasks, the performance using cross-task head-selection parameters closely matches or even exceeds that of using their own, suggesting that injection locations transfer well across closely related tasks. Additional results for Llama-3.1-8B and other larger LLMs are provided in Tables 27-30 of Appendix F.

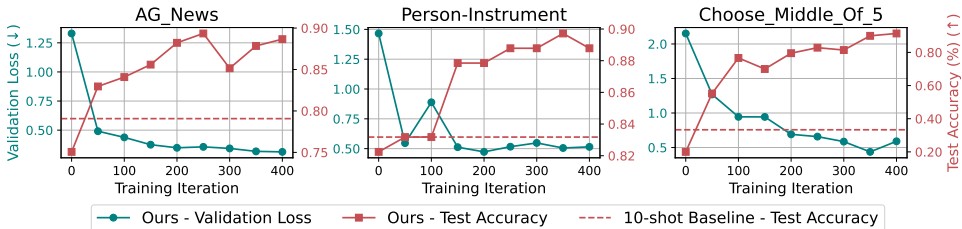

Figure 4: **Training dynamics of soft head-selection parameters for three ICL tasks.** Validation loss (left y-axis) and test accuracy (right y-axis) are plotted over 400 training iterations for AG_News, Person-Instrument, and Choose_Middle_Of_5. Dashed lines indicate the 10-shot baseline accuracies for reference. Plots for all 57 tasks are provided in Figures 12-14 of Appendix G.1.

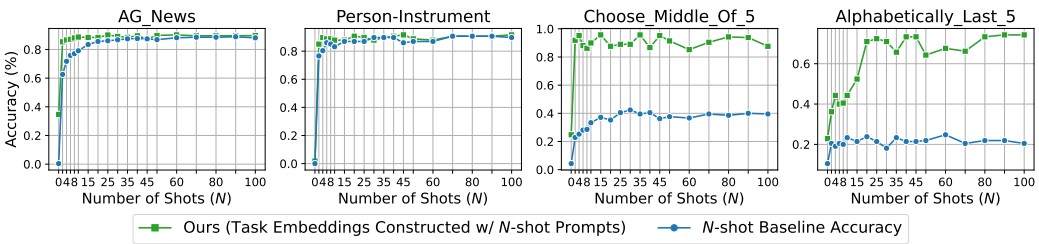

Figure 5: **Impact of shot count on task performance for four ICL tasks.** The plots show the performance of the $N$-shot baseline as the shot count ($N$) increases from 0 to 100. For comparison, our method is also evaluated using different values of $N$ used for task embedding construction (default: $N=10$), while keeping $M=50$ fixed.

**Task-agnostic head-selection analysis.** To derive task-agnostic soft head-selection parameters, we optimize them across all 57 tasks by randomly sampling a task at each training iteration. During this process, we use task-specific task embeddings (i.e., what to inject) while sharing a single set of task-agnostic head-selection parameters (i.e., where to inject). To ensure convergence, we increase the number of training iterations from 400 to 1000. Table 3 presents the results along with 0-shot and 10-shot baselines, as well as our method using task-specific head-selection parameters. We observe a substantial drop in performance when using task-agnostic head-selection parameters. This result is consistent with the cross-task analysis in Table 2, where performance significantly degraded when head-selection parameters were taken from semantically dissimilar tasks. Together, these findings suggest that task-specific perspectives more accurately capture the functional roles of attention heads, while task-agnostic approaches tend to obscure important task-dependent behaviors.

## 5 EMPIRICAL FINDINGS AND EFFICIENCY ANALYSIS

In this section, we provide four additional analysis results, all based on Llama-3.1-8B.

**Training dynamics of head selection.** In Figure 4, we plot the validation loss and test accuracy of our method during the optimization of soft head-selection parameters. The figure shows training dynamics for three selected tasks, with the 10-shot baseline accuracies included for reference. While the training dynamics vary slightly across tasks, the overall trends are consistent: (1) At the beginning of training, the injection already yields relatively high accuracy compared to the 0-shot results in Table 1. This is because all soft head-selection parameters are initialized to 0.5, meaning the task embeddings are equally mixed with the original head activations across all attention heads (see Equation 4). (2) As training progresses, validation loss decreases and test accuracy improves, indicating that gradient descent effectively tunes the selection parameters to identify meaningful head positions for injecting task-specific information. The full set of training curves for all 57 tasks is provided in Appendix G.1, and some selected results for larger LLMs are included in Appendix G.2.

**Impact of shot count on task performance.** Increasing the number of in-prompt examples (or shots) is a common strategy for improving few-shot ICL performance. While some studies (Agarwal et al., 2024) report further gains as the number of shots increases from *a few to many*, oth-

ers (Zhang et al., 2025) observe that ICL performance plateaus after only a few examples. A plausible explanation–also suggested by recent work (Zou et al., 2024)–is that the benefit of additional shots varies across tasks. To investigate this, we analyze whether increasing the shot count in the $N$-shot baseline can close the performance gap with our method. Figure 5 presents results for four selected tasks, with our method also evaluated using varying values of $N$ used for task embedding construction. For AG_News, and Person-Instrument, $N$-shot baseline performance improves as $N$ increase, eventually matching ours at 100 shots. In contrast, tasks such as Choose_Middle_Of_5 and Alphabetically_Last_5 show limited performance gains even with many-shot prompting. This suggests that increasing the number of shots from a few to many is not always effective, particularly given the additional time and memory costs. Interestingly, our method reaches peak performance with as few as $N = 2$ for most tasks, with Alphabetically_Last_5 being a notable exception where larger $N$ yields further improvement.

**Robustness to prompt count in task embedding construction.** We construct task embeddings by averaging activations from $M = 50$ ICL prompts, each containing $N = 10$ input-output pairs. To assess the impact of the prompt count $M$ in task embedding construction, we conduct an ablation study by varying $M$ while keeping $N = 10$ fixed. As shown in Table 4, even a single 10-shot prompt ($M = 1$) achieves strong performance, outperforming the vanilla 10-shot baseline by 12.0%. Performance saturates beyond $M = 5$, closely matching that of our default setting ($M = 50$). These results suggest that the strength of our method stems not from aggregating many demonstrations into the task embeddings, but from identifying effective soft head-selection parameters for each task.

Table 4: **Ablation of prompt count $M$ used in task embedding construction.** Average accuracy is reported for 29 abstractive, 28 extractive, and all 57 tasks. We ablate our method by constructing task embeddings using 10-shot prompts ($N = 10$), varying $M$ across $\{1, 3, 5, 10, 50\}$.

| Task Type | 0-shot | 10-shot | Ours ($M$=1) | Ours ($M$=3) | Ours ($M$=5) | Ours ($M$=10) | Ours ($M$=50, default) |
|---|---|---|---|---|---|---|---|
| Abstractive (29 tasks) | 2.7 | 76.1 | 86.5 | 87.1 | 87.4 | 87.2 | 88.0 |
| Extractive (28 tasks) | 17.3 | 77.3 | 91.0 | 91.7 | 92.1 | 92.3 | 92.1 |
| All (57 tasks) | 9.9 | 76.7 | 88.7 | 89.4 | 89.7 | 89.7 | 90.0 |

**Computational efficiency.** To evaluate the efficiency of our method, Table 5 reports the total runtime as the number of test prompts increases. Since both task embeddings and soft head-selection parameters are computed only once (Stages 1 and 2 of our method) and reused, their cost does not grow with the number of test prompts. This allows our method to scale efficiently, maintaining total runtime close to the 0-shot baseline and substantially lower than the 10-shot baseline. In contrast, FV incurs significant overhead from its task embedding construction, while MTV is slowed by a subopti-

Table 5: **Runtime comparison as the number of test prompts increases.** Total runtime (in minutes) is reported for 1000, 5000, and 10000 prompts on AG_News using Llama-3.1-8B. Our method scales efficiently as the number of test prompts increases. All runtimes were measured on a single NVIDIA A6000 GPU.

| # Test Prompts | 0-shot | 10-shot | FV | MTV | Ours |
|---|---|---|---|---|---|
| 1000 | **15.7** | 24.8 | 269.8 | 41.9 | 22.0 |
| 5000 | **78.3** | 124.2 | 332.7 | 169.4 | 88.6 |
| 10000 | **156.7** | 248.4 | 411.2 | 328.8 | 172.0 |

mal loop structure in its original codebase; even with code-level optimization, its runtime remains slightly higher than ours. Our method is also lightweight at inference, requiring only 0.5 MB in float32 precision to store the task embeddings and head-selection parameters for Llama-3.1-8B. Overall, our approach achieves strong performance gains over few-shot ICL while maintaining the time and memory efficiency of 0-shot inference, particularly as the number of test prompts increases.

## 6 CONCLUSION

This paper introduces SITE (Soft Injection of Task Embeddings), a method that injects pre-computed task embeddings into LLMs using pre-optimized soft head-selection parameters. Extensive experiments across 57 tasks and 12 LLMs show that SITE significantly outperforms 10-shot ICL, without requiring any in-prompt demonstrations at inference time. Beyond its performance gains, SITE provides a novel lens for analyzing task-relevant attention heads, revealing their task-specific functional roles. Overall, SITE offers a scalable and effective alternative to prompt-based ICL, with promising implications for both practical deployment and interpretability research in LLMs.

## ETHICS STATEMENT

Our method, like other task embedding injection approaches, uses pre-computed task embeddings– which encode task-relevant information–instead of in-prompt input-output examples at inference time. In real-world applications, sharing task embeddings rather than raw examples can offer benefits such as improved inference efficiency, enhanced privacy protection, and stronger data security. However, such embeddings could also be misused to conceal harmful instructions and be distributed to users without their awareness. To mitigate these risks, we recommend implementing safeguards, such as pre-release embedding screening, well-defined usage policies, and runtime output filtering tied to embedding identities, prior to the deployment of systems that rely on pre-computed embeddings.

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

CONTENTS

# A RELATED WORK

## A.1 IN-CONTEXT LEARNING IN LARGE LANGUAGE MODELS

In-context learning (ICL) is a paradigm where LLMs perform tasks by conditioning on input-output examples in the prompt, without updating model weights. This enables LLMs to leverage their pre-trained knowledge to dynamically infer tasks or patterns during inference. Numerous studies (Xie et al., 2021; Olsson et al., 2022; Von Oswald et al., 2023; Todd et al., 2023; Yan et al., 2023; Falck et al., 2024) have sought to explain or better understand the mechanisms behind the strong ICL capabilities of LLMs. Several of these highlight the roles of specialized attention heads, such as induction heads (Olsson et al., 2022) or function vector heads (Todd et al., 2023), as critical for task execution. While our work is closely related to this perspective, we reconceptualize head attribution as a performance optimization problem, rather than merely treating it as an analytical tool for understanding LLM dynamics. This reconceptualization enabled us to develop a scalable approach that identifies task-relevant attention heads via gradient-based optimization over soft head-selection parameters.

## A.2 TASK EMBEDDING INJECTION IN LARGE LANGUAGE MODELS

Previous studies have shown that task information is often encoded in the hidden representation of the last token in a few-shot ICL prompt, and that injecting this information, typically in the form of a *task embedding* (also known as a task vector or function vector), can guide LLMs to perform tasks without any in-prompt demonstrations (Hendel et al., 2023; Todd et al., 2023; Huang et al., 2024). FV (Todd et al., 2023) constructs a task embedding by aggregating activations from a few selected attention heads and injects it into the model by adding it to the hidden representation at a fixed layer, typically around one-third of the model depth. The attention heads are selected based on their individual impact on ICL performance, which is computationally expensive given the large number of heads in modern LLMs. In addition, FV's performance is constrained by the heuristic choice of injection location. MTV (Multimodal Task Vectors) (Huang et al., 2024), originally developed for vison-language models (VLMs) but naturally extensible to LLMs, addresses some of these limitations by directly replacing selected head activations with task embeddings, using reinforcement learning to optimize the head-sampling distribution. Nevertheless, MTV relies on repeated sampling and hard replacement, which may limit both efficiency and flexibility. In contrast, our method introduces a *soft injection* mechanism that blends each head's original activation with a task embedding using learnable interpolation weights. This continuous formulation avoids discrete selection, enables efficient gradient-based optimization, and allows for head-wise control over how much task information is injected—preserving the original activation when helpful and overriding it when necessary.

# B  ADDITIONAL EXPERIMENTAL SETUP

## B.1  12 LARGE LANGUAGE MODELS USED FOR EVALUATION

Our evaluation covers 12 large language models in total. From the Llama-3.1 family (Grattafiori et al., 2024), we include Llama-3.1-8B, Llama-3.1-8B-Instruct, Llama-3.1-70B, and Llama-3.1-70B-Instruct. From the Mistral family (Jiang et al., 2023; 2024), we include Mistral-7B-v0.3, Mistral-7B-Instruct-v0.3, Mixtral-8x7B-v0.1, and Mixtral-8x7B-Instruct-v0.1. From the Qwen3 family (Yang et al., 2025), we include Qwen3-8B and Qwen3-32B. From the Gemma-3 family (Team et al., 2025), we include Gemma-3-4B-pt and Gemma-3-4B-it. The corresponding variation type, the number of attention layers ($L$), and the number of heads per layer ($H$) for each model are provided in Table 6.

Table 6: **Models used for evaluation.** We consider 12 large language models spanning four model families, three variation types, and model sizes ranging from 4B to 70B parameters.

| Model Family | Model Name | Variation Type | Attention Layers ($L$) | Heads per Layer ($H$) |
|---|---|---|---|---|
| Llama-3.1 | Llama-3.1-8B | Pretrained (Base) | 32 | 32 |
| | Llama-3.1-8B-Instruct | Instruction-tuned (Chat/Alignment) | 32 | 32 |
| | Llama-3.1-70B | Pretrained (Base) | 80 | 64 |
| | Llama-3.1-70B-Instruct | Instruction-tuned (Chat/Alignment) | 80 | 64 |
| Mistral | Mistral-7B-v0.3 | Pretrained (Base) | 32 | 32 |
| | Mistral-7B-Instruct-v0.3 | Instruction-tuned (Chat/Alignment) | 32 | 32 |
| | Mixtral-8x7B-v0.1 | Mixture of Experts, Pretrained | 32 | 32 |
| | Mixtral-8x7B-Instruct-v0.1 | Mixture of Experts, Instruction-tuned | 32 | 32 |
| Qwen3 | Qwen3-8B | Instruction-tuned (Chat/Alignment) | 36 | 32 |
| | Qwen3-32B | Instruction-tuned (Chat/Alignment) | 64 | 64 |
| Gemma-3 | Gemma-3-4B-pt | Pretrained (Base) | 34 | 8 |
| | Gemma-3-4B-it | Instruction-tuned (Chat/Alignment) | 34 | 8 |

## B.2  IMPLEMENTATION DETAILS FOR FV, MTV, AND SITE

As described in Section 3.1, we follow the default configurations provided in the official code and papers for both FV (Todd et al., 2023) and MTV (Huang et al., 2024). For FV, we adopt the hyperparameter settings specified for Llama-2-7B in the official repository and apply them to experiments on Llama-3.1-8B, as both models share the same number of attention layers and attention heads per layer. For MTV, we train the head-sampling distribution on the full training dataset for 100 iterations. After training, we independently sample 10 candidate head configurations, evaluate each on up to 100 validation examples, and select the best-performing set. To avoid excessive runtime, we cap the number of validation examples at 100–an increase from the 50 used in MTV's original repository. To ensure a fair comparison, we apply the same cap in our method, selecting the checkpoint with the lowest validation loss computed on up to 100 examples, evaluated every 50 iterations during the optimization of soft head-selection parameters. Algorithm 1 details the optimization of soft head-selection parameters (Stage 2 of our method, shown in Figure 1), where we set the number of training iterations to $J = 400$ for all 12 LLMs and all tasks. Performance of all methods is evaluated using exact match, comparing the ground-truth answer with the initial tokens of the LLM's output. For tasks unrelated to capitalization or lowercasing, we allow case-insensitive matches, following prior work (Huang et al., 2024), as a few tasks contain inconsistencies in the capitalization of their ground-truth answers.

---

**Algorithm 1** Optimization of Soft Head-Selection Parameters

---

**Require:** $L$: Number of attention layers in LLM
**Require:** $H$: Number of attention heads per layer
**Require:** $J$: Number of training iterations
**Require:** $\mathbb{T}$: Training set of tuples (0-shot prompt, ground-truth answer)
**Require:** $\{\mathbf{t}^{(l,h)}\}_{l=1,\ldots,L;\,h=1,\ldots,H}$: Task embeddings for a given task
**Require:** $\mathrm{SA}^{(l,h)}(\cdot)$: Self-attention at head $h$ in layer $l$ (before output projection)
**Require:** $W_o^{(l)}$: Output projection of the multi-head self-attention block in layer $l$
**Require:** $\mathrm{MLP}^{(l)}(\cdot)$: MLP block of layer $l$
1: Initialize soft head-selection parameters $\mathbf{A} = [\alpha^{(l,h)}]_{l=1,\ldots,L;\,h=1,\ldots,H} \in \mathbb{R}^{L \times H}$ with $\alpha^{(l,h)} = \sigma(0) = 0.5$ for all $l, h$, where $\sigma(\cdot)$ denotes the sigmoid function
2: **for** each iteration $j = 1, 2, \ldots, J$ **do**
3:     Sample $(P_j, a_j) \sim \mathbb{T}$
4:     $\mathbf{e} \leftarrow \mathrm{Tokenize}(P_j)$
5:     $\mathbf{v}_1 \leftarrow \mathrm{Embed}(\mathbf{e})$
6:     **for** all $l = 1, 2, \ldots, L$ **do**
7:        $[\mathbf{u}^{(l,1)}, \mathbf{u}^{(l,2)}, \ldots, \mathbf{u}^{(l,H)}] \leftarrow [\mathrm{SA}^{(l,1)}(\mathbf{v}_1), \mathrm{SA}^{(l,2)}(\mathbf{v}_1), \ldots, \mathrm{SA}^{(l,H)}(\mathbf{v}_1)]$
8:        **for** all $h = 1, 2, \ldots, H$ **do**
9:          $\mathbf{u}^{(l,h)}[-1,:] \leftarrow (1 - \alpha^{(l,h)}) \cdot \mathbf{u}^{(l,h)}[-1,:] + \alpha^{(l,h)} \cdot \mathbf{t}^{(l,h)}$
10:        **end for**
11:        $\mathbf{v} \leftarrow (\mathbf{u}^{(l,1)} \oplus \mathbf{u}^{(l,2)} \oplus \cdots \oplus \mathbf{u}^{(l,H)}) W_o^{(l)}$
12:        $\mathbf{v}_2 \leftarrow \mathbf{v}_1 + \mathbf{v}$
13:        $\mathbf{v} \leftarrow \mathrm{MLP}^{(l)}(\mathbf{v}_2)$
14:        $\mathbf{v}_1 \leftarrow \mathbf{v}_2 + \mathbf{v}$
15:     **end for**
16:     Compute output logits: $\mathbf{p}_j \leftarrow \mathrm{OutputProj}(\mathbf{v}_1)$
17:     $\mathcal{L}_j \leftarrow \mathrm{CrossEntropy}(\mathbf{p}_j, a_j)$
18:     Update $\mathbf{A}$ with the Adam optimizer: $\mathbf{A} \leftarrow \mathrm{Adam}(\mathbf{A}, \nabla_{\mathbf{A}} \mathcal{L}_j)$
19: **end for**
20: **Return:** Optimized soft head-selection parameters $\mathbf{A} = [\alpha^{(l,h)}]_{l=1,\ldots,L;\,h=1,\ldots,H} \in \mathbb{R}^{L \times H}$
21: *Note:* $\oplus$ denotes concatenation.
22: *Note:* $\mathbf{o}^{(l,h)}$ in Eq. 4 corresponds to $\mathbf{o}^{(l,h)} = \mathbf{u}^{(l,h)}[-1,:] \in \mathbb{R}^{d_v}$.
23: *Note:* Although Lines 8–10 are written with a loop for clarity, they are implemented as a vectorized operation in practice.

---

### B.3 OVERVIEW OF ALL 57 ICL TASKS

We evaluate our method on all 57 ICL tasks provided in the official repository of FV (Todd et al., 2023), comprising 29 abstractive and 28 extractive tasks (see Table 1 of Section 3.2). Several of these tasks originate from prior work but were filtered or reformatted by FV, including AG_News (Zhang et al., 2015), Antonym (Nguyen et al., 2017), Synonym (Nguyen et al., 2017), Commonsense_QA (Talmor et al., 2018), English-French (Conneau et al., 2017), English-German (Conneau et al., 2017), English-Spanish (Conneau et al., 2017), Landmark-Country (Hernandez et al., 2023), Person-Instrument (Hernandez et al., 2023), Person-Occupation (Hernandez et al., 2023), Person-Sport (Hernandez et al., 2023), Product-Company (Hernandez et al., 2023), Sentiment (Socher et al., 2013; Honovich et al., 2022), Conll2003_Location (Sang & De Meulder, 2003), Conll2003_Organization (Sang & De Meulder, 2003), and Conll2003_Person (Sang & De Meulder, 2003). The remaining tasks were constructed by FV. For completeness and clarity, we present task descriptions and input-output examples for all 57 tasks in Tables 7-10.

Table 7: **Task descriptions and input-output examples for 57 ICL tasks (Part 1 of 4).** This table provides task names, descriptions, and representative input-output examples for the ICL tasks used in our experiments. The remaining tasks are provided in Tables 8-10.

| Task Name | Task Description |
|---|---|
| | **Input-Output Example** |
| AG_News | Classify a news article based on its headline and opening sentences into one of: *World*, *Sports*, *Business*, or *Science/Technology*. |
| | **Input:** Surviving Biotech's Downturns Charly Travers offers advice on withstanding the volatility of the biotech sector. **Output:** Business |
| Antonym | Generate the antonym of a given word. |
| | **Input:** overnight **Output:** daytime |
| Capitalize | Capitalize the given word. |
| | **Input:** without **Output:** Without |
| Capitalize_First_Letter | Generate the first letter of a given word in capital form. |
| | **Input:** deliver **Output:** D |
| Capitalize_Last_Letter | Generate the last letter of a given word in capital form. |
| | **Input:** clean **Output:** N |
| Capitalize_Second_Letter | Generate the second letter of a given word in capital form. |
| | **Input:** amazing **Output:** M |
| Commonsense_QA | Select the most plausible answer to a commonsense question from five given options. |
| | **Input:** Sammy wanted to go to where the people were. Where might he go? a: race track   b: populated areas   c: the desert   d: apartment   e: roadblock **Output:** b |
| Country-Capital | Generate the capital city of a given country. |
| | **Input:** United States of America **Output:** Washington, D.C. |
| Country-Currency | Generate the currency used in a given country. |
| | **Input:** Singapore **Output:** Singapore Dollar (SGD) |
| English-French | Translate the given English word into French. |
| | **Input:** know **Output:** savoir |
| English-German | Translate the given English word into German. |
| | **Input:** drink **Output:** trinken |
| English-Spanish | Translate the given English word into Spanish. |
| | **Input:** sometimes **Output:** a veces |
| Landmark-Country | Generate the country of a given landmark. |
| | **Input:** South East Forests National Park **Output:** Australia |
| Lowercase_First_Letter | Generate the first letter of a given word in lowercase. |
| | **Input:** CLEVER **Output:** c |
| Lowercase_Last_Letter | Generate the last letter of a given word in lowercase. |
| | **Input:** PILLOW **Output:** w |

Table 8: **Task descriptions and input-output examples for 57 ICL tasks (Part 2 of 4).** This table continues from Table 7, providing task names, descriptions, and representative input-output examples for the ICL tasks used in our experiments. The remaining tasks are provided in Tables 9-10.

| Task Name | Task Description |
|---|---|
| | **Input-Output Example** |
| National_Parks | Generate the U.S. state of a given national park unit. |
| | **Input:** Glacier Bay National Park 
 **Output:** Alaska |
| Next_Capital_Letter | Generate the next capital letter of the first letter in a given word. |
| | **Input:** microphone 
 **Output:** N |
| Next_Item | Generate the next item in a known sequence (e.g., days, months, letters, or numbers). |
| | **Input:** Friday 
 **Output:** Saturday |
| Park-Country | Generate the country of a given national park. |
| | **Input:** Dartmoor National Park 
 **Output:** United Kingdom |
| Person-Instrument | Generate the musical instrument played by a given musician. |
| | **Input:** Andor Toth 
 **Output:** violin |
| Person-Occupation | Generate the occupation of a given individual. |
| | **Input:** Li Yining 
 **Output:** economist |
| Person-Sport | Generate the sport played by a given athlete. |
| | **Input:** Andrea Pirlo 
 **Output:** soccer |
| Present-Past | Generate the past-tense form of a given present-tense verb. |
| | **Input:** write 
 **Output:** wrote |
| Prev_Item | Generate the previous item in a known sequence (e.g., days, months, letters, or numbers). |
| | **Input:** april 
 **Output:** march |
| Product-Company | Generate the company associated with a given commercial product. |
| | **Input:** Wii Balance Board 
 **Output:** Nintendo |
| Sentiment | Generate the sentiment of a given movie review. |
| | **Input:** Very well-written and very well-acted. 
 **Output:** positive |
| Singular-Plural | Generate the plural form of a given singular noun. |
| | **Input:** island 
 **Output:** islands |
| Synonym | Generate the synonym of a given word. |
| | **Input:** identify 
 **Output:** recognize |
| Word_Length | Generate the number of letters in a given word. |
| | **Input:** discuss 
 **Output:** 7 |
| Adjective_V_Verb_3 | Select the adjective from a list of 3 words (1 adjective, 2 verbs). |
| | **Input:** prepare, faithful, develop 
 **Output:** faithful |

Table 9: **Task descriptions and input-output examples for 57 ICL tasks (Part 3 of 4).** This table continues from Tables 7-8, providing task names, descriptions, and representative input-output examples for the ICL tasks used in our experiments. The remaining tasks are provided in Table 10.

| Task Name | Task Description |
|---|---|
| | **Input-Output Example** |
| Adjective_V_Verb_5 | Select the adjective from a list of 5 words (1 adjective, 4 verbs). |
| | **Input:** remember, teach, knowledgeable, doubt, write
**Output:** knowledgeable |
| Alphabetically_First_3 | Select the word that comes first in alphabetical order from a list of 3 words. |
| | **Input:** grapefruit, thoughtful, diligent
**Output:** diligent |
| Alphabetically_First_5 | Select the word that comes first in alphabetical order from a list of 5 words. |
| | **Input:** test, prepare, hammer, beyond, pigeon
**Output:** beyond |
| Alphabetically_Last_3 | Select the word that comes last in alphabetical order from a list of 3 words. |
| | **Input:** sample, garlic, cream
**Output:** sample |
| Alphabetically_Last_5 | Select the word that comes last in alphabetical order from a list of 5 words. |
| | **Input:** about, navy, gentle, duster, green
**Output:** navy |
| Animal_V_Object_3 | Select the animal from a list of 3 words (1 animal, 2 non-animals). |
| | **Input:** lettuce, basketball, dog
**Output:** dog |
| Animal_V_Object_5 | Select the animal from a list of 5 words (1 animal, 4 non-animals). |
| | **Input:** soda, rice, potato, snorkel, sloth
**Output:** sloth |
| Choose_First_Of_3 | Select the first word from a list of 3 words. |
| | **Input:** ostrich, since, out
**Output:** ostrich |
| Choose_First_Of_5 | Select the first word from a list of 5 words. |
| | **Input:** reach, puzzle, passionate, silver, complete
**Output:** reach |
| Choose_Last_Of_3 | Select the last word from a list of 3 words. |
| | **Input:** salmon, rice, socks
**Output:** socks |
| Choose_Last_Of_5 | Select the last word from a list of 5 words. |
| | **Input:** spicy, cowardly, hoop, komodo, toward
**Output:** toward |
| Choose_Middle_Of_3 | Select the middle word from a list of 3 words. |
| | **Input:** garlic, candle, argue
**Output:** candle |
| Choose_Middle_Of_5 | Select the middle word from a list of 5 words. |
| | **Input:** table, qualify, airplane, harmonious, happy
**Output:** airplane |
| Color_V_Animal_3 | Select the color from a list of 3 words (1 color, 2 animals). |
| | **Input:** camel, penguin, brown
**Output:** brown |
| Color_V_Animal_5 | Select the color from a list of 5 words (1 color, 4 animals). |
| | **Input:** salamander, chinchilla, flamingo, black, tiger
**Output:** black |

Table 10: **Task descriptions and input-output examples for 57 ICL tasks (Part 4 of 4).** This table concludes the series from Tables 7-9, providing task names, descriptions, and representative input-output examples for the ICL tasks used in our experiments.

| Task Name | Task Description / Input-Output Example |
|---|---|
| Concept_V_Object_3 | Select the concept from a list of 3 words (1 abstract concept, 2 concrete entities). |
| | **Input:** radio, whimsical, robot 
 **Output:** whimsical |
| Concept_V_Object_5 | Select the concept from a list of 5 words (1 abstract concept, 4 concrete entities). |
| | **Input:** towel, map, hammock, read, blanket 
 **Output:** read |
| Conll2003_Location | Select the location entity from a given sentence. |
| | **Input:** Clinton arrives in Chicago on day of re-nomination. 
 **Output:** Chicago |
| Conll2003_Organization | Select the organization entity from a given sentence. |
| | **Input:** Advertising revenues at The Times grew 20 percent. 
 **Output:** The Times |
| Conll2003_Person | Select the person entity from a given sentence. |
| | **Input:** They contained $ 650,000 in jewelry and $ 40,000 in cash, Andrews said. 
 **Output:** Andrews |
| Fruit_V_Animal_3 | Select the fruit from a list of 3 words (1 fruit, 2 animals). |
| | **Input:** pineapple, iguana, leopard 
 **Output:** pineapple |
| Fruit_V_Animal_5 | Select the fruit from a list of 5 words (1 fruit, 4 animals). |
| | **Input:** walrus, lizard, panther, lion, cranberry 
 **Output:** cranberry |
| Object_V_Concept_3 | Select the concrete entity from a list of 3 words (1 concrete entity, 2 abstract concepts). |
| | **Input:** need, lamp, beneath 
 **Output:** lamp |
| Object_V_Concept_5 | Select the concrete entity from a list of 5 words (1 concrete entity, 4 abstract concepts). |
| | **Input:** passionate, jigsaw, remove, expensive, fearless 
 **Output:** jigsaw |
| Squad_Val | Retrieve the answer to a given question based on a provided context paragraph. |
| | **Input:** The Panthers offense, which led the NFL in scoring (500 points), was loaded with talent, boasting six Pro Bowl selections. Pro Bowl quarterback Cam Newton had one of his best seasons, throwing for 3,837 yards and rushing for 636, while recording a career-high and league-leading 45 total touchdowns (35 passing, 10 rushing), a career-low 10 interceptions, and a career-best quarterback rating of 99.4. Newton's leading receivers were tight end Greg Olsen, who caught a career-high 77 passes for 1,104 yards and seven touchdowns, and wide receiver Ted Ginn, Jr., who caught 44 passes for 739 yards and 10 touchdowns; Ginn also rushed for 60 yards and returned 27 punts for 277 yards. Other key receivers included veteran Jerricho Cotchery (39 receptions for 485 yards), rookie Devin Funchess (31 receptions for 473 yards and five touchdowns), and second-year receiver Corey Brown (31 receptions for 447 yards). The Panthers backfield featured Pro Bowl running back Jonathan Stewart, who led the team with 989 rushing yards and six touchdowns in 13 games, along with Pro Bowl fullback Mike Tolbert, who rushed for 256 yards and caught 18 passes for another 154 yards. Carolina's offensive line also featured two Pro Bowl selections: center Ryan Kalil and guard Trai Turner. 
 What position does Jerricho Cotchery play? 
 **Output:** receivers |
| Verb_V_Adjective_3 | Select the verb from a list of 3 words (1 verb, 2 adjectives). |
| | **Input:** dirty, dance, diligent 
 **Output:** dance |
| Verb_V_Adjective_5 | Select the verb from a list of 5 words (1 verb, 4 adjectives). |
| | **Input:** heavy, overcome, quick, modern, dazzling 
 **Output:** overcome |

### B.4    ABLATION STUDY ON PROMPT TEMPLATES

To assess the robustness of our method to prompt formatting, we conduct an ablation study using five prompt templates–each provided by FV (Todd et al., 2023)–including the default template used in all other experiments. These templates are listed in Table 11. For each template, we evaluate our method along with 0-shot and 10-shot baselines across all 57 tasks using Llama-3.1-8B, with results reported in Table 12. Across all five templates, our method consistently achieves strong performance, with average accuracies ranging from 89.0% to 91.2%, significantly outperforming the 10-shot baseline (76.7%-77.8%). These results demonstrate the robustness of our method to variations in prompt format.

Table 11: **Prompt templates used in the ablation study.** Each template shows how a single input-output pair $(\{x_{ik}\}, \{y_{ik}\})$ is formatted. All templates are sourced from FV (Todd et al., 2023). Template 1 serves as the default prompt format used in all main experiments.

| Prompt Template | Format of a single $(\{x_{ik}\}, \{y_{ik}\})$ pair |
|---|---|
| Template 1 (Default) | `Q:`$\{x_{ik}\}$`\nA:`$\{y_{ik}\}$`\n\n` |
| Template 2 | `question:`$\{x_{ik}\}$`\nanswer:`$\{y_{ik}\}$`\n\n` |
| Template 3 | `A:`$\{x_{ik}\}$`\nB:`$\{y_{ik}\}$`\n\n` |
| Template 4 | $\{x_{ik}\}$ `→`$\{y_{ik}\}$`\n\n` |
| Template 5 | `input:`$\{x_{ik}\}$ `output:`$\{y_{ik}\}$`\n` |

Table 12: **Results of prompt template ablation using Llama-3.1-8B.** Average accuracies for our method and the 0-shot/10-shot baselines are reported across five prompt templates. Results are shown for 29 abstractive tasks, 28 extractive tasks, and all 57 tasks, respectively. Our method consistently demonstrates strong performance across all templates.

| Task Type | Template 1 (default) | | | Template 2 | | | Template 3 | | | Template 4 | | | Template 5 | | |
|---|---|---|---|---|---|---|---|---|---|---|---|---|---|---|---|
| | 0-shot | 10-shot | Ours | 0-shot | 10-shot | Ours | 0-shot | 10-shot | Ours | 0-shot | 10-shot | Ours | 0-shot | 10-shot | Ours |
| Abstractive | 2.7 | 76.1 | 88.0 | 3.6 | 77.0 | 87.8 | 3.7 | 75.1 | 87.4 | 3.3 | 75.9 | 87.6 | 2.4 | 76.4 | 86.3 |
| Extractive | 17.3 | 77.3 | 92.1 | 26.9 | 76.6 | 90.9 | 23.2 | 79.2 | 93.0 | 5.0 | 78.7 | 94.8 | 25.0 | 79.2 | 91.7 |
| All (57 tasks) | 9.9 | 76.7 | **90.0** | 15.0 | 76.8 | **89.3** | 13.3 | 77.1 | **90.2** | 4.2 | 77.3 | **91.2** | 13.5 | 77.8 | **89.0** |

## C    TASK-WISE PERFORMANCE ACROSS 11 ADDITIONAL LLMS

Figure 2 in Section 3.2 shows the average performance of our method across all 57 tasks, compared to the 0-shot and 10-shot baselines, for all 12 LLMs listed in Table 6. Task-wise results for Llama-3.1-8B are presented in Table 1 of Section 3.2, while results for the remaining 11 LLMs are provided in Tables 13-23.

Table 13: **Task-wise performance on 57 ICL tasks using Gemma-3-4B-pt.** Our method is evaluated along with 0-shot and 10-shot baselines. (a) Results on 29 abstractive tasks. (b) Results on 28 extractive tasks. The best results are shown in **bold**, and the second-best results are underlined.

(a) Abstractive task results

| Task Name | 0-shot | 10-shot | Ours |
|---|---|---|---|
| AG_News | 0.3 | 79.7 | 84.3 |
| Antonym | 0.4 | 66.3 | 66.7 |
| Capitalize | 1.2 | 99.4 | 99.4 |
| Capitalize_First_Letter | 1.2 | 99.4 | 99.4 |
| Capitalize_Last_Letter | 0.6 | 15.8 | 71.9 |
| Capitalize_Second_Letter | 0.0 | 25.5 | 64.2 |
| Commonsense_QA | 30.5 | 67.5 | 59.8 |
| Country-Capital | 0.0 | 92.9 | 88.1 |
| Country-Currency | 0.0 | 81.0 | 78.6 |
| English-French | 0.5 | 81.2 | 75.7 |
| English-German | 0.4 | 75.5 | 69.9 |
| English-Spanish | 0.3 | 84.3 | 81.6 |
| Landmark-Country | 1.7 | 82.3 | 78.3 |
| Lowercase_First_Letter | 0.0 | 100.0 | 100.0 |
| Lowercase_Last_Letter | 0.6 | 37.4 | 94.7 |
| National_Parks | 7.4 | 79.0 | 79.0 |
| Next_Capital_Letter | 2.3 | 1.8 | 35.7 |
| Next_Item | 0.0 | 89.4 | 91.5 |
| Park-Country | 21.7 | 82.2 | 77.1 |
| Person-Instrument | 0.9 | 65.4 | 69.2 |
| Person-Occupation | 0.0 | 52.3 | 64.5 |
| Person-Sport | 0.0 | 94.0 | 98.5 |
| Present-Past | 1.6 | 100.0 | 100.0 |
| Prev_Item | 0.0 | 66.0 | 83.0 |
| Product-Company | 2.8 | 78.9 | 78.9 |
| Sentiment | 0.0 | 95.9 | 94.7 |
| Singular-Plural | 2.3 | 100.0 | 100.0 |
| Synonym | 6.8 | 50.0 | 52.3 |
| Word_Length | 0.0 | 18.1 | 28.1 |
| Average | 2.9 | 71.1 | **78.1** |

(b) Extractive task results

| Task Name | 0-shot | 10-shot | Ours |
|---|---|---|---|
| Adjective_V_Verb_3 | 16.7 | 74.3 | 95.2 |
| Adjective_V_Verb_5 | 15.2 | 66.7 | 94.3 |
| Alphabetically_First_3 | 24.3 | 29.5 | 33.3 |
| Alphabetically_First_5 | 22.9 | 23.3 | 28.6 |
| Alphabetically_Last_3 | 21.9 | 30.5 | 37.1 |
| Alphabetically_Last_5 | 11.9 | 19.1 | 29.5 |
| Animal_V_Object_3 | 10.5 | 70.5 | 98.1 |
| Animal_V_Object_5 | 16.7 | 64.8 | 95.7 |
| Choose_First_Of_3 | 65.2 | 99.5 | 100.0 |
| Choose_First_Of_5 | 82.9 | 98.6 | 100.0 |
| Choose_Last_Of_3 | 4.8 | 96.7 | 99.5 |
| Choose_Last_Of_5 | 0.0 | 92.4 | 100.0 |
| Choose_Middle_Of_3 | 1.4 | 55.7 | 94.8 |
| Choose_Middle_Of_5 | 0.5 | 21.9 | 51.9 |
| Color_V_Animal_3 | 12.9 | 83.8 | 100.0 |
| Color_V_Animal_5 | 11.0 | 84.3 | 99.1 |
| Concept_V_Object_3 | 20.5 | 70.5 | 94.8 |
| Concept_V_Object_5 | 17.1 | 62.9 | 95.7 |
| Conll2003_Location | 9.3 | 82.1 | 91.3 |
| Conll2003_Organization | 12.6 | 75.6 | 88.4 |
| Conll2003_Person | 12.9 | 87.9 | 95.7 |
| Fruit_V_Animal_3 | 6.2 | 74.8 | 98.6 |
| Fruit_V_Animal_5 | 3.3 | 71.0 | 99.5 |
| Object_V_Concept_3 | 15.2 | 71.4 | 97.1 |
| Object_V_Concept_5 | 14.8 | 61.9 | 96.2 |
| Squad_Val | 53.1 | 85.8 | 86.0 |
| Verb_V_Adjective_3 | 11.4 | 67.1 | 98.1 |
| Verb_V_Adjective_5 | 5.2 | 71.0 | 98.6 |
| Average | 17.9 | 67.6 | **85.6** |

Table 14: **Task-wise performance on 57 ICL tasks using Gemma-3-4B-it.** Our method is evaluated along with 0-shot and 10-shot baselines. (a) Results on 29 abstractive tasks. (b) Results on 28 extractive tasks. The best results are shown in **bold**, and the second-best results are underlined.

(a) Abstractive task results

| Task Name | 0-shot | 10-shot | Ours |
|---|---|---|---|
| AG_News | 0.0 | 75.4 | 84.9 |
| Antonym | 0.0 | 66.3 | 70.2 |
| Capitalize | 0.0 | 99.4 | 99.4 |
| Capitalize_First_Letter | 4.7 | 96.5 | 100.0 |
| Capitalize_Last_Letter | 2.9 | 18.1 | 79.0 |
| Capitalize_Second_Letter | 7.3 | 18.8 | 93.9 |
| Commonsense_QA | 59.6 | 68.8 | 61.6 |
| Country-Capital | 0.0 | 90.5 | 90.5 |
| Country-Currency | 0.0 | 69.1 | 78.6 |
| English-French | 0.3 | 81.4 | 71.4 |
| English-German | 0.1 | 75.7 | 64.2 |
| English-Spanish | 0.0 | 84.2 | 81.2 |
| Landmark-Country | 0.0 | 76.6 | 74.9 |
| Lowercase_First_Letter | 0.0 | 99.4 | 100.0 |
| Lowercase_Last_Letter | 0.0 | 39.8 | 90.6 |
| National_Parks | 0.0 | 69.5 | 70.5 |
| Next_Capital_Letter | 2.3 | 4.7 | 65.5 |
| Next_Item | 2.1 | 97.9 | 95.7 |
| Park-Country | 0.0 | 76.4 | 72.6 |
| Person-Instrument | 0.0 | 48.6 | 57.0 |
| Person-Occupation | 0.0 | 39.5 | 62.8 |
| Person-Sport | 0.0 | 94.0 | 97.0 |
| Present-Past | 0.0 | 100.0 | 98.4 |
| Prev_Item | 2.1 | 74.5 | 87.2 |
| Product-Company | 2.8 | 72.5 | 66.1 |
| Sentiment | 0.0 | 91.0 | 94.3 |
| Singular-Plural | 0.0 | 100.0 | 97.7 |
| Synonym | 6.0 | 50.3 | 51.7 |
| Word_Length | 0.0 | 38.0 | 68.4 |
| Average | 3.1 | 69.5 | **80.2** |

(b) Extractive task results

| Task Name | 0-shot | 10-shot | Ours |
|---|---|---|---|
| Adjective_V_Verb_3 | 3.8 | 77.6 | 97.6 |
| Adjective_V_Verb_5 | 15.7 | 72.9 | 95.7 |
| Alphabetically_First_3 | 21.9 | 28.1 | 31.9 |
| Alphabetically_First_5 | 22.9 | 21.4 | 26.7 |
| Alphabetically_Last_3 | 12.9 | 39.1 | 39.5 |
| Alphabetically_Last_5 | 16.7 | 25.7 | 29.1 |
| Animal_V_Object_3 | 21.9 | 81.0 | 98.6 |
| Animal_V_Object_5 | 23.3 | 91.0 | 97.1 |
| Choose_First_Of_3 | 31.0 | 99.5 | 100.0 |
| Choose_First_Of_5 | 25.7 | 99.5 | 100.0 |
| Choose_Last_Of_3 | 10.0 | 99.1 | 100.0 |
| Choose_Last_Of_5 | 15.7 | 94.8 | 100.0 |
| Choose_Middle_Of_3 | 7.6 | 70.0 | 94.3 |
| Choose_Middle_Of_5 | 12.4 | 35.2 | 66.7 |
| Color_V_Animal_3 | 17.1 | 71.4 | 100.0 |
| Color_V_Animal_5 | 13.8 | 79.5 | 99.1 |
| Concept_V_Object_3 | 27.6 | 61.9 | 97.6 |
| Concept_V_Object_5 | 34.3 | 75.2 | 93.3 |
| Conll2003_Location | 7.7 | 86.4 | 92.4 |
| Conll2003_Organization | 19.7 | 77.8 | 88.2 |
| Conll2003_Person | 23.1 | 93.4 | 96.4 |
| Fruit_V_Animal_3 | 11.4 | 74.8 | 97.6 |
| Fruit_V_Animal_5 | 4.8 | 82.9 | 100.0 |
| Object_V_Concept_3 | 3.3 | 77.6 | 97.6 |
| Object_V_Concept_5 | 3.3 | 67.6 | 96.2 |
| Squad_Val | 71.7 | 87.9 | 86.4 |
| Verb_V_Adjective_3 | 3.8 | 69.5 | 95.7 |
| Verb_V_Adjective_5 | 6.7 | 71.4 | 99.1 |
| Average | 17.5 | 71.9 | **86.3** |

Table 15: **Task-wise performance on 57 ICL tasks using Mistral-7B-v0.3.** Our method is evaluated along with 0-shot and 10-shot baselines. (a) Results on 29 abstractive tasks. (b) Results on 28 extractive tasks. The best results are shown in **bold**, and the second-best results are underlined.

(a) Abstractive task results

| Task Name | 0-shot | 10-shot | Ours |
|---|---|---|---|
| AG_News | 0.4 | 81.0 | 88.9 |
| Antonym | 7.9 | 68.3 | 67.9 |
| Capitalize | 6.5 | 100.0 | 100.0 |
| Capitalize_First_Letter | 6.5 | 90.0 | 99.4 |
| Capitalize_Last_Letter | 0.6 | 33.9 | 86.6 |
| Capitalize_Second_Letter | 0.6 | 27.9 | 96.4 |
| Commonsense_QA | 21.1 | 70.8 | 59.0 |
| Country-Capital | 4.8 | 90.5 | 88.1 |
| Country-Currency | 0.0 | 78.6 | 78.6 |
| English-French | 0.3 | 79.8 | 77.7 |
| English-German | 1.4 | 74.0 | 63.2 |
| English-Spanish | 0.3 | 84.6 | 79.6 |
| Landmark-Country | 0.0 | 85.1 | 82.9 |
| Lowercase_First_Letter | 0.0 | 83.0 | 100.0 |
| Lowercase_Last_Letter | 0.0 | 49.1 | 95.3 |
| National_Parks | 1.1 | 79.0 | 77.9 |
| Next_Capital_Letter | 0.6 | 5.3 | 98.8 |
| Next_Item | 0.0 | 97.9 | 97.9 |
| Park-Country | 0.0 | 87.3 | 79.6 |
| Person-Instrument | 0.0 | 75.7 | 76.6 |
| Person-Occupation | 0.0 | 59.9 | 70.0 |
| Person-Sport | 0.0 | 92.5 | 97.0 |
| Present-Past | 1.6 | 98.4 | 100.0 |
| Prev_Item | 0.0 | 91.5 | 95.7 |
| Product-Company | 0.9 | 82.6 | 80.7 |
| Sentiment | 0.0 | 94.7 | 93.9 |
| Singular-Plural | 2.3 | 97.7 | 97.7 |
| Synonym | 1.7 | 51.2 | 47.9 |
| Word_Length | 0.0 | 31.6 | 63.7 |
| Average | 2.0 | 73.8 | **84.2** |

(b) Extractive task results

| Task Name | 0-shot | 10-shot | Ours |
|---|---|---|---|
| Adjective_V_Verb_3 | 34.8 | 76.7 | 98.1 |
| Adjective_V_Verb_5 | 16.7 | 79.1 | 97.1 |
| Alphabetically_First_3 | 31.0 | 32.9 | 53.8 |
| Alphabetically_First_5 | 20.5 | 22.9 | 86.2 |
| Alphabetically_Last_3 | 24.8 | 27.1 | 45.7 |
| Alphabetically_Last_5 | 11.0 | 18.6 | 51.4 |
| Animal_V_Object_3 | 24.3 | 71.9 | 96.7 |
| Animal_V_Object_5 | 24.3 | 88.1 | 99.1 |
| Choose_First_Of_3 | 81.4 | 100.0 | 100.0 |
| Choose_First_Of_5 | 73.8 | 100.0 | 99.1 |
| Choose_Last_Of_3 | 2.9 | 99.5 | 100.0 |
| Choose_Last_Of_5 | 1.9 | 97.1 | 100.0 |
| Choose_Middle_Of_3 | 5.7 | 42.9 | 98.6 |
| Choose_Middle_Of_5 | 0.5 | 33.3 | 70.5 |
| Color_V_Animal_3 | 28.6 | 84.3 | 99.1 |
| Color_V_Animal_5 | 17.6 | 85.2 | 99.1 |
| Concept_V_Object_3 | 19.1 | 77.6 | 99.1 |
| Concept_V_Object_5 | 17.1 | 88.1 | 97.1 |
| Conll2003_Location | 9.7 | 87.2 | 94.5 |
| Conll2003_Organization | 9.3 | 77.1 | 92.0 |
| Conll2003_Person | 9.7 | 92.1 | 97.6 |
| Fruit_V_Animal_3 | 29.1 | 87.1 | 98.6 |
| Fruit_V_Animal_5 | 13.3 | 93.3 | 98.6 |
| Object_V_Concept_3 | 27.6 | 81.4 | 98.6 |
| Object_V_Concept_5 | 14.3 | 81.0 | 97.6 |
| Squad_Val | 58.4 | 84.9 | 88.9 |
| Verb_V_Adjective_3 | 24.3 | 67.6 | 97.1 |
| Verb_V_Adjective_5 | 9.1 | 80.0 | 98.1 |
| Average | 22.9 | 73.5 | **91.1** |

Table 16: **Task-wise performance on 57 ICL tasks using Mistral-7B-Instruct-v0.3.** Our method is evaluated along with 0-shot and 10-shot baselines. (a) Results on 29 abstractive tasks. (b) Results on 28 extractive tasks. The best results are shown in **bold**, and the second-best results are underlined.

(a) Abstractive task results

| Task Name | 0-shot | 10-shot | Ours |
|---|---|---|---|
| AG_News | 0.0 | 79.5 | 88.4 |
| Antonym | 1.2 | 69.8 | 70.0 |
| Capitalize | 31.8 | 99.4 | 100.0 |
| Capitalize_First_Letter | 30.6 | 98.2 | 99.4 |
| Capitalize_Last_Letter | 0.6 | 30.4 | 90.6 |
| Capitalize_Second_Letter | 1.8 | 26.1 | 95.8 |
| Commonsense_QA | 24.0 | 71.8 | 66.1 |
| Country-Capital | 4.8 | 88.1 | 88.1 |
| Country-Currency | 0.0 | 78.6 | 71.4 |
| English-French | 0.4 | 82.4 | 79.3 |
| English-German | 1.5 | 75.5 | 60.5 |
| English-Spanish | 0.3 | 85.4 | 80.0 |
| Landmark-Country | 0.0 | 84.6 | 80.6 |
| Lowercase_First_Letter | 0.0 | 97.7 | 100.0 |
| Lowercase_Last_Letter | 0.0 | 42.7 | 95.3 |
| National_Parks | 1.1 | 80.0 | 75.8 |
| Next_Capital_Letter | 0.0 | 4.1 | 97.7 |
| Next_Item | 0.0 | 97.9 | 95.7 |
| Park-Country | 0.0 | 84.7 | 80.9 |
| Person-Instrument | 1.9 | 71.0 | 75.7 |
| Person-Occupation | 0.6 | 54.7 | 66.3 |
| Person-Sport | 0.0 | 92.5 | 94.0 |
| Present-Past | 3.3 | 98.4 | 100.0 |
| Prev_Item | 4.3 | 89.4 | 97.9 |
| Product-Company | 0.0 | 79.8 | 81.7 |
| Sentiment | 0.0 | 94.3 | 94.3 |
| Singular-Plural | 7.0 | 100.0 | 97.7 |
| Synonym | 1.2 | 53.2 | 49.5 |
| Word_Length | 0.6 | 53.2 | 62.6 |
| Average | 4.0 | 74.6 | **84.0** |

(b) Extractive task results

| Task Name | 0-shot | 10-shot | Ours |
|---|---|---|---|
| Adjective_V_Verb_3 | 14.8 | 78.6 | 99.1 |
| Adjective_V_Verb_5 | 1.0 | 83.8 | 98.1 |
| Alphabetically_First_3 | 16.7 | 31.9 | 45.2 |
| Alphabetically_First_5 | 4.8 | 24.8 | 88.6 |
| Alphabetically_Last_3 | 11.4 | 31.4 | 48.1 |
| Alphabetically_Last_5 | 6.7 | 22.4 | 42.4 |
| Animal_V_Object_3 | 10.5 | 84.3 | 97.1 |
| Animal_V_Object_5 | 4.3 | 94.8 | 98.1 |
| Choose_First_Of_3 | 41.4 | 99.5 | 99.1 |
| Choose_First_Of_5 | 8.6 | 96.2 | 99.1 |
| Choose_Last_Of_3 | 5.2 | 88.1 | 100.0 |
| Choose_Last_Of_5 | 1.0 | 87.1 | 100.0 |
| Choose_Middle_Of_3 | 5.7 | 45.2 | 96.2 |
| Choose_Middle_Of_5 | 1.9 | 41.0 | 96.7 |
| Color_V_Animal_3 | 14.3 | 82.4 | 100.0 |
| Color_V_Animal_5 | 1.9 | 91.4 | 99.5 |
| Concept_V_Object_3 | 8.1 | 88.1 | 97.6 |
| Concept_V_Object_5 | 5.2 | 90.5 | 99.1 |
| Conll2003_Location | 5.6 | 84.0 | 95.2 |
| Conll2003_Organization | 9.7 | 79.6 | 92.8 |
| Conll2003_Person | 22.3 | 89.9 | 97.7 |
| Fruit_V_Animal_3 | 13.8 | 94.3 | 99.1 |
| Fruit_V_Animal_5 | 1.0 | 97.6 | 100.0 |
| Object_V_Concept_3 | 15.2 | 78.6 | 98.1 |
| Object_V_Concept_5 | 3.3 | 84.3 | 98.1 |
| Squad_Val | 60.4 | 87.5 | 88.1 |
| Verb_V_Adjective_3 | 10.0 | 68.1 | 96.7 |
| Verb_V_Adjective_5 | 2.9 | 79.5 | 97.6 |
| Average | 11.0 | 75.2 | **91.7** |

Table 17: **Task-wise performance on 57 ICL tasks using Llama-3.1-8B-Instruct.** Our method is evaluated along with 0-shot and 10-shot baselines. (a) Results on 29 abstractive tasks. (b) Results on 28 extractive tasks. The best results are shown in **bold**, and the second-best results are underlined.

(a) Abstractive task results

| Task Name | 0-shot | 10-shot | Ours |
|---|---|---|---|
| AG_News | 0.0 | 77.6 | 90.0 |
| Antonym | 0.4 | 70.8 | 71.6 |
| Capitalize | 0.6 | 99.4 | 99.4 |
| Capitalize_First_Letter | 2.4 | 100.0 | 100.0 |
| Capitalize_Last_Letter | 2.3 | 49.7 | 95.3 |
| Capitalize_Second_Letter | 1.8 | 49.7 | 100.0 |
| Commonsense_QA | 71.3 | 74.0 | 72.0 |
| Country-Capital | 2.4 | 90.5 | 90.5 |
| Country-Currency | 0.0 | 81.0 | 85.7 |
| English-French | 0.7 | 83.1 | 82.0 |
| English-German | 0.7 | 76.7 | 70.1 |
| English-Spanish | 0.2 | 84.8 | 84.3 |
| Landmark-Country | 0.0 | 88.0 | 82.9 |
| Lowercase_First_Letter | 0.0 | 100.0 | 99.4 |
| Lowercase_Last_Letter | 0.0 | 60.2 | 97.1 |
| National_Parks | 1.1 | 86.3 | 75.8 |
| Next_Capital_Letter | 1.2 | 2.9 | 99.4 |
| Next_Item | 0.0 | 97.9 | 97.9 |
| Park-Country | 1.3 | 89.2 | 84.1 |
| Person-Instrument | 0.0 | 82.2 | 88.8 |
| Person-Occupation | 0.0 | 65.1 | 77.3 |
| Person-Sport | 0.0 | 95.5 | 97.0 |
| Present-Past | 1.6 | 100.0 | 100.0 |
| Prev_Item | 4.3 | 91.5 | 95.7 |
| Product-Company | 1.8 | 84.4 | 84.4 |
| Sentiment | 0.0 | 94.7 | 97.1 |
| Singular-Plural | 4.7 | 100.0 | 95.4 |
| Synonym | 32.1 | 53.3 | 55.1 |
| Word_Length | 0.0 | 74.3 | 83.6 |
| Average | 4.5 | 79.4 | **88.0** |

(b) Extractive task results

| Task Name | 0-shot | 10-shot | Ours |
|---|---|---|---|
| Adjective_V_Verb_3 | 14.3 | 87.6 | 100.0 |
| Adjective_V_Verb_5 | 13.3 | 91.0 | 97.1 |
| Alphabetically_First_3 | 26.7 | 37.6 | 51.4 |
| Alphabetically_First_5 | 18.6 | 22.4 | 92.4 |
| Alphabetically_Last_3 | 16.2 | 37.6 | 49.1 |
| Alphabetically_Last_5 | 11.0 | 28.6 | 74.3 |
| Animal_V_Object_3 | 21.9 | 95.7 | 99.5 |
| Animal_V_Object_5 | 6.7 | 96.7 | 100.0 |
| Choose_First_Of_3 | 19.5 | 97.6 | 100.0 |
| Choose_First_Of_5 | 9.1 | 97.1 | 100.0 |
| Choose_Last_Of_3 | 29.5 | 93.3 | 100.0 |
| Choose_Last_Of_5 | 26.2 | 94.3 | 100.0 |
| Choose_Middle_Of_3 | 15.2 | 53.3 | 99.1 |
| Choose_Middle_Of_5 | 9.5 | 33.8 | 96.7 |
| Color_V_Animal_3 | 29.5 | 99.5 | 100.0 |
| Color_V_Animal_5 | 8.6 | 97.6 | 100.0 |
| Concept_V_Object_3 | 26.2 | 91.9 | 99.5 |
| Concept_V_Object_5 | 17.1 | 95.2 | 97.1 |
| Conll2003_Location | 10.2 | 90.1 | 94.7 |
| Conll2003_Organization | 33.6 | 80.6 | 93.7 |
| Conll2003_Person | 22.9 | 93.4 | 97.5 |
| Fruit_V_Animal_3 | 4.8 | 99.5 | 99.1 |
| Fruit_V_Animal_5 | 0.5 | 98.1 | 99.5 |
| Object_V_Concept_3 | 26.7 | 94.8 | 98.6 |
| Object_V_Concept_5 | 17.1 | 94.8 | 99.5 |
| Squad_Val | 76.6 | 87.8 | 90.1 |
| Verb_V_Adjective_3 | 10.5 | 96.7 | 98.6 |
| Verb_V_Adjective_5 | 8.6 | 97.1 | 99.1 |
| Average | 18.9 | 81.6 | **93.8** |

Table 18: **Task-wise performance on 57 ICL tasks using Qwen3-8B.** Our method is evaluated along with 0-shot and 10-shot baselines. (a) Results on 29 abstractive tasks. (b) Results on 28 extractive tasks. The best results are shown in **bold**, and the second-best results are underlined.

(a) Abstractive task results

| Task Name | 0-shot | 10-shot | Ours |
|---|---|---|---|
| AG_News | 0.0 | 77.8 | 87.7 |
| Antonym | 0.0 | 69.4 | 66.5 |
| Capitalize | 19.4 | 99.4 | 98.8 |
| Capitalize_First_Letter | 16.5 | 97.7 | 100.0 |
| Capitalize_Last_Letter | 4.1 | 24.6 | 93.0 |
| Capitalize_Second_Letter | 11.5 | 30.3 | 97.6 |
| Commonsense_QA | 42.2 | 80.8 | 79.8 |
| Country-Capital | 4.8 | 88.1 | 88.1 |
| Country-Currency | 0.0 | 81.0 | 64.3 |
| English-French | 0.3 | 82.3 | 69.4 |
| English-German | 0.7 | 73.3 | 57.5 |
| English-Spanish | 0.2 | 84.4 | 73.9 |
| Landmark-Country | 0.0 | 81.1 | 77.7 |
| Lowercase_First_Letter | 0.0 | 98.8 | 100.0 |
| Lowercase_Last_Letter | 0.0 | 60.2 | 97.7 |
| National_Parks | 0.0 | 79.0 | 70.5 |
| Next_Capital_Letter | 0.6 | 8.8 | 92.4 |
| Next_Item | 0.0 | 95.7 | 95.7 |
| Park-Country | 0.0 | 80.9 | 70.1 |
| Person-Instrument | 0.0 | 62.6 | 62.6 |
| Person-Occupation | 0.0 | 46.5 | 60.5 |
| Person-Sport | 0.0 | 89.6 | 97.0 |
| Present-Past | 1.6 | 100.0 | 98.4 |
| Prev_Item | 4.3 | 97.9 | 93.6 |
| Product-Company | 0.0 | 77.1 | 80.7 |
| Sentiment | 0.0 | 95.1 | 95.9 |
| Singular-Plural | 4.7 | 97.7 | 90.7 |
| Synonym | 0.2 | 49.0 | 48.3 |
| Word_Length | 0.0 | 67.8 | 73.7 |
| Average | 3.8 | 75.1 | **82.1** |

(b) Extractive task results

| Task Name | 0-shot | 10-shot | Ours |
|---|---|---|---|
| Adjective_V_Verb_3 | 0.5 | 77.6 | 98.1 |
| Adjective_V_Verb_5 | 0.0 | 87.1 | 98.6 |
| Alphabetically_First_3 | 5.2 | 31.0 | 42.4 |
| Alphabetically_First_5 | 2.4 | 20.5 | 89.5 |
| Alphabetically_Last_3 | 2.9 | 38.1 | 42.4 |
| Alphabetically_Last_5 | 2.4 | 20.0 | 80.0 |
| Animal_V_Object_3 | 1.4 | 96.7 | 95.7 |
| Animal_V_Object_5 | 0.5 | 98.1 | 96.7 |
| Choose_First_Of_3 | 5.7 | 95.7 | 100.0 |
| Choose_First_Of_5 | 1.0 | 94.8 | 100.0 |
| Choose_Last_Of_3 | 2.4 | 91.4 | 100.0 |
| Choose_Last_Of_5 | 1.4 | 66.2 | 100.0 |
| Choose_Middle_Of_3 | 2.9 | 62.9 | 99.5 |
| Choose_Middle_Of_5 | 0.5 | 31.4 | 98.1 |
| Color_V_Animal_3 | 2.9 | 99.5 | 100.0 |
| Color_V_Animal_5 | 4.3 | 98.6 | 100.0 |
| Concept_V_Object_3 | 1.0 | 91.9 | 100.0 |
| Concept_V_Object_5 | 0.0 | 90.5 | 95.7 |
| Conll2003_Location | 7.7 | 91.0 | 95.1 |
| Conll2003_Organization | 15.1 | 83.6 | 90.7 |
| Conll2003_Person | 26.8 | 94.0 | 96.6 |
| Fruit_V_Animal_3 | 6.7 | 100.0 | 100.0 |
| Fruit_V_Animal_5 | 5.2 | 99.5 | 99.5 |
| Object_V_Concept_3 | 0.0 | 91.4 | 97.1 |
| Object_V_Concept_5 | 0.0 | 93.8 | 96.7 |
| Squad_Val | 38.1 | 90.7 | 88.3 |
| Verb_V_Adjective_3 | 1.9 | 91.9 | 99.5 |
| Verb_V_Adjective_5 | 0.0 | 97.1 | 96.2 |
| Average | 5.0 | 79.5 | **92.7** |

Table 19: **Task-wise performance on 57 ICL tasks using Qwen3-32B.** Our method is evaluated along with 0-shot and 10-shot baselines. (a) Results on 29 abstractive tasks. (b) Results on 28 extractive tasks. The best results are shown in **bold**, and the second-best results are underlined.

(a) Abstractive task results

| Task Name | 0-shot | 10-shot | Ours |
|---|---|---|---|
| AG_News | 0.1 | 81.8 | 88.2 |
| Antonym | 8.5 | 67.1 | 65.7 |
| Capitalize | 4.7 | 93.5 | 99.4 |
| Capitalize_First_Letter | 6.5 | 97.7 | 100.0 |
| Capitalize_Last_Letter | 7.0 | 24.0 | 95.3 |
| Capitalize_Second_Letter | 3.0 | 29.1 | 97.0 |
| Commonsense_QA | 38.9 | 86.2 | 84.5 |
| Country-Capital | 7.1 | 76.2 | 90.5 |
| Country-Currency | 0.0 | 81.0 | 81.0 |
| English-French | 0.4 | 80.2 | 77.2 |
| English-German | 0.5 | 76.1 | 66.2 |
| English-Spanish | 0.0 | 82.9 | 80.1 |
| Landmark-Country | 0.0 | 84.0 | 81.1 |
| Lowercase_First_Letter | 0.6 | 89.5 | 99.4 |
| Lowercase_Last_Letter | 0.6 | 39.8 | 95.3 |
| National_Parks | 0.0 | 83.2 | 79.0 |
| Next_Capital_Letter | 2.3 | 2.9 | 98.3 |
| Next_Item | 0.0 | 85.1 | 95.7 |
| Park-Country | 0.0 | 84.1 | 77.7 |
| Person-Instrument | 0.0 | 56.1 | 71.0 |
| Person-Occupation | 0.0 | 36.6 | 69.2 |
| Person-Sport | 0.0 | 88.1 | 95.5 |
| Present-Past | 0.0 | 83.6 | 98.4 |
| Prev_Item | 0.0 | 78.7 | 97.9 |
| Product-Company | 0.0 | 83.5 | 84.4 |
| Sentiment | 2.0 | 93.9 | 92.2 |
| Singular-Plural | 2.3 | 97.7 | 95.4 |
| Synonym | 1.2 | 47.5 | 50.2 |
| Word_Length | 0.0 | 77.8 | 73.1 |
| Average | 3.0 | 72.0 | **85.5** |

(b) Extractive task results

| Task Name | 0-shot | 10-shot | Ours |
|---|---|---|---|
| Adjective_V_Verb_3 | 6.2 | 78.1 | 99.5 |
| Adjective_V_Verb_5 | 4.8 | 81.0 | 98.1 |
| Alphabetically_First_3 | 7.1 | 38.1 | 97.6 |
| Alphabetically_First_5 | 3.3 | 22.9 | 92.9 |
| Alphabetically_Last_3 | 5.2 | 36.2 | 43.3 |
| Alphabetically_Last_5 | 2.4 | 17.6 | 71.0 |
| Animal_V_Object_3 | 11.0 | 97.6 | 99.1 |
| Animal_V_Object_5 | 11.0 | 98.6 | 98.6 |
| Choose_First_Of_3 | 14.8 | 95.7 | 100.0 |
| Choose_First_Of_5 | 6.7 | 95.2 | 100.0 |
| Choose_Last_Of_3 | 1.4 | 92.9 | 100.0 |
| Choose_Last_Of_5 | 2.9 | 85.7 | 100.0 |
| Choose_Middle_Of_3 | 1.4 | 56.2 | 100.0 |
| Choose_Middle_Of_5 | 0.0 | 32.9 | 98.1 |
| Color_V_Animal_3 | 4.8 | 96.2 | 100.0 |
| Color_V_Animal_5 | 8.6 | 97.6 | 99.5 |
| Concept_V_Object_3 | 1.9 | 88.1 | 99.1 |
| Concept_V_Object_5 | 1.0 | 97.1 | 98.1 |
| Conll2003_Location | 2.1 | 89.4 | 95.7 |
| Conll2003_Organization | 8.7 | 83.8 | 92.3 |
| Conll2003_Person | 15.1 | 93.8 | 96.8 |
| Fruit_V_Animal_3 | 5.2 | 100.0 | 100.0 |
| Fruit_V_Animal_5 | 6.2 | 100.0 | 99.5 |
| Object_V_Concept_3 | 4.3 | 96.2 | 99.5 |
| Object_V_Concept_5 | 2.9 | 94.3 | 98.6 |
| Squad_Val | 31.6 | 90.8 | 90.5 |
| Verb_V_Adjective_3 | 0.5 | 91.4 | 100.0 |
| Verb_V_Adjective_5 | 1.4 | 98.6 | 99.5 |
| Average | 6.2 | 80.2 | **95.3** |

Table 20: **Task-wise performance on 57 ICL tasks using Mixtral-8x7B-v0.1.** Our method is evaluated along with 0-shot and 10-shot baselines. (a) Results on 29 abstractive tasks. (b) Results on 28 extractive tasks. The best results are shown in **bold**, and the second-best results are underlined.

(a) Abstractive task results

| Task Name | 0-shot | 10-shot | Ours |
|---|---|---|---|
| AG_News | 0.3 | 81.5 | 89.9 |
| Antonym | 3.8 | 70.4 | 67.3 |
| Capitalize | 8.8 | 99.4 | 100.0 |
| Capitalize_First_Letter | 8.2 | 97.7 | 100.0 |
| Capitalize_Last_Letter | 0.6 | 39.2 | 91.8 |
| Capitalize_Second_Letter | 0.0 | 32.1 | 95.8 |
| Commonsense_QA | 39.5 | 73.9 | 61.8 |
| Country-Capital | 4.8 | 90.5 | 85.7 |
| Country-Currency | 0.0 | 83.3 | 83.3 |
| English-French | 0.2 | 84.3 | 82.0 |
| English-German | 0.8 | 78.2 | 74.4 |
| English-Spanish | 0.2 | 86.4 | 87.6 |
| Landmark-Country | 0.0 | 90.3 | 85.1 |
| Lowercase_First_Letter | 0.0 | 93.6 | 100.0 |
| Lowercase_Last_Letter | 0.0 | 46.2 | 94.7 |
| National_Parks | 1.1 | 85.3 | 79.0 |
| Next_Capital_Letter | 1.8 | 5.3 | 98.3 |
| Next_Item | 0.0 | 97.9 | 97.9 |
| Park-Country | 0.0 | 91.7 | 87.9 |
| Person-Instrument | 0.0 | 84.1 | 87.9 |
| Person-Occupation | 0.0 | 77.9 | 82.6 |
| Person-Sport | 0.0 | 95.5 | 98.5 |
| Present-Past | 1.6 | 100.0 | 100.0 |
| Prev_Item | 2.1 | 97.9 | 97.9 |
| Product-Company | 0.0 | 89.9 | 88.1 |
| Sentiment | 0.0 | 96.3 | 95.1 |
| Singular-Plural | 2.3 | 100.0 | 100.0 |
| Synonym | 0.7 | 54.6 | 49.5 |
| Word_Length | 0.0 | 75.4 | 74.3 |
| Average | 2.6 | 79.3 | **87.5** |

(b) Extractive task results

| Task Name | 0-shot | 10-shot | Ours |
|---|---|---|---|
| Adjective_V_Verb_3 | 26.2 | 84.3 | 99.5 |
| Adjective_V_Verb_5 | 18.1 | 83.8 | 97.6 |
| Alphabetically_First_3 | 28.1 | 37.1 | 46.2 |
| Alphabetically_First_5 | 20.5 | 22.9 | 89.1 |
| Alphabetically_Last_3 | 16.2 | 39.1 | 51.0 |
| Alphabetically_Last_5 | 13.8 | 20.5 | 46.2 |
| Animal_V_Object_3 | 21.4 | 94.3 | 97.1 |
| Animal_V_Object_5 | 24.8 | 91.9 | 98.1 |
| Choose_First_Of_3 | 65.7 | 99.1 | 100.0 |
| Choose_First_Of_5 | 73.3 | 99.1 | 100.0 |
| Choose_Last_Of_3 | 2.9 | 99.5 | 100.0 |
| Choose_Last_Of_5 | 2.4 | 95.7 | 99.5 |
| Choose_Middle_Of_3 | 4.3 | 50.5 | 98.1 |
| Choose_Middle_Of_5 | 1.4 | 28.1 | 89.5 |
| Color_V_Animal_3 | 22.9 | 97.6 | 100.0 |
| Color_V_Animal_5 | 9.1 | 97.1 | 99.5 |
| Concept_V_Object_3 | 18.6 | 76.2 | 99.1 |
| Concept_V_Object_5 | 13.8 | 86.7 | 95.7 |
| Conll2003_Location | 6.8 | 88.6 | 93.7 |
| Conll2003_Organization | 12.3 | 77.2 | 92.3 |
| Conll2003_Person | 8.1 | 93.8 | 97.9 |
| Fruit_V_Animal_3 | 31.0 | 97.1 | 99.1 |
| Fruit_V_Animal_5 | 6.7 | 97.1 | 99.5 |
| Object_V_Concept_3 | 22.9 | 91.4 | 99.1 |
| Object_V_Concept_5 | 12.9 | 86.7 | 97.1 |
| Squad_Val | 58.9 | 86.2 | 87.5 |
| Verb_V_Adjective_3 | 8.1 | 70.5 | 96.7 |
| Verb_V_Adjective_5 | 4.3 | 90.0 | 98.6 |
| Average | 19.8 | 77.9 | **91.7** |

Table 21: **Task-wise performance on 57 ICL tasks using Mixtral-8x7B-Instruct-v0.1.** Our method is evaluated along with 0-shot and 10-shot baselines. (a) Results on 29 abstractive tasks. (b) Results on 28 extractive tasks. The best results are shown in **bold**, and the second-best results are underlined.

(a) Abstractive task results

| Task Name | 0-shot | 10-shot | Ours |
|---|---|---|---|
| AG_News | 0.0 | 81.3 | 89.2 |
| Antonym | 0.0 | 72.4 | 67.7 |
| Capitalize | 4.7 | 99.4 | 100.0 |
| Capitalize_First_Letter | 8.2 | 99.4 | 100.0 |
| Capitalize_Last_Letter | 0.6 | 25.7 | 87.1 |
| Capitalize_Second_Letter | 0.6 | 27.9 | 92.1 |
| Commonsense_QA | 56.8 | 73.9 | 67.5 |
| Country-Capital | 4.8 | 90.5 | 83.3 |
| Country-Currency | 0.0 | 73.8 | 81.0 |
| English-French | 0.0 | 84.7 | 82.0 |
| English-German | 0.2 | 77.2 | 71.5 |
| English-Spanish | 0.1 | 85.8 | 85.4 |
| Landmark-Country | 0.0 | 92.0 | 82.3 |
| Lowercase_First_Letter | 0.0 | 97.1 | 100.0 |
| Lowercase_Last_Letter | 0.0 | 48.0 | 97.7 |
| National_Parks | 2.1 | 80.0 | 76.8 |
| Next_Capital_Letter | 0.6 | 5.3 | 99.4 |
| Next_Item | 0.0 | 97.9 | 95.7 |
| Park-Country | 0.0 | 90.5 | 87.9 |
| Person-Instrument | 0.0 | 84.1 | 88.8 |
| Person-Occupation | 0.0 | 74.4 | 82.6 |
| Person-Sport | 0.0 | 95.5 | 98.5 |
| Present-Past | 0.0 | 100.0 | 100.0 |
| Prev_Item | 0.0 | 93.6 | 95.7 |
| Product-Company | 0.0 | 91.7 | 88.1 |
| Sentiment | 0.0 | 94.7 | 93.9 |
| Singular-Plural | 0.0 | 100.0 | 100.0 |
| Synonym | 0.2 | 51.7 | 47.9 |
| Word_Length | 0.0 | 58.5 | 65.5 |
| Average | 2.7 | 77.5 | **86.5** |

(b) Extractive task results

| Task Name | 0-shot | 10-shot | Ours |
|---|---|---|---|
| Adjective_V_Verb_3 | 30.0 | 87.6 | 97.1 |
| Adjective_V_Verb_5 | 17.6 | 89.5 | 98.1 |
| Alphabetically_First_3 | 21.9 | 39.1 | 39.1 |
| Alphabetically_First_5 | 14.3 | 24.8 | 77.6 |
| Alphabetically_Last_3 | 14.8 | 30.5 | 48.6 |
| Alphabetically_Last_5 | 8.6 | 24.3 | 42.9 |
| Animal_V_Object_3 | 12.9 | 93.3 | 97.6 |
| Animal_V_Object_5 | 8.1 | 95.7 | 98.6 |
| Choose_First_Of_3 | 59.5 | 98.1 | 98.6 |
| Choose_First_Of_5 | 35.7 | 94.8 | 99.1 |
| Choose_Last_Of_3 | 5.7 | 97.1 | 100.0 |
| Choose_Last_Of_5 | 6.7 | 95.7 | 100.0 |
| Choose_Middle_Of_3 | 1.9 | 53.3 | 97.1 |
| Choose_Middle_Of_5 | 2.4 | 28.1 | 79.1 |
| Color_V_Animal_3 | 20.0 | 98.6 | 100.0 |
| Color_V_Animal_5 | 1.0 | 96.7 | 99.1 |
| Concept_V_Object_3 | 16.7 | 86.2 | 99.1 |
| Concept_V_Object_5 | 7.1 | 92.9 | 96.7 |
| Conll2003_Location | 6.5 | 89.4 | 94.1 |
| Conll2003_Organization | 7.6 | 80.6 | 93.9 |
| Conll2003_Person | 20.8 | 92.6 | 96.6 |
| Fruit_V_Animal_3 | 17.1 | 97.1 | 97.1 |
| Fruit_V_Animal_5 | 1.0 | 97.6 | 98.6 |
| Object_V_Concept_3 | 13.8 | 92.4 | 97.1 |
| Object_V_Concept_5 | 6.7 | 87.6 | 96.2 |
| Squad_Val | 59.3 | 84.8 | 87.3 |
| Verb_V_Adjective_3 | 14.8 | 71.4 | 97.6 |
| Verb_V_Adjective_5 | 2.4 | 85.7 | 100.0 |
| Average | 15.5 | 78.8 | **90.2** |

Table 22: **Task-wise performance on 57 ICL tasks using Llama-3.1-70B.** Our method is evaluated along with 0-shot and 10-shot baselines. (a) Results on 29 abstractive tasks. (b) Results on 28 extractive tasks. The best results are shown in **bold**, and the second-best results are underlined.

(a) Abstractive task results

| Task Name | 0-shot | 10-shot | Ours |
|---|---|---|---|
| AG_News | 0.4 | 84.3 | 91.0 |
| Antonym | 16.7 | 71.6 | 71.4 |
| Capitalize | 0.0 | 99.4 | 100.0 |
| Capitalize_First_Letter | 0.6 | 100.0 | 100.0 |
| Capitalize_Last_Letter | 0.0 | 35.1 | 97.7 |
| Capitalize_Second_Letter | 0.0 | 37.6 | 98.2 |
| Commonsense_QA | 31.1 | 78.7 | 73.9 |
| Country-Capital | 4.8 | 92.9 | 92.9 |
| Country-Currency | 0.0 | 78.6 | 83.3 |
| English-French | 0.3 | 85.5 | 85.6 |
| English-German | 1.0 | 81.5 | 80.0 |
| English-Spanish | 0.3 | 89.8 | 89.2 |
| Landmark-Country | 0.0 | 89.1 | 84.0 |
| Lowercase_First_Letter | 0.0 | 98.8 | 100.0 |
| Lowercase_Last_Letter | 0.0 | 42.7 | 99.4 |
| National_Parks | 20.0 | 81.1 | 75.8 |
| Next_Capital_Letter | 0.6 | 9.4 | 100.0 |
| Next_Item | 4.3 | 95.7 | 95.7 |
| Park-Country | 48.4 | 91.7 | 86.0 |
| Person-Instrument | 0.0 | 79.4 | 83.2 |
| Person-Occupation | 0.0 | 66.9 | 83.7 |
| Person-Sport | 0.0 | 97.0 | 98.5 |
| Present-Past | 1.6 | 100.0 | 100.0 |
| Prev_Item | 2.1 | 97.9 | 97.9 |
| Product-Company | 1.8 | 90.8 | 88.1 |
| Sentiment | 0.0 | 98.0 | 96.3 |
| Singular-Plural | 2.3 | 100.0 | 97.7 |
| Synonym | 2.7 | 55.6 | 60.3 |
| Word_Length | 0.0 | 77.2 | 87.1 |
| Average | 4.8 | 79.5 | **89.5** |

(b) Extractive task results

| Task Name | 0-shot | 10-shot | Ours |
|---|---|---|---|
| Adjective_V_Verb_3 | 29.1 | 89.1 | 100.0 |
| Adjective_V_Verb_5 | 17.1 | 86.7 | 100.0 |
| Alphabetically_First_3 | 30.0 | 37.1 | 98.6 |
| Alphabetically_First_5 | 22.4 | 30.5 | 96.7 |
| Alphabetically_Last_3 | 23.3 | 34.8 | 62.9 |
| Alphabetically_Last_5 | 12.9 | 23.8 | 93.8 |
| Animal_V_Object_3 | 22.4 | 98.6 | 97.6 |
| Animal_V_Object_5 | 21.9 | 97.1 | 99.1 |
| Choose_First_Of_3 | 81.9 | 100.0 | 100.0 |
| Choose_First_Of_5 | 88.6 | 100.0 | 100.0 |
| Choose_Last_Of_3 | 0.0 | 94.8 | 100.0 |
| Choose_Last_Of_5 | 0.0 | 99.1 | 100.0 |
| Choose_Middle_Of_3 | 0.5 | 68.1 | 98.6 |
| Choose_Middle_Of_5 | 0.0 | 36.2 | 98.1 |
| Color_V_Animal_3 | 26.7 | 99.5 | 100.0 |
| Color_V_Animal_5 | 13.3 | 98.6 | 100.0 |
| Concept_V_Object_3 | 20.5 | 93.3 | 99.1 |
| Concept_V_Object_5 | 18.1 | 91.9 | 97.6 |
| Conll2003_Location | 20.6 | 92.3 | 96.8 |
| Conll2003_Organization | 36.9 | 85.1 | 93.6 |
| Conll2003_Person | 13.7 | 95.0 | 98.7 |
| Fruit_V_Animal_3 | 17.6 | 100.0 | 99.5 |
| Fruit_V_Animal_5 | 6.7 | 99.5 | 99.5 |
| Object_V_Concept_3 | 21.4 | 98.6 | 99.1 |
| Object_V_Concept_5 | 15.7 | 89.1 | 99.1 |
| Squad_Val | 48.8 | 88.4 | 90.4 |
| Verb_V_Adjective_3 | 22.4 | 90.5 | 100.0 |
| Verb_V_Adjective_5 | 12.9 | 96.7 | 99.5 |
| Average | 23.0 | 82.7 | **97.1** |

Table 23: **Task-wise performance on 57 ICL tasks using Llama-3.1-70B-Instruct.** Our method is evaluated along with 0-shot and 10-shot baselines. (a) Results on 29 abstractive tasks. (b) Results on 28 extractive tasks. The best results are shown in **bold**, and the second-best results are underlined.

(a) Abstractive task results

| Task Name | 0-shot | 10-shot | Ours |
|---|---|---|---|
| AG_News | 0.4 | 83.6 | 91.4 |
| Antonym | 23.4 | 70.6 | 71.0 |
| Capitalize | 4.1 | 100.0 | 99.4 |
| Capitalize_First_Letter | 3.5 | 100.0 | 100.0 |
| Capitalize_Last_Letter | 0.6 | 54.4 | 97.7 |
| Capitalize_Second_Letter | 0.6 | 41.8 | 98.8 |
| Commonsense_QA | 76.0 | 80.8 | 77.6 |
| Country-Capital | 7.1 | 90.5 | 88.1 |
| Country-Currency | 0.0 | 76.2 | 85.7 |
| English-French | 0.3 | 86.2 | 86.3 |
| English-German | 1.4 | 81.2 | 80.4 |
| English-Spanish | 0.5 | 88.6 | 88.2 |
| Landmark-Country | 1.7 | 88.0 | 84.6 |
| Lowercase_First_Letter | 0.0 | 100.0 | 100.0 |
| Lowercase_Last_Letter | 0.0 | 77.8 | 98.3 |
| National_Parks | 6.3 | 83.2 | 76.8 |
| Next_Capital_Letter | 1.2 | 18.1 | 98.8 |
| Next_Item | 6.4 | 97.9 | 95.7 |
| Park-Country | 3.8 | 89.8 | 84.1 |
| Person-Instrument | 0.0 | 76.6 | 81.3 |
| Person-Occupation | 0.0 | 63.4 | 72.7 |
| Person-Sport | 0.0 | 97.0 | 98.5 |
| Present-Past | 3.3 | 93.4 | 100.0 |
| Prev_Item | 8.5 | 97.9 | 97.9 |
| Product-Company | 1.8 | 87.2 | 88.1 |
| Sentiment | 0.0 | 94.3 | 95.9 |
| Singular-Plural | 4.7 | 100.0 | 100.0 |
| Synonym | 9.4 | 55.0 | 57.1 |
| Word_Length | 0.0 | 87.1 | 83.6 |
| Average | 5.7 | 81.4 | **88.9** |

(b) Extractive task results

| Task Name | 0-shot | 10-shot | Ours |
|---|---|---|---|
| Adjective_V_Verb_3 | 26.7 | 94.8 | 99.5 |
| Adjective_V_Verb_5 | 8.1 | 95.7 | 99.5 |
| Alphabetically_First_3 | 18.1 | 37.6 | 96.7 |
| Alphabetically_First_5 | 6.2 | 25.7 | 96.2 |
| Alphabetically_Last_3 | 14.3 | 38.6 | 93.3 |
| Alphabetically_Last_5 | 0.5 | 21.0 | 78.6 |
| Animal_V_Object_3 | 5.2 | 99.5 | 99.1 |
| Animal_V_Object_5 | 1.0 | 100.0 | 99.5 |
| Choose_First_Of_3 | 27.6 | 98.6 | 100.0 |
| Choose_First_Of_5 | 17.6 | 98.6 | 100.0 |
| Choose_Last_Of_3 | 5.2 | 94.8 | 100.0 |
| Choose_Last_Of_5 | 2.9 | 92.4 | 100.0 |
| Choose_Middle_Of_3 | 1.4 | 60.5 | 100.0 |
| Choose_Middle_Of_5 | 1.4 | 31.9 | 99.5 |
| Color_V_Animal_3 | 3.8 | 99.1 | 100.0 |
| Color_V_Animal_5 | 0.5 | 98.1 | 100.0 |
| Concept_V_Object_3 | 3.8 | 96.2 | 100.0 |
| Concept_V_Object_5 | 1.4 | 97.1 | 99.1 |
| Conll2003_Location | 24.3 | 92.1 | 97.1 |
| Conll2003_Organization | 40.9 | 81.4 | 94.2 |
| Conll2003_Person | 21.9 | 96.9 | 98.1 |
| Fruit_V_Animal_3 | 5.2 | 100.0 | 99.5 |
| Fruit_V_Animal_5 | 1.4 | 100.0 | 99.5 |
| Object_V_Concept_3 | 13.3 | 99.1 | 99.1 |
| Object_V_Concept_5 | 4.8 | 94.8 | 100.0 |
| Squad_Val | 66.8 | 88.9 | 91.4 |
| Verb_V_Adjective_3 | 11.9 | 94.3 | 99.1 |
| Verb_V_Adjective_5 | 4.3 | 97.1 | 99.5 |
| Average | 12.2 | 83.0 | **97.8** |

# D    ADDITIONAL EVALUATION ON MMLU-PRO

Throughout this paper, we use 57 ICL tasks introduced in FV (Todd et al., 2023), which span diverse problem types and provide a broad testbed for validating the overall potential of our soft injection method. These tasks include both closely related and contrasting variants, making them well suited for analyzing how well task-relevant attention heads transfer across tasks of varying similarity (see Section 4). However, they do not cover the broader range of real-world problems typically used to evaluate LLM performance. To address this gap and test whether our method generalizes to a widely adopted benchmark, we further evaluate it on the recently proposed MMLU-Pro (Wang et al., 2024). MMLU-Pro extends the original MMLU (Hendrycks et al., 2020) by covering 14 diverse disciplines, including mathematics, law, and health, with more challenging and realistic question sets.

## D.1    EXPERIMENTAL SETUP

Since the MMLU-Pro benchmark does not provide training or validation splits, we partitioned the original test set of each discipline into training, validation, and test splits, using the same ratio as in the 57 ICL tasks. For each discipline, the training and validation splits were used to construct task embeddings and optimize soft head-selection parameters, while the test split was reserved exclusively for performance evaluation. We evaluated our method on two LLMs, Llama-3.1-8B and Llama-3.1-70B, against both 0-shot and 10-shot baselines. Results for MTV (Huang et al., 2024) are also reported on both models, while FV is omitted due to its prohibitively long inference time and comparatively low accuracy. All hyperparameters follow the settings described in Section 3.1 and Section B.2.

## D.2    EXPERIMENTAL RESULTS

Table 24 presents results on the MMLU-Pro benchmark, which spans 14 disciplines. Despite using 0-shot prompts at inference, our method achieves the highest performance on both Llama-3.1-8B and Llama-3.1-70B, surpassing the 10-shot baseline by 3.2% and 7.7%, respectively. MTV shows mixed behavior: on Llama-3.1-70B it exceeds the 10-shot baseline but still falls well short of our method, while on Llama-3.1-8B it performs worse than the 10-shot baseline. These outcomes highlight the effectiveness of our approach: by applying gradient descent to a continuous reformulation of head selection, our method more reliably identifies task-relevant head positions than MTV's RL-based optimization of a head-sampling distribution. Overall, these results demonstrate that our approach generalizes effectively to a widely adopted LLM benchmark, delivering strong task performance without requiring in-prompt demonstrations at inference.

Table 24: **Evaluation on MMLU-Pro (14 disciplines).** Two injection methods (MTV, Ours) are evaluated along with 0-shot and 10-shot baselines. (a) Results on Llama-3.1-8B. (b) Results on Llama-3.1-70B. The best results are shown in **bold**, and the second-best results are underlined.

(a) Results on Llama-3.1-8B

| Disciplines | 0-shot | 10-shot | MTV | Ours |
|---|---|---|---|---|
| Biology | 34.4 | 68.9 | 65.6 | **69.5** |
| Business | 7.3 | 21.8 | 20.6 | **29.1** |
| Chemistry | 3.8 | 23.5 | 26.5 | **31.9** |
| Computer Science | 18.6 | 36.1 | 33.7 | **37.2** |
| Economics | 21.5 | 40.1 | 39.6 | **44.6** |
| Engineering | 3.0 | 22.7 | 25.1 | **37.9** |
| Health | 30.2 | 44.2 | 40.1 | **47.7** |
| History | 22.5 | **48.8** | 38.8 | 47.5 |
| Law | 16.9 | **29.4** | 21.2 | 27.3 |
| Math | 1.8 | 20.1 | 19.4 | **22.2** |
| Philosophy | 29.5 | **36.2** | 31.4 | **36.2** |
| Physics | 2.9 | 23.1 | 22.0 | **27.8** |
| Psychology | 33.3 | **56.0** | 50.0 | 55.4 |
| Other | 25.8 | 36.1 | 36.6 | **37.1** |
| Average | 18.0 | 36.2 | 33.6 | **39.4** |

(b) Results on Llama-3.1-70B

| Disciplines | 0-shot | 10-shot | MTV | Ours |
|---|---|---|---|---|
| Biology | 28.5 | 74.2 | 80.1 | **82.8** |
| Business | 9.1 | 30.9 | 35.2 | **38.8** |
| Chemistry | 8.8 | 26.9 | 27.7 | **33.2** |
| Computer Science | 19.8 | 37.2 | 43.0 | **53.5** |
| Economics | 20.9 | 59.9 | 61.0 | **62.2** |
| Engineering | 5.9 | 24.6 | 34.5 | **46.3** |
| Health | 48.3 | **56.4** | 55.2 | **56.4** |
| History | 11.3 | 47.5 | 56.3 | **62.5** |
| Law | 19.1 | 23.4 | 35.9 | **36.8** |
| Math | 8.8 | 22.9 | 21.5 | **24.3** |
| Philosophy | 25.7 | 52.4 | 47.6 | **54.3** |
| Physics | 11.0 | 29.3 | 30.8 | **39.6** |
| Psychology | 34.5 | 64.3 | 64.9 | **67.3** |
| Other | 21.1 | **50.5** | 48.5 | **50.5** |
| Average | 19.5 | 42.9 | 45.9 | **50.6** |

# E EXTENDED ANALYSIS OF TASK-RELEVANT HEAD IDENTIFICATION

## E.1 OPTIMIZED SOFT HEAD-SELECTION VALUES FOR ALL 57 TASKS

In this section, we present extended results of Figure 3 in Section 4.1, showing the optimized values of the soft head-selection parameters for all 57 tasks using Llama-3.1-8B. The full set of results is provided in Figures 6-8

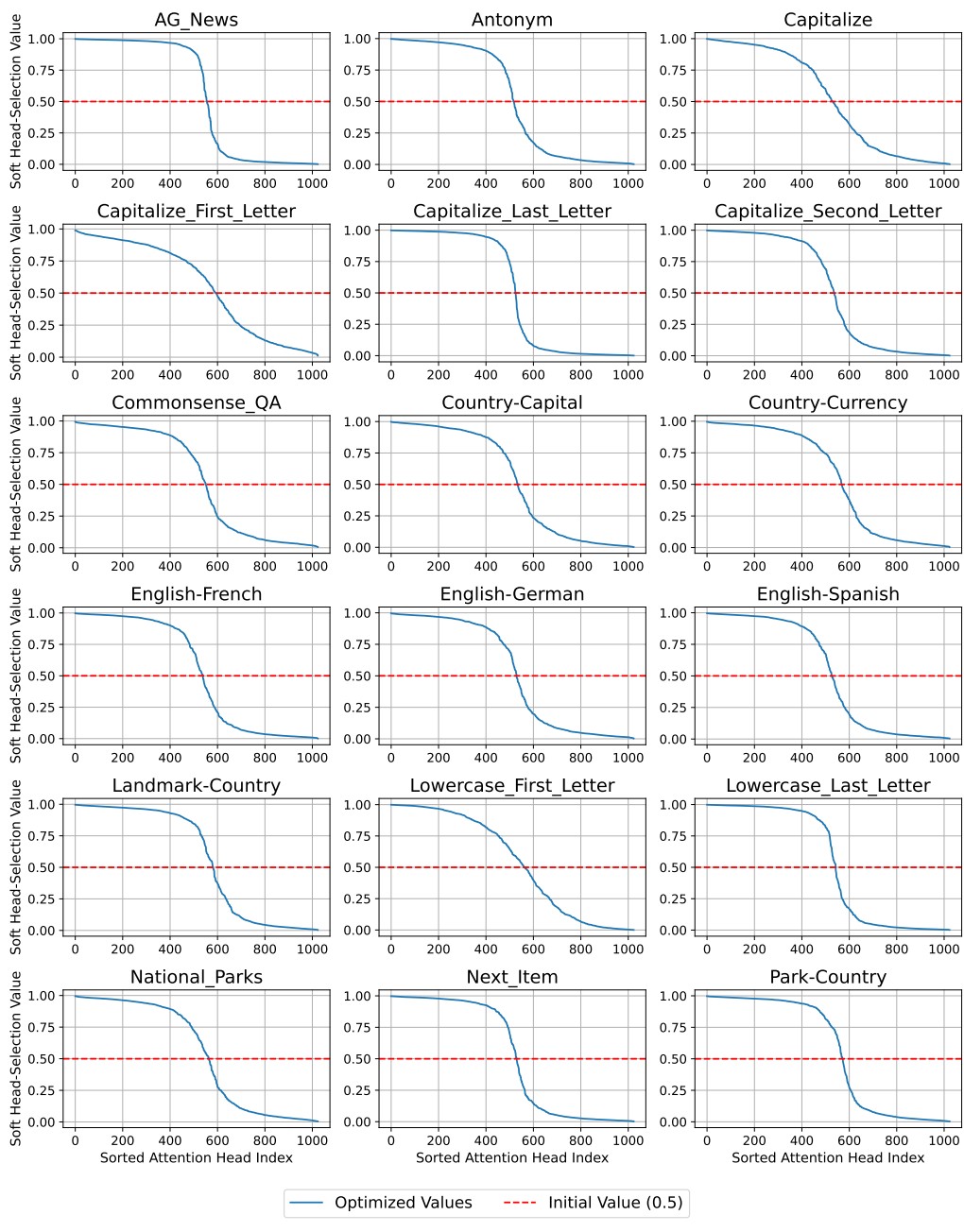

Figure 6: **Optimized values of soft head-selection parameters for 57 ICL tasks (Part 1 of 3).** Each plot shows the optimized values of the soft head-selection parameters for all 1024 attention heads in Llama-3.1-8B, sorted in descending order. Dashed lines indicate the initial value of 0.5 assigned to all selection parameters at the start of training. Plots for the remaining tasks are provided in Figures 7-8.

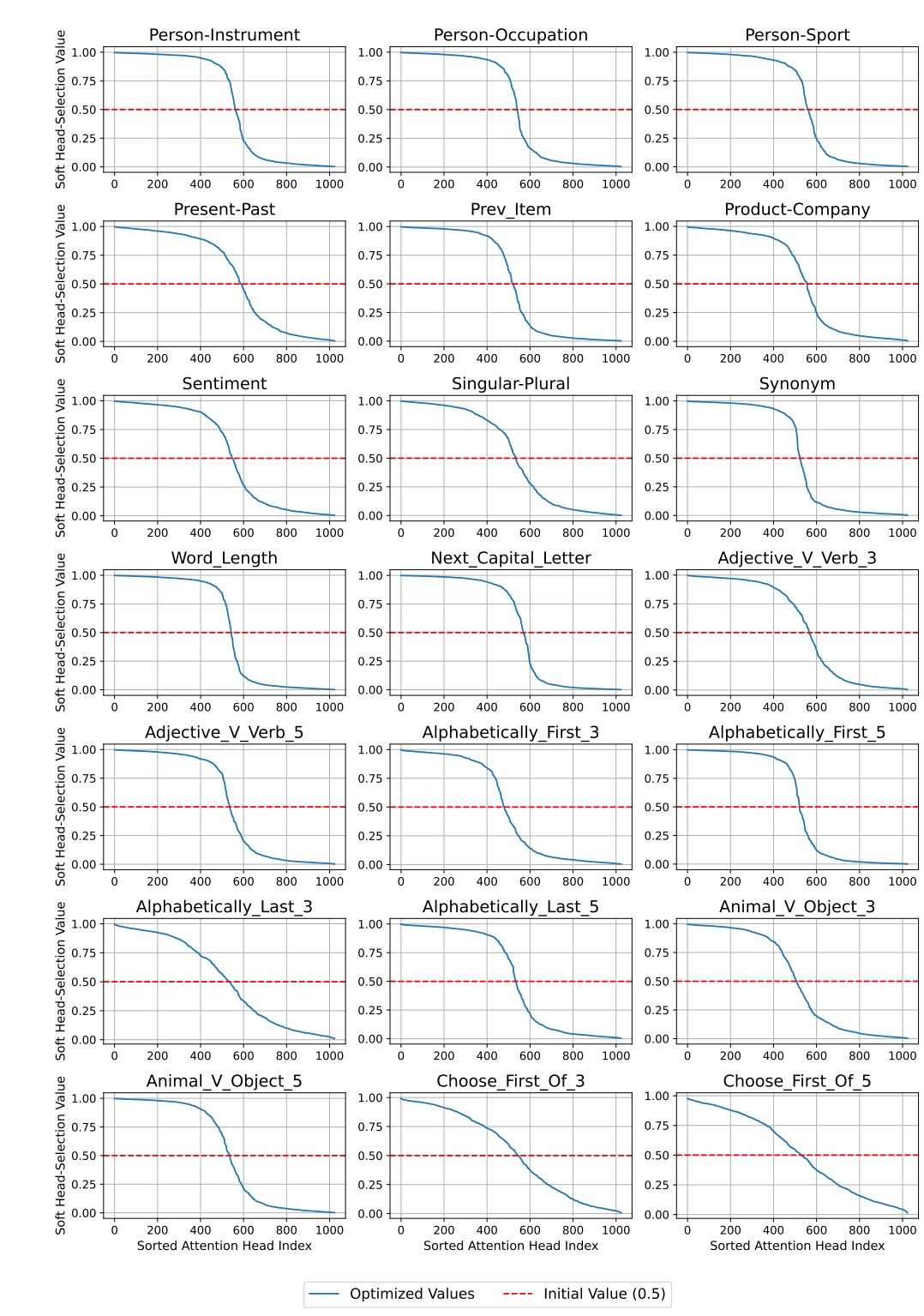

Figure 7: **Optimized values of soft head-selection parameters for 57 ICL tasks (Part 2 of 3).** This figure continues from Figure 6. Each plot shows the optimized values of the soft head-selection parameters for all 1024 attention heads in Llama-3.1-8B, sorted in descending order. Dashed lines indicate the initial value of 0.5 assigned to all selection parameters at the start of training. Plots for the remaining tasks are provided in Figure 8.

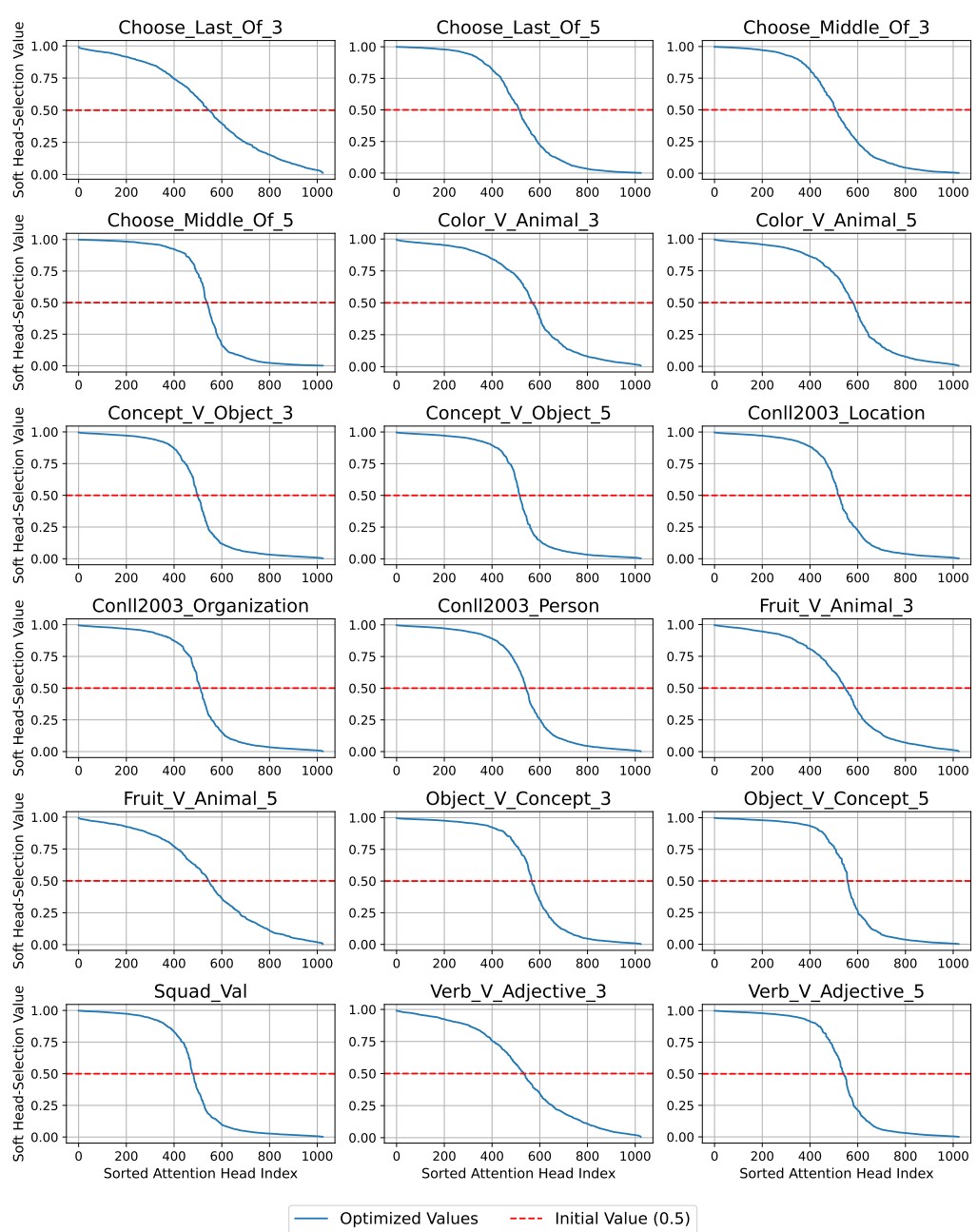

Figure 8: **Optimized values of soft head-selection parameters for 57 ICL tasks (Part 3 of 3).** This figure concludes the series from Figures 6-7. Each plot shows the optimized values of the soft head-selection parameters for all 1024 attention heads in Llama-3.1-8B, sorted in descending order. Dashed lines indicate the initial value of 0.5 assigned to all selection parameters at the start of training.

## E.2 Optimized soft head-selection values for larger language models

Figures 9-11 present the optimized values of the soft head-selection parameters for larger models–Qwen3-32B, Mixtral-8x7B-v0.1, and Llama-3.1-70B–across six selected tasks. The plots reveal consistent overall patterns, closely matching those observed for Llama-3.1-8B in Section 4.1.

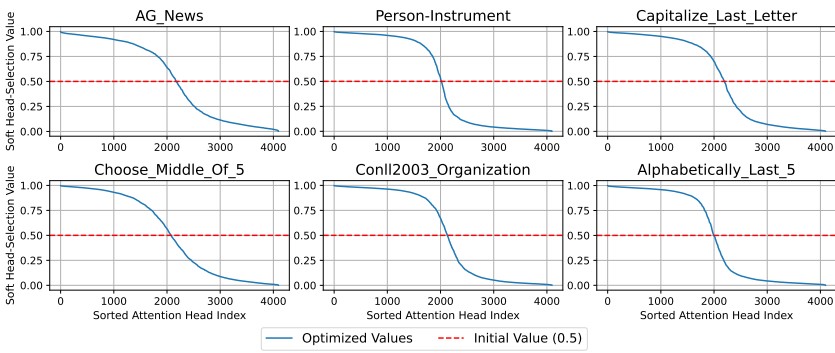

Figure 9: **Optimized values of soft head-selection parameters for six ICL tasks using Qwen3-32B.** Each plot shows the optimized values of the soft head-selection parameters for all 4096 attention heads in Qwen3-32B, sorted in descending order. Dashed lines indicate the initial value of 0.5 assigned to all selection parameters at the start of training.

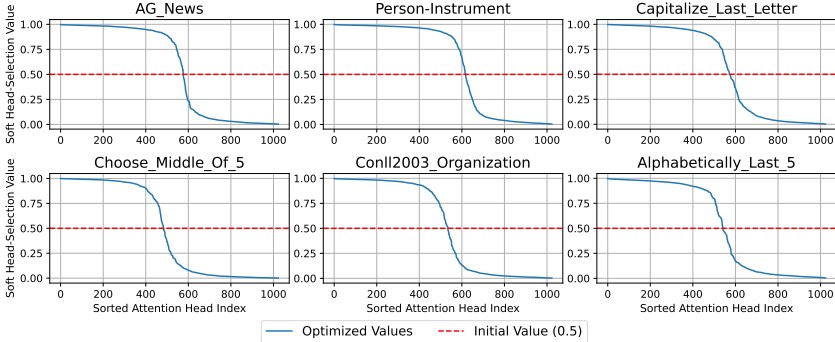

Figure 10: **Optimized values of soft head-selection parameters for six ICL tasks using Mixtral-8x7B-v0.1.** Each plot shows the optimized values of the soft head-selection parameters for all 1024 attention heads in Mixtral-8x7B-v0.1, sorted in descending order. Dashed lines indicate the initial value of 0.5 assigned to all selection parameters at the start of training.

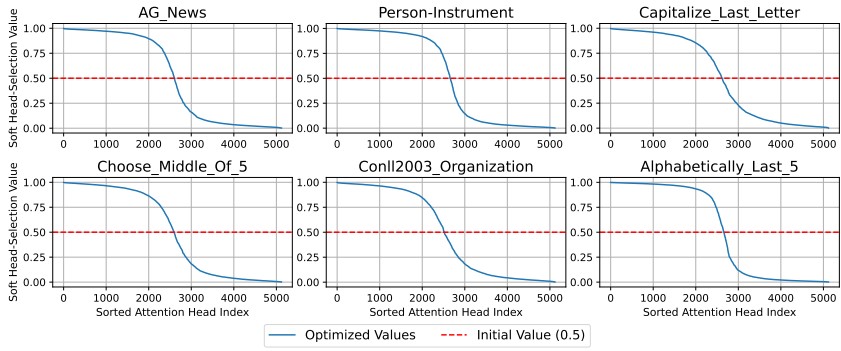

Figure 11: **Optimized values of soft head-selection parameters for six ICL tasks using Llama-3.1-70B.** Each plot shows the optimized values of the soft head-selection parameters for all 5120 attention heads in Llama-3.1-70B, sorted in descending order. Dashed lines indicate the initial value of 0.5 assigned to all selection parameters at the start of training.

## E.3 TASK-WISE PERFORMANCE OF HARD INJECTION

In this section, we present the task-wise performance of the *hard injection* variant described in Section 4.1, evaluated across all 57 tasks using Llama-3.1-8B. Table 25 presents the results in comparison with our default soft injection method, as well as the 0-shot and 10-shot baselines. Note that the hard injection variant uses binary (0 or 1) head-selection parameters, obtained by thresholding the optimized soft head-selection parameters at 0.5.

Table 25: **Task-wise performance of the hard injection variant across 57 ICL tasks using Llama-3.1-8B.** We compare the hard injection variant with our default soft injection method, as well as the 0-shot and 10-shot baselines. '**Ours (Hard)**' denotes the hard injection variant described in Section 4.1, while '**Ours**' refers to the default soft injection method. (a) Results on 29 abstractive tasks. (b) Results on 28 extractive tasks. The best results are shown in **bold**, and the second-best results are underlined.

(a) Abstractive task results

| Task Name | 0-shot | 10-shot | Ours | Ours (Hard) |
|---|---|---|---|---|
| AG_News | 0.4 | 79.1 | 88.7 | 88.2 |
| Antonym | 0.0 | 69.6 | 71.2 | 70.0 |
| Capitalize | 5.3 | 99.4 | 100.0 | 100.0 |
| Capitalize_First_Letter | 10.0 | 100.0 | 100.0 | 100.0 |
| Capitalize_Last_Letter | 1.2 | 24.0 | 93.0 | 87.7 |
| Capitalize_Second_Letter | 1.2 | 28.5 | 97.0 | 97.0 |
| Commonsense_QA | 40.3 | 72.3 | 62.7 | 61.8 |
| Country-Capital | 4.8 | 92.9 | 95.2 | 92.9 |
| Country-Currency | 0.0 | 78.6 | 81.0 | 81.0 |
| English-French | 0.5 | 81.7 | 81.7 | 81.2 |
| English-German | 1.2 | 75.5 | 69.3 | 68.2 |
| English-Spanish | 0.2 | 84.1 | 83.1 | 82.9 |
| Landmark-Country | 0.0 | 92.6 | 86.9 | 86.3 |
| Lowercase_First_Letter | 0.0 | 99.4 | 100.0 | 100.0 |
| Lowercase_Last_Letter | 0.0 | 39.8 | 96.5 | 94.7 |
| National_Parks | 0.0 | 86.3 | 81.1 | 81.1 |
| Next_Capital_Letter | 0.6 | 2.9 | 98.8 | 98.8 |
| Next_Item | 2.1 | 97.9 | 97.9 | 97.9 |
| Park-Country | 0.0 | 89.8 | 84.7 | 84.1 |
| Person-Instrument | 0.0 | 83.2 | 87.9 | 87.9 |
| Person-Occupation | 0.0 | 64.5 | 80.8 | 80.2 |
| Person-Sport | 0.0 | 95.5 | 97.0 | 97.0 |
| Present-Past | 3.3 | 100.0 | 100.0 | 98.4 |
| Prev_Item | 2.1 | 97.9 | 97.9 | 97.9 |
| Product-Company | 0.0 | 87.2 | 88.1 | 88.1 |
| Sentiment | 0.0 | 95.1 | 96.3 | 96.3 |
| Singular-Plural | 2.3 | 100.0 | 100.0 | 97.7 |
| Synonym | 1.8 | 50.3 | 53.0 | 51.2 |
| Word_Length | 0.0 | 38.6 | 82.5 | 81.9 |
| Average | 2.7 | 76.1 | **88.0** | 87.2 |

(b) Extractive task results

| Task Name | 0-shot | 10-shot | Ours | Ours (Hard) |
|---|---|---|---|---|
| Adjective_V_Verb_3 | 14.3 | 87.6 | 99.5 | 97.1 |
| Adjective_V_Verb_5 | 9.1 | 84.8 | 98.1 | 94.3 |
| Alphabetically_First_3 | 21.9 | 42.4 | 57.1 | 52.9 |
| Alphabetically_First_5 | 16.7 | 18.6 | 90.5 | 84.8 |
| Alphabetically_Last_3 | 16.2 | 36.2 | 47.6 | 46.2 |
| Alphabetically_Last_5 | 10.5 | 23.3 | 44.3 | 39.1 |
| Animal_V_Object_3 | 12.4 | 79.5 | 99.0 | 98.6 |
| Animal_V_Object_5 | 19.1 | 81.0 | 98.1 | 98.1 |
| Choose_First_Of_3 | 52.9 | 98.6 | 100.0 | 100.0 |
| Choose_First_Of_5 | 52.4 | 97.6 | 100.0 | 99.5 |
| Choose_Last_Of_3 | 1.0 | 97.6 | 100.0 | 100.0 |
| Choose_Last_Of_5 | 3.8 | 94.3 | 100.0 | 100.0 |
| Choose_Middle_Of_3 | 2.9 | 52.4 | 99.0 | 98.6 |
| Choose_Middle_Of_5 | 4.3 | 33.3 | 90.0 | 83.8 |
| Color_V_Animal_3 | 16.7 | 95.7 | 99.5 | 99.5 |
| Color_V_Animal_5 | 15.7 | 91.0 | 99.5 | 99.1 |
| Concept_V_Object_3 | 14.3 | 83.8 | 99.5 | 99.1 |
| Concept_V_Object_5 | 17.6 | 83.3 | 93.8 | 91.9 |
| Conll2003_Location | 21.8 | 87.7 | 94.3 | 93.4 |
| Conll2003_Organization | 39.3 | 77.5 | 91.4 | 90.6 |
| Conll2003_Person | 12.4 | 91.7 | 97.3 | 96.4 |
| Fruit_V_Animal_3 | 23.3 | 82.9 | 99.0 | 99.1 |
| Fruit_V_Animal_5 | 10.0 | 79.1 | 99.5 | 97.6 |
| Object_V_Concept_3 | 17.6 | 97.6 | 100.0 | 100.0 |
| Object_V_Concept_5 | 5.7 | 92.4 | 98.1 | 98.1 |
| Squad_Val | 39.4 | 85.4 | 86.7 | 84.1 |
| Verb_V_Adjective_3 | 11.4 | 94.3 | 97.6 | 97.1 |
| Verb_V_Adjective_5 | 1.9 | 93.8 | 99.0 | 98.1 |
| Average | 17.3 | 77.3 | **92.1** | 90.6 |

E.4  COMPARISON OF HARD INJECTION AND ITS COMPLEMENTARY VARIANT

In Appendix E.3, we presented results for the hard injection variant of our method, where task embeddings replace the activations of attention heads whose binarized head-selection parameters are 1 (see Equation 4 of Section 2). In this section, we additionally evaluate its complementary variant, which instead replaces the activations of heads whose binarized head-selection parameters are 0. Since both variants use the same set of task embeddings but inject them into non-overlapping sets of attention heads whose union covers all attention heads, this comparison provides a direct test of whether heads with optimized values near 1 are indeed task-relevant. Table 26 presents results on Llama-3.1-8B, showing a substantial performance gap between the two variants–86.5% on the 29 abstractive tasks and 83.8% on the 28 extractive tasks. Notably, this performance gap arises even though both variants modify a similar number of attention heads (approximately half of the 1024 total), as observed in the distributions of optimized soft head-selection values shown in Figures 6-8. Together, these findings further support our claim that the optimized soft head-selection parameters successfully identify task-relevant attention heads.

Table 26: **Comparison of hard injection and its complementary variant across 57 ICL tasks using Llama-3.1-8B.** We compare two injection variants that are identical except for the choice of attention heads into which task embeddings are injected. '**Ours (Hard)**' denotes the hard injection variant described in Section 4.1, where task embeddings replace the activations of heads whose binarized head-selection values are 1. '**Ours (Hard-Comp)**' refers to its complementary variant, where task embeddings replace the activation of heads whose binarized head-selection values are 0. Both variants use the same set of task embeddings but inject them into non-overlapping sets of attention heads whose union covers all 1024 heads. '**Gap ($\Delta$)**' denotes the accuracy difference between the two variants: **Ours (Hard) – Ours (Hard-Comp)**. (a) Results on 29 abstractive tasks. (b) Results on 28 extractive tasks.

(a) Abstractive task results

| Task Name | Ours (Hard) | Ours (Hard-Comp) | Gap ($\Delta$) |
|---|---|---|---|
| AG_News | 88.2 | 4.6 | 83.6 |
| Antonym | 70.0 | 0.0 | 70.0 |
| Capitalize | 100.0 | 0.0 | 100.0 |
| Capitalize_First_Letter | 100.0 | 0.0 | 100.0 |
| Capitalize_Last_Letter | 87.7 | 0.0 | 87.7 |
| Capitalize_Second_Letter | 97.0 | 4.2 | 92.8 |
| Commonsense_QA | 61.8 | 3.6 | 58.2 |
| Country-Capital | 92.9 | 0.0 | 92.9 |
| Country-Currency | 81.0 | 0.0 | 81.0 |
| English-French | 81.2 | 0.0 | 81.2 |
| English-German | 68.2 | 0.0 | 68.2 |
| English-Spanish | 82.9 | 0.0 | 82.9 |
| Landmark-Country | 86.3 | 0.0 | 86.3 |
| Lowercase_First_Letter | 100.0 | 0.0 | 100.0 |
| Lowercase_Last_Letter | 94.7 | 0.0 | 94.7 |
| National_Parks | 81.1 | 0.0 | 81.1 |
| Next_Capital_Letter | 98.8 | 0.6 | 98.2 |
| Next_Item | 97.9 | 0.0 | 97.9 |
| Park-Country | 84.1 | 0.0 | 84.1 |
| Person-Instrument | 87.9 | 0.0 | 87.9 |
| Person-Occupation | 80.2 | 0.0 | 80.2 |
| Person-Sport | 97.0 | 0.0 | 97.0 |
| Present-Past | 98.4 | 0.0 | 98.4 |
| Prev_Item | 97.9 | 4.3 | 93.6 |
| Product-Company | 88.1 | 0.0 | 88.1 |
| Sentiment | 96.3 | 0.0 | 96.3 |
| Singular-Plural | 97.7 | 0.0 | 97.7 |
| Synonym | 51.2 | 0.0 | 51.2 |
| Word_Length | 81.9 | 1.8 | 80.1 |
| Average | 87.2 | 0.7 | 86.5 |

(b) Extractive task results

| Task Name | Ours (Hard) | Ours (Hard-Comp) | Gap ($\Delta$) |
|---|---|---|---|
| Adjective_V_Verb_3 | 97.1 | 0.5 | 96.6 |
| Adjective_V_Verb_5 | 94.3 | 1.0 | 93.3 |
| Alphabetically_First_3 | 52.9 | 28.6 | 24.3 |
| Alphabetically_First_5 | 84.8 | 10.5 | 74.3 |
| Alphabetically_Last_3 | 46.2 | 7.6 | 38.6 |
| Alphabetically_Last_5 | 39.1 | 3.3 | 35.8 |
| Animal_V_Object_3 | 98.6 | 9.5 | 89.1 |
| Animal_V_Object_5 | 98.1 | 6.2 | 91.9 |
| Choose_First_Of_3 | 100.0 | 0.0 | 100.0 |
| Choose_First_Of_5 | 99.5 | 2.4 | 97.1 |
| Choose_Last_Of_3 | 100.0 | 17.6 | 82.4 |
| Choose_Last_Of_5 | 100.0 | 17.1 | 82.9 |
| Choose_Middle_Of_3 | 98.6 | 27.1 | 71.5 |
| Choose_Middle_Of_5 | 83.8 | 19.5 | 64.3 |
| Color_V_Animal_3 | 99.5 | 1.9 | 97.6 |
| Color_V_Animal_5 | 99.1 | 7.1 | 92.0 |
| Concept_V_Object_3 | 99.1 | 0.0 | 99.1 |
| Concept_V_Object_5 | 91.9 | 1.4 | 90.5 |
| Conll2003_Location | 93.4 | 0.1 | 93.3 |
| Conll2003_Organization | 90.6 | 2.0 | 88.6 |
| Conll2003_Person | 96.4 | 0.3 | 96.1 |
| Fruit_V_Animal_3 | 99.1 | 5.2 | 93.9 |
| Fruit_V_Animal_5 | 97.6 | 11.9 | 85.7 |
| Object_V_Concept_3 | 100.0 | 0.0 | 100.0 |
| Object_V_Concept_5 | 98.1 | 1.0 | 97.1 |
| Squad_Val | 84.1 | 7.3 | 76.8 |
| Verb_V_Adjective_3 | 97.1 | 1.0 | 96.1 |
| Verb_V_Adjective_5 | 98.1 | 0.0 | 98.1 |
| Average | 90.6 | 6.8 | 83.8 |

# F   ADDITIONAL RESULTS ON CROSS-TASK ANALYSIS

## F.1   ADDITIONAL CROSS-TASK RESULTS FOR LLAMA-3.1-8B

In Table 27, we present the results of the cross-task analysis for 12 additional tasks using Llama-3.1-8B, following the procedure described in Section 4.2. As explained there, cross-task analysis evaluates each evaluation task using its own task embedding (i.e., *what* to inject), while applying soft head-selection parameters (i.e., *where* to inject) derived from other head-selection tasks. We vary the head-selection task across all 57 tasks and report the top-3 and bottom-3 based on accuracy. The overall trends are consistent with those reported in Table 2 of Section 4.2.

Table 27: **Cross-task analysis for 12 additional evaluation tasks using Llama-3.1-8B.** For each evaluation task, soft head-selection parameters are replaced with those from other tasks–changing *where* task information is injected, but not *what* is injected. Among the 57 tasks, the table reports the top-3 and bottom-3 head-selection tasks ranked by accuracy.

| Evaluation Task | Task Description | Top-3 Head-Selection Tasks (Accuracy, %) | Bottom-3 Head-Selection Tasks (Accuracy, %) |
|---|---|---|---|
| Adjective_V_Verb_3 | Select the adjective from a list of 3 words (1 adjective, 2 verbs) | Adjective_V_Verb_3 (99.5) Adjective_V_Verb_5 (99.0) Fruit_V_Animal_5 (89.0) | Verb_V_Adjective_3 (0.5) Verb_V_Adjective_5 (7.6) Squad_Val (15.7) |
| Verb_V_Adjective_3 | Select the verb from a list of 3 words (1 verb, 2 adjectives) | Verb_V_Adjective_5 (99.5) Verb_V_Adjective_3 (97.6) Color_V_Animal_5 (80.5) | Adjective_V_Verb_3 (0.5) Adjective_V_Verb_5 (4.8) Synonym (14.8) |
| Alphabetically_First_3 | Select the word that comes first in alphabetical order from a list of 3 words | Alphabetically_First_5 (86.7) Alphabetically_First_3 (57.1) Next_Item (45.2) | Alphabetically_Last_3 (20.5) Alphabetically_Last_5 (23.3) Park-Country (24.8) |
| Alphabetically_Last_3 | Select the word that comes last in alphabetical order from a list of 3 words | Alphabetically_Last_5 (50.0) Alphabetically_Last_3 (47.6) Park-Country (38.6) | Alphabetically_First_5 (1.0) Alphabetically_First_3 (16.2) English-French (23.3) |
| Concept_V_Object_3 | Select the concept from a list of 3 words (1 abstract concept, 2 concrete entities) | Concept_V_Object_3 (99.5) Concept_V_Object_5 (98.1) Animal_V_Object_5 (71.9) | Object_V_Concept_5 (0.5) Object_V_Concept_3 (2.9) English-French (13.8) |
| Concept_V_Object_5 | Select the concept from a list of 5 words (1 abstract concept, 4 concrete entities) | Concept_V_Object_3 (95.2) Concept_V_Object_5 (93.8) Animal_V_Object_5 (73.3) | Object_V_Concept_5 (2.4) Object_V_Concept_3 (6.2) Park-Country (6.7) |
| Object_V_Concept_3 | Select the concrete entity from a list of 3 words (1 concrete entity, 2 abstract concepts) | Object_V_Concept_3 (100.0) Object_V_Concept_5 (99.0) Color_V_Animal_3 (95.7) | Concept_V_Object_3 (5.7) Concept_V_Object_5 (8.6) Squad_Val (15.2) |
| Object_V_Concept_5 | Select the concrete entity from a list of 5 words (1 concrete entity, 4 abstract concepts) | Object_V_Concept_5 (98.1) Object_V_Concept_3 (96.2) Fruit_V_Animal_3 (92.4) | Concept_V_Object_3 (3.3) Concept_V_Object_5 (3.8) Squad_Val (10.5) |
| Capitalize_First_Letter | Generate the first letter of a given word in captial form | Capitalize_First_Letter (100.0) Lowercase_First_Letter (100.0) Capitalize (100.0) | Choose_Middle_Of_3 (0.0) Conll2003_Organization (0.0) Person-Sport (0.0) |
| Lowercase_First_Letter | Generate the first letter of a given word in lowercase | Lowercase_First_Letter (100.0) Capitalize_First_Letter (100.0) Capitalize (100.0) | Conll2003_Organization (0.0) Person-Sport (0.0) Next_Capital_Letter (0.6) |
| Capitalize_Last_Letter | Generate the last letter of a given word in captial form | Capitalize_Last_Letter (93.0) Lowercase_Last_Lettter (86.0) Choose_Middle_Of_5 (37.4) | Choose_Middle_Of_3 (0.0) Choose_First_Of_5 (0.0) Conll2003_Organization (0.6) |
| Lowercase_Last_Letter | Generate the last letter of a given word in lowercase | Lowercase_Last_Letter (96.5) Capitalize_Last_Letter (93.6) National_Parks (50.3) | Conll2003_Organization (0.0) Conll2003_Location (0.0) Prev_Item (0.0) |

## F.2 CROSS-TASK ANALYSIS RESULTS FOR LARGER LANGUAGE MODELS

Tables 28-30 present the results of the cross-task analysis for larger models: Qwen3-32B, Mixtral-8x7B-v0.1, and Llama-3.1-70B, across 19 evaluation tasks. The overall trends are consistent with those reported for Llama-3.1-8B in Table 2 of Section 4.2 and Table 27.

Table 28: **Cross-task analysis for 19 evaluation tasks using Qwen3-32B.** For each evaluation task, soft head-selection parameters are replaced with those from other tasks–changing *where* task information is injected, but not *what* is injected. Among the 57 tasks, the table reports the top-3 and bottom-3 head-selection tasks ranked by accuracy.

| Evaluation Task | Task Description | Top-3 Head-Selection Tasks (Accuracy, %) | Bottom-3 Head-Selection Tasks (Accuracy, %) |
|---|---|---|---|
| Adjective_V_Verb_3 | Select the adjective from a list of 3 words (1 adjective, 2 verbs) | Adjective_V_Verb_3 (99.5) Adjective_V_Verb_5 (99.0) Animal_V_Object_3 (86.2) | Verb_V_Adjective_3 (3.8) Sentiment (4.8) Person-Occupation (4.8) |
| Adjective_V_Verb_5 | Select the only adjective from a list of 5 words (1 adjective, 4 verbs) | Adjective_V_Verb_5 (98.1) Adjective_V_Verb_3 (97.1) Animal_V_Object_3 (85.2) | Person-Occupation (5.2) Sentiment (8.1) Verb_V_Adjective_3 (9.0) |
| Verb_V_Adjective_3 | Select the verb from a list of 3 words (1 verb, 2 adjectives) | Verb_V_Adjective_3 (100.0) Verb_V_Adjective_5 (98.1) Singular-Plural (86.7) | AG_News (1.4) Person-Occupation (4.8) Adjective_V_Verb_3 (6.7) |
| Verb_V_Adjective_5 | Select the only verb from a list of 5 words (1 verb, 4 adjectives) | Verb_V_Adjective_5 (99.5) Verb_V_Adjective_3 (98.6) Color_V_Animal_3 (94.3) | AG_News (4.3) Person-Occupation (5.2) Sentiment (5.7) |
| Alphabetically_First_3 | Select the word that comes first in alphabetical order from a list of 3 words | Alphabetically_First_3 (97.6) Alphabetically_First_5 (96.2) Animal_V_Object_3 (41.4) | Sentiment (1.4) Person-Occupation (1.9) AG_News (6.7) |
| Alphabetically_First_5 | Choose the word that comes first in alphabetical order from a list of 5 words | Alphabetically_First_5 (92.9) Alphabetically_First_3 (87.6) Antonym (27.6) | Sentiment (3.3) Next_Capital_Letter (8.1) Person-Occupation (8.6) |
| Alphabetically_Last_3 | Select the word that comes last in alphabetical order from a list of 3 words | Alphabetically_Last_5 (45.2) Alphabetically_Last_3 (43.3) Present-Past (42.4) | Alphabetically_First_5 (1.0) Alphabetically_First_3 (1.0) Sentiment (2.4) |
| Alphabetically_Last_5 | Choose the word that comes last in alphabetical order from a list of 5 words | Alphabetically_Last_5 (43.8) Alphabetically_Last_3 (26.7) Choose_Middle_Of_5 (25.2) | Alphabetically_First_5 (0.0) Alphabetically_First_3 (0.0) Sentiment (7.1) |
| Concept_V_Object_3 | Select the concept from a list of 3 words (1 abstract concept, 2 concrete entities) | Concept_V_Object_3 (99.0) Concept_V_Object_5 (96.7) English-Spanish (81.9) | AG_News (4.3) Sentiment (6.2) Object_V_Concept_3 (11.0) |
| Concept_V_Object_5 | Select the concept from a list of 5 words (1 abstract concept, 4 concrete entities) | Concept_V_Object_5 (98.1) Concept_V_Object_3 (97.1) Fruit_V_Animal_3 (83.8) | Sentiment (8.1) AG_News (10.0) Next_Capital_Letter (15.2) |
| Object_V_Concept_3 | Select the concrete entity from a list of 3 words (1 concrete entity, 2 abstract concepts) | Object_V_Concept_3 (99.5) Object_V_Concept_5 (99.0) Country-Capital (91.4) | Person-Occupation (4.3) Sentiment (5.2) AG_News (7.1) |
| Object_V_Concept_5 | Select the concrete entity from a list of 5 words (1 concrete entity, 4 abstract concepts) | Object_V_Concept_3 (99.0) Object_V_Concept_5 (98.6) Adjective_V_Verb_5 (91.0) | Person-Occupation (5.7) Sentiment (6.7) AG_News (9.0) |
| English-French | Translate the given English word into French | English-French (77.2) English-German (77.1) English-Spanish (75.4) | Alphabetically_Last_5 (2.3) Person-Instrument (7.5) Commonsense_QA (8.9) |
| English-German | Translate the given English word into German | English-French (68.1) English-German (66.2) English-Spanish (63.6) | Person-Instrument (4.6) Alphabetically_Last_5 (4.7) Adjective_V_Verb_3 (12.6) |
| English-Spanish | Translate the given English word into Spanish | English-French (82.5) English-German (82.3) English-Spanish (80.1) | Alphabetically_Last_5 (6.3) Person-Instrument (14.3) Next_Capital_Letter (22.5) |
| Capitalize_First_Letter | Generate the first letter of a given word in captial form | Capitalize_First_Letter (100.0) Lowercase_First_Letter (100.0) Capitalize (100.0) | Capitalize_Second_Letter (6.5) Person-Occupation (14.7) Capitalize_Last_Letter (18.8) |
| Lowercase_First_Letter | Generate the first letter of a given word in lowercase | Capitalize_First_Letter (100.0) Concept_V_Object_3 (100.0) Object_V_Concept_3 (100.0) | Conll2003_Person (0.0) Conll2003_Location (0.0) Prev_Item (0.0) |
| Capitalize_Last_Letter | Generate the last letter of a given word in captial form | Capitalize_Last_Letter (93.0) Lowercase_Last_Lettter (84.2) Alphabetically_First_5 (46.8) | Prev_Item (0.6) English-French (2.3) Choose_Middle_Of_5 (4.7) |
| Lowercase_Last_Letter | Generate the last letter of a given word in lowercase | Lowercase_Last_Letter (95.3) Capitalize_Last_Letter (90.1) Verb_V_Adjective_5 (81.9) | Conll2003_Person (0.0) Conll2003_Location (0.0) Prev_Item (0.0) |

Table 29: **Cross-task analysis for 19 evaluation tasks using Mixtral-8x7B-v0.1.** For each evaluation task, soft head-selection parameters are replaced with those from other tasks–changing *where* task information is injected, but not *what* is injected. Among the 57 tasks, the table reports the top-3 and bottom-3 head-selection tasks ranked by accuracy.

| Evaluation Task | Task Description | Top-3 Head-Selection Tasks (Accuracy, %) | Bottom-3 Head-Selection Tasks (Accuracy, %) |
|---|---|---|---|
| Adjective_V_Verb_3 | Select the adjective from a list of 3 words (1 adjective, 2 verbs) | Adjective_V_Verb_3 (99.5) Adjective_V_Verb_5 (99.5) Conll2003_Organization (81.9) | Verb_V_Adjective_3 (1.9) Verb_V_Adjective_5 (6.7) Landmark-Country (22.9) |
| Adjective_V_Verb_5 | Select the only adjective from a list of 5 words (1 adjective, 4 verbs) | Adjective_V_Verb_5 (97.6) Adjective_V_Verb_3 (96.7) Animal_V_Object_5 (81.0) | Verb_V_Adjective_3 (2.4) Verb_V_Adjective_5 (6.2) Word_Length (19.5) |
| Verb_V_Adjective_3 | Select the verb from a list of 3 words (1 verb, 2 adjectives) | Verb_V_Adjective_5 (98.6) Verb_V_Adjective_3 (96.7) Concept_V_Object_5 (62.4) | Adjective_V_Verb_5 (0.0) Adjective_V_Verb_3 (0.5) English-French (17.6) |
| Verb_V_Adjective_5 | Select the only verb from a list of 5 words (1 verb, 4 adjectives) | Verb_V_Adjective_5 (98.6) Verb_V_Adjective_3 (95.7) Color_V_Animal_5 (71.9) | Adjective_V_Verb_3 (0.0) Adjective_V_Verb_5 (1.0) Capitalize_First_Letter (6.7) |
| Alphabetically_First_3 | Select the word that comes first in alphabetical order from a list of 3 words | Alphabetically_First_5 (76.7) Alphabetically_First_3 (46.2) Object_V_Concept_3 (39.0) | Alphabetically_Last_5 (18.6) Alphabetically_Last_3 (20.5) Prev_Item (23.3) |
| Alphabetically_First_5 | Choose the word that comes first in alphabetical order from a list of 5 words | Alphabetically_First_5 (89.0) Alphabetically_First_3 (25.2) Adjective_V_Verb_5 (24.3) | Alphabetically_Last_5 (6.2) Alphabetically_Last_3 (10.0) Fruit_V_Animal_3 (14.3) |
| Alphabetically_Last_3 | Select the word that comes last in alphabetical order from a list of 3 words | Alphabetically_Last_3 (51.0) Alphabetically_Last_5 (50.5) Prev_Item (42.4) | Alphabetically_First_5 (14.3) Alphabetically_First_3 (25.7) Conll2003_Person (27.6) |
| Alphabetically_Last_5 | Choose the word that comes last in alphabetical order from a list of 5 words | Alphabetically_Last_5 (41.9) Alphabetically_Last_3 (33.3) AG_News (26.2) | Alphabetically_First_5 (0.5) Natial_Parks (11.0) Alphabetically_First_3 (11.4) |
| Concept_V_Object_3 | Select the concept from a list of 3 words (1 abstract concept, 2 concrete entities) | Concept_V_Object_5 (100.0) Concept_V_Object_3 (99.0) Fruit_V_Animal_3 (63.8) | Object_V_Concept_3 (2.9) Object_V_Concept_5 (6.7) Person-Instrument (12.9) |
| Concept_V_Object_5 | Select the concept from a list of 5 words (1 abstract concept, 4 concrete entities) | Concept_V_Object_5 (95.7) Concept_V_Object_3 (94.8) Animal_V_Object_5 (76.7) | Object_V_Concept_3 (5.2) Object_V_Concept_5 (5.7) Next_Capital_Letter (12.4) |
| Object_V_Concept_3 | Select the concrete entity from a list of 3 words (1 concrete entity, 2 abstract concepts) | Object_V_Concept_3 (99.0) Object_V_Concept_5 (96.7) Fruit_V_Animal_5 (80.5) | Concept_V_Object_5 (2.4) Concept_V_Object_3 (2.9) Adjective_V_Verb_3 (20.0) |
| Object_V_Concept_5 | Select the concrete entity from a list of 5 words (1 concrete entity, 4 abstract concepts) | Object_V_Concept_5 (97.1) Object_V_Concept_3 (95.7) Fruit_V_Animal_5 (86.7) | Concept_V_Object_3 (1.0) Concept_V_Object_5 (1.9) Adjective_V_Verb_3 (10.0) |
| English-French | Translate the given English word into French | English-Spanish (82.6) English-French (82.0) Capitalize (80.1) | Alphabetically_First_5 (1.1) Conll2003_Organization (4.0) Conll2003_Location (16.7) |
| English-German | Translate the given English word into German | English-Spanish (78.3) English-French (75.2) English-German (74.4) | Alphabetically_First_5 (3.9) Conll2003_Organization (7.6) Animal_V_Object_3 (24.4) |
| English-Spanish | Translate the given English word into Spanish | English-Spanish (87.6) English-German (84.9) English-French (83.8) | Alphabetically_First_5 (8.2) Conll2003_Organization (12.3) Animal_V_Object_3 (42.0) |
| Capitalize_First_Letter | Generate the first letter of a given word in captial form | Capitalize_First_Letter (100.0) Lowercase_First_Letter (100.0) English-French (100.0) | AG_News (0.0) Park-Country (1.2) Landmark-Country (4.1) |
| Lowercase_First_Letter | Generate the first letter of a given word in lowercase | Lowercase_First_Letter (100.0) Capitalize_First_Letter (100.0) Capitalize (100.0) | AG_News (0.0) Conll2003_Organization (0.6) Park-Country (1.8) |
| Capitalize_Last_Letter | Generate the last letter of a given word in captial form | Capitalize_Last_Letter (90.1) Lowercase_Last_Lettter (84.8) Present-Past (56.7) | Next_Capital_Letter (1.8) Choose_First_Of_3 (2.3) Prev_Item (4.1) |
| Lowercase_Last_Letter | Generate the last letter of a given word in lowercase | Lowercase_Last_Letter (94.7) Capitalize_Last_Letter (89.5) Present-Past (70.8) | Conll2003_Organization (0.0) Choose_Middle_Of_5 (0.0) Prev_Item (0.0) |

Table 30: **Cross-task analysis for 19 evaluation tasks using Llama-3.1-70B.** For each evaluation task, soft head-selection parameters are replaced with those from other tasks–changing *where* task information is injected, but not *what* is injected. Among the 57 tasks, the table reports the top-3 and bottom-3 head-selection tasks ranked by accuracy.

| Evaluation Task | Task Description | Top-3 Head-Selection Tasks (Accuracy, %) | Bottom-3 Head-Selection Tasks (Accuracy, %) |
|---|---|---|---|
| Adjective_V_Verb_3 | Select the adjective from a list of 3 words (1 adjective, 2 verbs) | Adjective_V_Verb_3 (100.0) Adjective_V_Verb_5 (99.5) Animal_V_Object_5 (90.5) | Verb_V_Adjective_5 (4.3) Verb_V_Adjective_3 (11.9) Synonym (12.4) |
| Adjective_V_Verb_5 | Select the only adjective from a list of 5 words (1 adjective, 4 verbs) | Adjective_V_Verb_5 (100.0) Adjective_V_Verb_3 (96.7) Animal_V_Object_5 (85.7) | Capitalize_Second_Letter (9.5) Word_Length (10.0) Verb_V_Adjective_5 (10.5) |
| Verb_V_Adjective_3 | Select the verb from a list of 3 words (1 verb, 2 adjectives) | Verb_V_Adjective_3 (100.0) Verb_V_Adjective_5 (100.0) Animal_V_Object_5 (93.3) | Antonym (6.2) Adjective_V_Verb_3 (7.1) English-German (13.8) |
| Verb_V_Adjective_5 | Select the only verb from a list of 5 words (1 verb, 4 adjectives) | Verb_V_Adjective_3 (99.5) Verb_V_Adjective_5 (99.5) Animal_V_Object_5 (95.7) | Antonym (1.0) English-German (3.8) Capitalize_Second_Letter (6.2) |
| Alphabetically_First_3 | Select the word that comes first in alphabetical order from a list of 3 words | Alphabetically_First_3 (98.6) Alphabetically_First_5 (95.7) National_Parks (42.9) | Alphabetically_Last_5 (11.0) Alphabetically_Last_3 (13.8) Person-Instrument (13.8) |
| Alphabetically_First_5 | Choose the word that comes first in alphabetical order from a list of 5 words | Alphabetically_First_3 (97.1) Alphabetically_First_5 (96.7) Animal_V_Object_5 (32.9) | Alphabetically_Last_5 (0.5) Alphabetically_Last_3 (4.8) Present-Past (12.4) |
| Alphabetically_Last_3 | Select the word that comes last in alphabetical order from a list of 3 words | Alphabetically_Last_5 (87.1) Alphabetically_Last_3 (62.9) Verb_V_Adjective_5 (43.8) | Alphabetically_First_3 (0.0) Alphabetically_First_5 (0.5) Person-Instrument (11.4) |
| Alphabetically_Last_5 | Choose the word that comes last in alphabetical order from a list of 5 words | Alphabetically_Last_5 (91.4) Alphabetically_Last_3 (52.4) Country-Currency (33.8) | Alphabetically_First_5 (0.0) Alphabetically_First_3 (0.0) Fruit_V_Animal_3 (11.4) |
| Concept_V_Object_3 | Select the concept from a list of 3 words (1 abstract concept, 2 concrete entities) | Concept_V_Object_5 (99.5) Concept_V_Object_3 (99.0) Animal_V_Object_5 (90.0) | Object_V_Concept_5 (8.6) Capitalize_Second_Letter (16.2) Person-Instrument (17.1) |
| Concept_V_Object_5 | Select the concept from a list of 5 words (1 abstract concept, 4 concrete entities) | Concept_V_Object_3 (98.6) Concept_V_Object_5 (97.6) Animal_V_Object_5 (89.0) | Capitalize_Second_Letter (5.7) Word_Length (8.6) English-Spanish (13.8) |
| Object_V_Concept_3 | Select the concrete entity from a list of 3 words (1 concrete entity, 2 abstract concepts) | Object_V_Concept_5 (99.5) Object_V_Concept_3 (99.0) Animal_V_Object_5 (94.3) | Person-Instrument (13.3) English-German (18.6) Concept_V_Object_3 (21.9) |
| Object_V_Concept_5 | Select the concrete entity from a list of 5 words (1 concrete entity, 4 abstract concepts) | Object_V_Concept_5 (99.0) Object_V_Concept_3 (98.1) Animal_V_Object_5 (92.9) | Word_Length (14.8) Choose_First_Of_5 (16.7) Choose_First_Of_3 (16.7) |
| English-French | Translate the given English word into French | English-German (87.0) English-French (85.6) Enlgish-Spanish (85.5) | Person-Instrument (43.7) Choose_Middle_Of_3 (44.0) Person-Sport (51.0) |
| English-German | Translate the given English word into German | English-Spanish (80.6) English-French (80.1) English-German (80.0) | Person-Instrument (32.2) Choose_Middle_Of_3 (33.9) Choose_First_Of_3 (35.7) |
| English-Spanish | Translate the given English word into Spanish | English-Spanish (89.2) English-French (88.9) English-German (88.4) | Person-Instrument (52.5) Choose_Middle_Of_3 (57.2) Person-Sport (58.1) |
| Capitalize_First_Letter | Generate the first letter of a given word in captial form | Capitalize_First_Letter (100.0) Lowercase_First_Letter (100.0) Capitalize (99.4) | Choose_Middle_Of_5 (0.0) Choose_First_Of_5 (0.0) Prev_Item (0.0) |
| Lowercase_First_Letter | Generate the first letter of a given word in lowercase | Lowercase_First_Letter (100.0) Capitalize_First_Letter (100.0) Word_Length (100.0) | Conll2003_Organization (0.0) Choose_First_Of_5 (0.0) Prev_Item (0.0) |
| Capitalize_Last_Letter | Generate the last letter of a given word in captial form | Capitalize_Last_Letter (98.8) Lowercase_Last_Lettter (96.5) Concept_V_Object_3 (52.0) | Choose_Middle_Of_5 (0.0) Choose_First_Of_5 (0.0) Prev_Item (0.0) |
| Lowercase_Last_Letter | Generate the last letter of a given word in lowercase | Lowercase_Last_Letter (99.4) Capitalize_Last_Letter (98.8) Choose_Last_Of_5 (59.1) | Conll2003_Organization (0.0) Choose_First_Of_5 (0.0) Prev_Item (0.0) |

# G EXTENDED RESULTS ON HEAD-SELECTION TRAINING DYNAMICS

## G.1 FULL RESULTS FOR ALL 57 TASKS

In this section, we present extended results of Figure 4 in Section 5, showing the training dynamics–validation loss and test accuracy curves–for all 57 ICL tasks using Llama-3.1-8B. The full set of results is provided in Figures 12-14.

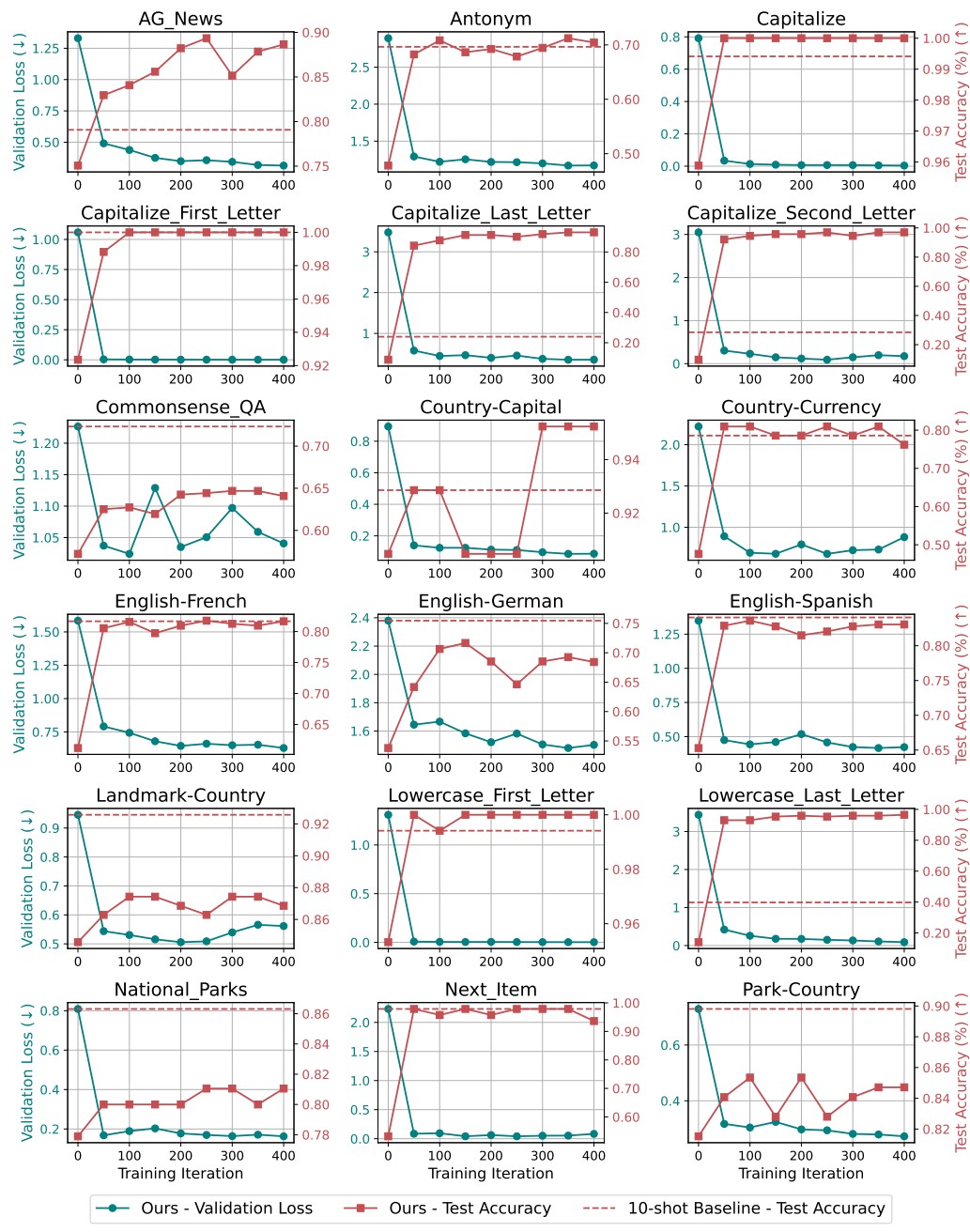

Figure 12: **Training dynamics of soft head-selection parameters for 57 ICL tasks (Part 1 of 3).** Validation loss (left y-axis) and test accuracy (right y-axis) are plotted over 400 training iterations. Dashed lines indicate the 10-shot baseline accuracies for reference. The results are based on Llama-3.1-8B. Plots for the remaining tasks are provided in Figure 13-14.

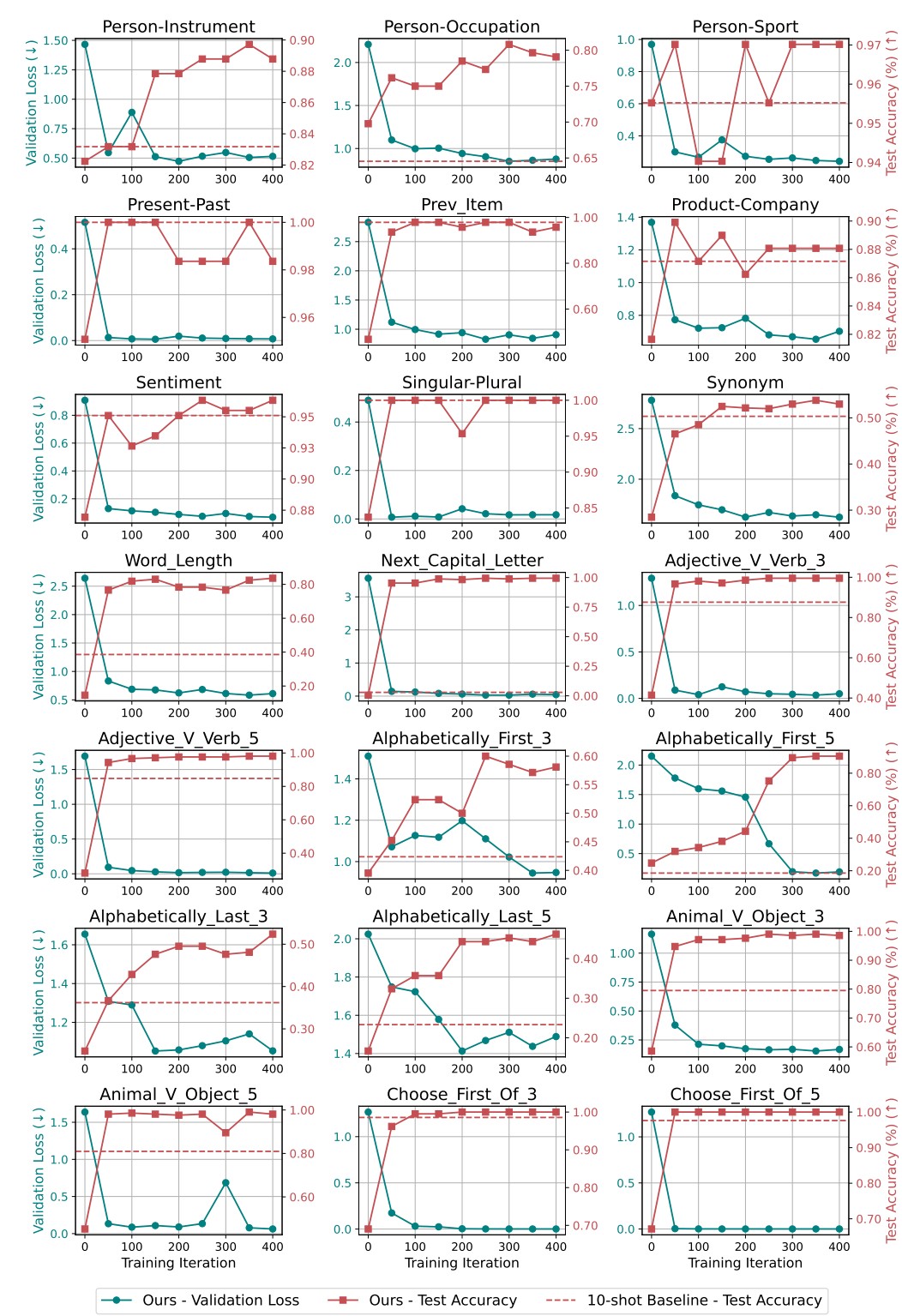

Figure 13: **Training dynamics of soft head-selection parameters for 57 ICL tasks (Part 2 of 3).** This figure continues from Figure 12. Validation loss (left y-axis) and test accuracy (right y-axis) are plotted over 400 training iterations. Dashed lines indicate the 10-shot baseline accuracies for reference. The results are based on Llama-3.1-8B. Plots for the remaining tasks are provided in Figure 14.

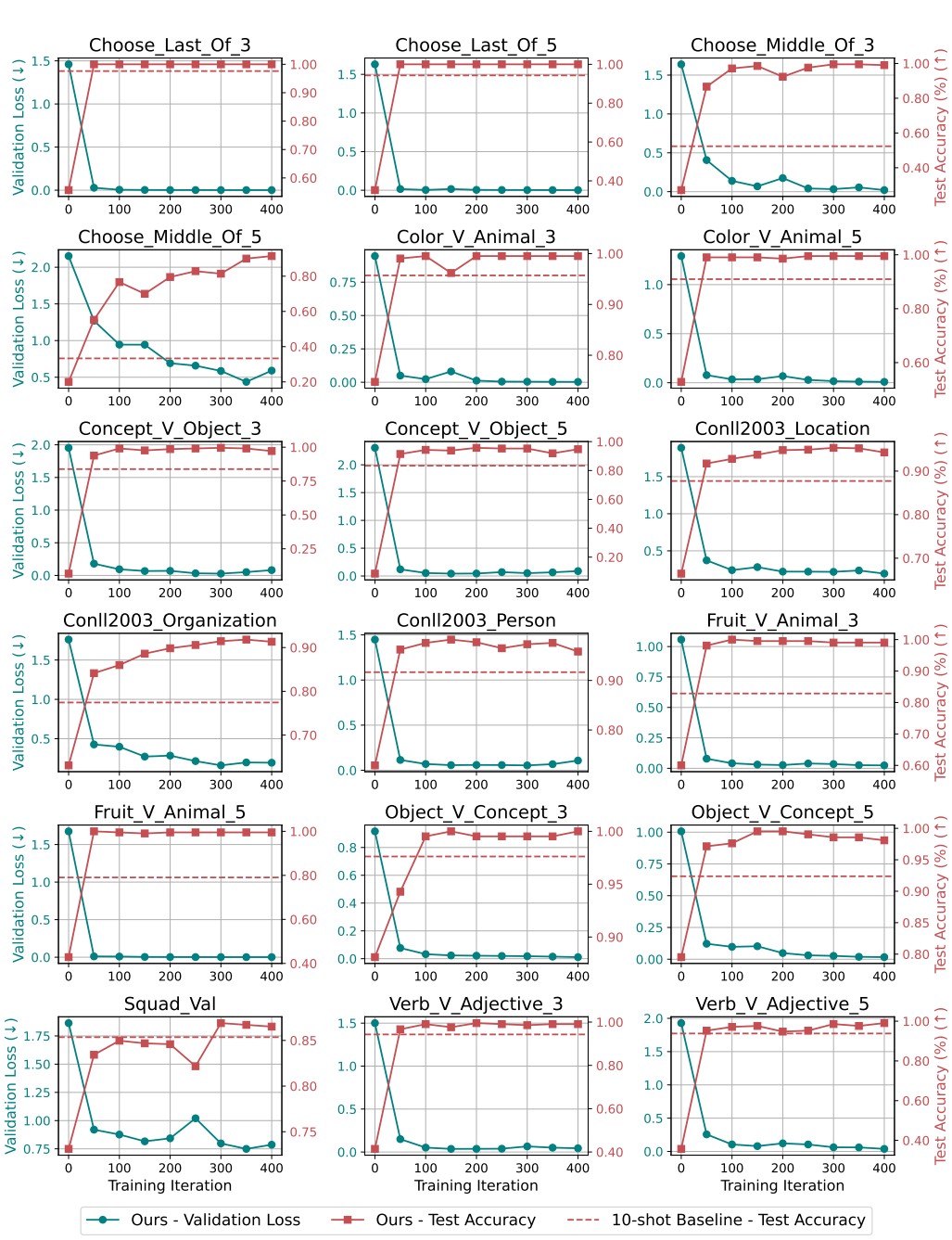

Figure 14: **Training dynamics of soft head-selection parameters for 57 ICL tasks (Part 3 of 3).** This figure concludes the series from Figures 12-13. Validation loss (left y-axis) and test accuracy (right y-axis) are plotted over 400 training iterations. Dashed lines indicate the 10-shot baseline accuracies for reference. The results are based on Llama-3.1-8B.

## G.2 RESULTS FOR LARGER LANGUAGE MODELS

Figures 15-17 present training dynamics for larger models–Qwen3-32B, Mixtral-8x7B-v0.1, and Llama-3.1-70B–across six selected tasks. The resulting plots exhibit consistent overall trends, similar to those observed with Llama-3.1-8B in Section 5.

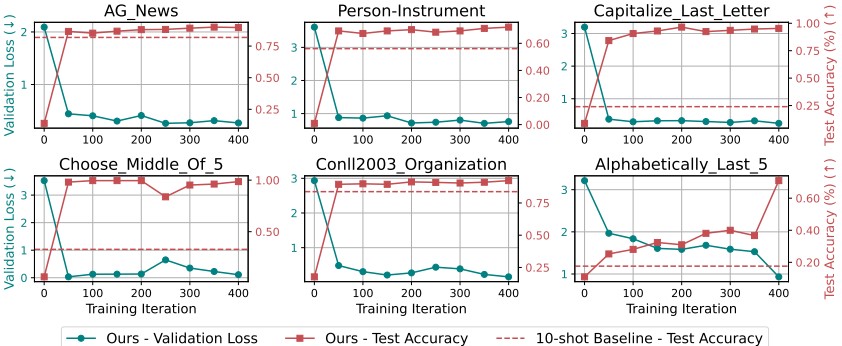

Figure 15: **Training dynamics of soft head-selection parameters for six ICL tasks using Qwen3-32B.** Validation loss (left y-axis) and test accuracy (right y-axis) are plotted over 400 training iterations. Dashed lines indicate the 10-shot baseline accuracies for reference.

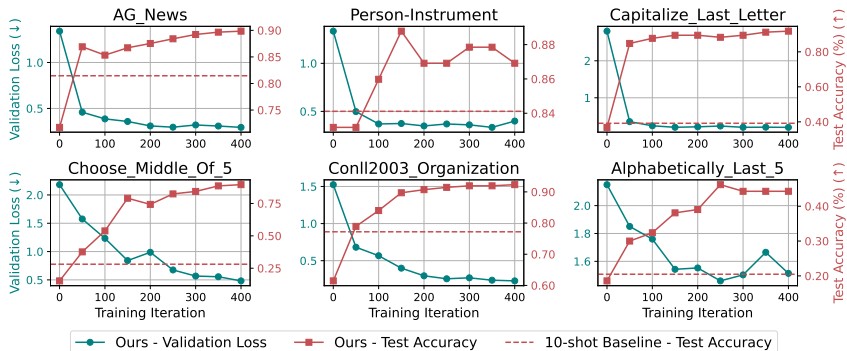

Figure 16: **Training dynamics of soft head-selection parameters for six ICL tasks using Mixtral-8x7B-v0.1.** Validation loss (left y-axis) and test accuracy (right y-axis) are plotted over 400 training iterations. Dashed lines indicate the 10-shot baseline accuracies for reference.

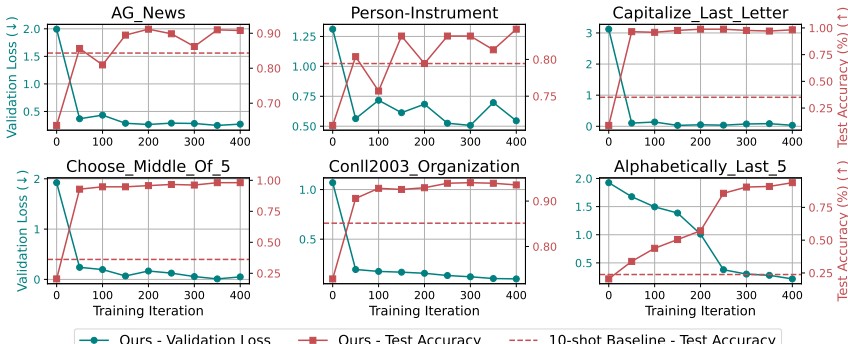

Figure 17: **Training dynamics of soft head-selection parameters for six ICL tasks using Llama-3.1-70B.** Validation loss (left y-axis) and test accuracy (right y-axis) are plotted over 400 training iterations. Dashed lines indicate the 10-shot baseline accuracies for reference.

## H  LIMITATION AND FUTURE WORK

While SITE achieves strong performance gains over 10-shot ICL with lower memory usage and compute cost at inference, it has two main limitations. First, it requires a modest number of labeled examples to construct task embeddings and optimize soft head-selection parameters. Although not included in our reported experiments, we observed that only 30-50 labeled examples were generally sufficient to achieve strong performance, likely because SITE optimizes only a small number of parameters, specifically $L \times H$ head-selection scalars (e.g., 1024 for Llama-3.1-8B). Nonetheless, acquiring even this amount of labeled data may be difficult in low-resource or newly defined tasks. A promising direction is to synthesize additional labeled examples using state-of-the-art LLMs conditioned on a small seed set, as recent studies have shown that LLM-generated synthetic data can rival or even surpass human-curated data in quality (Long et al., 2024; Yehudai et al., 2024; Nadas et al., 2025). Second, SITE requires access to internal model components, specifically attention head activations, which limits its applicability to open-source LLMs and prevents deployment on proprietary models such as GPT-5 (OpenAI, 2025) or Gemini 2.5 (Comanici et al., 2025). Future work may address these limitations by developing methods that further reduce supervision requirements through cross-task transfer and enable operation without architectural access, thereby extending the applicability to low-resource domains and closed-source LLMs.

Beyond performance improvements, SITE also serves as an effective tool for attributing functional roles to attention heads in LLMs. It reformulates head attribution as a continuous optimization problem, wherein soft head-selection parameters are optimized jointly via gradient descent. In contrast, previous head attribution methods (Olsson et al., 2022; Todd et al., 2023; Zhou et al., 2024; Wu et al., 2024) typically identify important heads by ablating them individually or by applying heuristic metrics to estimate their individual contributions. However, such methods may overlook synergistic interactions between heads or fail to account for functional redundancy, potentially missing heads that contribute in complementary or overlapping ways. Extending these methods to evaluate combinations of heads is computationally infeasible due to the exponential number of possible subsets. SITE naturally addresses this challenge by jointly tuning all soft head-selection parameters via gradient descent. As a result, it more effectively identifies task-specific heads while avoiding the limitations of discrete head evaluation. We believe this gradient-based attribution framework can be extended beyond task relevance to investigate other functional roles of attention heads, such as hallucination, safety, or reasoning, which we leave as an exciting direction for future work.

## I  USE OF LLMS IN THIS WORK

We used chat-based LLMs solely for sentence-level editing to check grammar and improve clarity during paper writing. All edits were reviewed and verified by the authors. No scientific ideas, methods, analyses, or results were produced by LLMs; all conceptual contributions and experimental work are solely by the authors.

