# OpenReview forum: "Soft Injection of Task Embeddings Outperforms Prompt-Based In-Context Learning"
_ICLR.cc/2026/Conference — ICLR 2026 Conference Withdrawn Submission_

### Official Review · Reviewer_czAe · 2025-10-22

**Soundness:** 2
**Presentation:** 3
**Contribution:** 2
**Rating:** 2
**Confidence:** 4

**Summary:**

This paper proposes SITE (Soft Injection of Task Embeddings), which:

- Constructs task embeddings by averaging attention head activations across few-shot ICL prompts
- Optimizes soft head-selection parameters (continuous values in [0,1]) via gradient descent to determine where to inject task information
- At inference, performs zero-shot generation by mixing task embeddings with original head activations using learned interpolation weights
- Claims 10.2%-14.3% average gains over 10-shot ICL across 57 tasks and 12 LLMs (4B-70B parameters)
- Additionally analyzes task-relevant attention heads, showing task-specific (not task-agnostic) head functionality

**Strengths:**

**Tasks Tested**

The authors test their method on 57 ICL tasks covering both abstractive and extractive categories, with additional validation on MMLU-Pro (14 disciplines). This breadth of evaluation is commendable.

---

**Models Used**

Extensive evaluation across modern models with scales ranging from 4B to 70B parameters, covering 4 model families (Llama-3.1, Mistral, Qwen3, Gemma-3) and 3 variation types (pretrained, instruction-tuned, MoE).

**Suggestion:** It would be nice to see smaller 1B models (e.g., Llama 3.2-1B, Qwen2.5-1.5B) as well as GPT-2 for comparison with prior work that uses GPT-2.

---

**Performance Gains**

**Accuracy:** Though not fully statistically credible (see my comments below), the current version of the experiments shows that the proposed method achieves consistent improvements of 10.2%–14.3% over 10-shot ICL across all tested models.

**Computational Efficiency:** The method demonstrates computational efficiency at inference time (Table 5), matching 0-shot runtime while requiring only one-time optimization of $L \times H$ parameters (where $L$ and $H$ are the number of layers and attention heads, respectively), with 2-3× speedup vs. 10-shot for large-scale inference (10K+ prompts).

**Weaknesses:**

### **Major**

**Missing Critical Related Work and Baseline**

A recent paper [1] introduces _Learnable Task Vectors_ (LTV) with a nearly identical approach:
1. formulating tasks as weighted sums of attention head activations,
2. learning weights via causal gradient descent,
3. and demonstrating effectiveness across both regression and language tasks.

The core formulation to compute a task embedding is: $v\_t^{\ell} \coloneqq \sum_j \omega\_{\ell, j} \cdot \bar{a}\_{\ell, j}$, where $\ell$ is the layer index, $j$ is the head index, $\bar{a}$ is the mean of the attention activations (given ICL prompts of a certain task), and $\omega$ is the weight assigned to each head. Here, weights are optimized through gradient descent on next-token prediction loss.

The overlap with SITE is substantial—both methods build on the insight that attention heads encode task information, optimize continuous parameters via gradient descent, and inject task-relevant information into attention mechanisms. As far as I could notice, the key apparent differences are:
1. LTV adds weighted task vectors to layer outputs (layer-level injection) while SITE uses soft interpolation at head level:

$
a^{(\ell,h)} \leftarrow (1 - \alpha^{(\ell,h)}) \cdot a^{(\ell,h)} + \alpha^{(\ell,h)} \cdot t^{(\ell,h)},
$


2. and SITE's interpolation formulation vs. LTV's additive combination.

Without citing [1] (with distinguishing how it's different than the proposed work) and experimentally comparing against LTV, the paper's novelty is unclear. _See also the next subsection._

Currently, the paper positions itself primarily against FV and MTV, which are less directly comparable methodologically.

---

**Unclear Design Choices**

The soft interpolation mechanism (using the $\alpha$ parameter) may be a meaningful contribution, though still insufficient in my opinion. Further, no ablation studies of $\alpha$ (to compare it to additive combination, e.g., LTV-style, to demonstrate its superiority) were provided.

That said, the paper lacks justification for key design choices:

1. Why is soft interpolation (convex combination) superior to additive or multiplicative mixing?

2. Why is sigmoid parameterization forcing $\alpha \in [0, 1]$ necessary?

3. Why initialize at 0.5 rather than 0, 1, or per-layer adaptive values? These or similar ablations are critical to validate the design decisions.

---

**Feasibility — Data Requirement**

The paper claims computational efficiency over standard ICL, but this only applies to inference runtime, not data requirements. SITE requires $M=50$ prompts $\times$ $N=10$ examples = 500 labeled examples to construct task embeddings, compared to only 10 examples for 10-shot ICL. This is 50× more data, which raises concerns about:

1. Is the performance gain from better encoding or simply seeing more data during task embedding construction?
2. In low-resource settings where acquiring 500 labeled examples is difficult, SITE may not be applicable, whereas 10-shot ICL remains viable.
3. Table 4 shows that $M=1$ (only 10 examples) already achieves 88.7% vs. 76.7% 10-shot baseline—a 12% gain with equal data. This is buried in the appendix but should be a primary result, as it demonstrates the encoding mechanism's effectiveness independent of data volume.

The authors should either:
1. emphasize the $M=1$ setting as the primary fair comparison,
2. or compare SITE ($M=50$) against 50-shot ICL to control for data usage. Currently, the 10-shot baseline comparison is misleading regarding data efficiency.

Additionally, FV and MTV baselines also construct task embeddings from fewer examples than SITE ($M×N = 500$ for SITE vs. fewer for FV/MTV), which may contribute to the observed performance differences beyond the methodological improvements. For example, comparisons with LTV would help answer this question.

---

**Statistical Reliability of the Experiments**

The experiments were run with greedy decoding only. So we don't have any variance estimates or statistical significance tests for the results, making the results not fully statistically reliable. Even with greedy decoding, variance can arise from:
- random initialization of $\alpha$ parameters (initialized to 0.5, but early gradient noise can lead to different solutions),
- random sampling of $M=50$ prompts for task embedding construction,
- and train/val/test split randomness.

We would expect to see experiments with temperature 0.05 or 0.1 (if deterministic behavior is intended) over at least 3 random seeds/trials, or at minimum, bootstrapped confidence intervals. Without error bars or significance tests, it's impossible to determine whether the 10-14% gains are statistically meaningful or within noise margins, especially for individual tasks where differences may be smaller.

**Suggestion:** Given the scale of experiments (57 tasks × 12 models = 684 conditions), running multiple seeds for all conditions may be expensive, but reporting variance on a representative subset (e.g., Llama-8B or a smaller model on 10 diverse tasks) would significantly strengthen the claims.

---

### **Minor**

**Data Requirement Clarification**
FV and MTV also construct task embeddings from 10-shot prompts (though fewer total examples than SITE's $M×N = 500$). This can be reasonable, as FV and MTV don't optimize parameters via gradient descent (this number is high for LTV as well, which uses ~10K samples for optimization). It would be better for authors to mention this explicitly in the paper to provide context for their design choices, or directly compare against LTV, which also uses a large amount of prompts for training.

---

**Discussion Might Help**
- Injection only at the last token position. For longer inputs, would multi-token injection help?
- How the method would scale (and if it could) to very long contexts (e.g., > 4K tokens)
- **Limited scope:** All 57 tasks are short-form (classification, extraction, single-word generation). Evaluation on long-form generation (summarization, essay writing) or complex reasoning (math, code) would strengthen generalization claims, especially given that you run your experiments with large-enough models (e.g., Llama-8/70B).

---

**Inaccurate Primary Area**

The paper isn't about foundation models. The authors don't introduce a new foundation or frontier model (e.g., an LLM that is continued-pretrained). While I'm not very familiar with this year's primary areas, it seems it falls more likely into representations or learning algorithms rather than foundation models.

**Questions:**

Other than the weaknesses I pointed out, I'd like to hear more about the following questions:
1. Why did the authors choose to inject at head granularity (1024 injection points for Llama-8B) rather than layer granularity (32 injection points), e.g., as done in LTV [1]? What is the conceptual or empirical advantage?
2. MTV's original configuration, which is 100 prompts × 4 shots (correct me if I'm wrong please), was changed to match your setting ($50 \times 10$). Did you ablate this choice? Would MTV perform better with its original configuration? Similarly, did you ensure FV uses comparable data/compute for fair comparison?
3. Table 4 shows $M=1$ (10 examples total) already yields 88.7% vs. 76.7% 10-shot baseline. Why not emphasize this as the primary, more fair comparison? How would SITE with $M=1$ compare to 10-shot ICL on both data efficiency and performance?
4. Have you experimentally compared soft interpolation (Eq. 4) to additive combination (LTV-style:
$a^{(\ell,h)} ← a^{(\ell,h)} + \beta^{(\ell,h)} · t^{(\ell,h)})?$
If soft interpolation is not superior, what is the contribution beyond LTV's formulation?
5. Section 4 shows ~80-90% of $\alpha$ values converge to binary (0 or 1). Which heads are selected? Do they align with induction heads or other known patterns from prior mechanistic interpretability work?

---

### Official Review · Reviewer_DTRG · 2025-10-29

**Soundness:** 3
**Presentation:** 3
**Contribution:** 2
**Rating:** 4
**Confidence:** 3

**Summary:**

The paper presents **SITE** (Soft Injection of Task Embeddings), an activation-level adaptation technique for large language models. The method constructs a compact task embedding for each attention head by running several few-shot prompts and averaging the head’s activations at the final token. It then learns a soft weighting parameter for each head, determining how strongly the task embedding should be blended with the head’s native activation. The base model remains entirely frozen, while only these weighting parameters are trained using next-token prediction loss on zero-shot inputs. During inference, a single injection at the last token is sufficient when the key–value cache is enabled.

Evaluated across 57 tasks and 12 open-source models ranging from 4B to 70B parameters, SITE achieves consistent gains of roughly 10–14% over 10-shot in-context learning and often outperforms prior task-vector baselines. The approach introduces minimal runtime and memory overhead since the task embeddings and head weights are computed once and reused. Moreover, the learned weights provide insight into which attention heads might be more critical.

**Strengths:**

- A well-defined methodology with task vectors that learns continuous per-head mixing rather than making brittle hard choices. This unifies performance steering and lightweight attribution.

- Strong, cross-model, comprehensive evaluation and the results are positive - there is a major gain over many tasks over 10-shot prompting while keeping the base model frozen.

- Near zero-shot runtime and tiny memory footprint make SITE attractive. The per-head weights provide a useful signal about which parts of the model matter for a given task.

- Clear, Well-Written Paper, Easy to follow & understand.

**Weaknesses:**

- **Limited Novelty relative to prior task-vector work:** The ingredients (i.e. head-level task vector injection) are inherited from previous works, and the key addition seems to be soft mixing and learning the per-head weights.

- **Limited analysis of task-level variance:** The paper provides partial patterns, i.e. cross-task transfer of head-selection parameters among semantically similar tasks and shot-count effects where many-shot ICL closes the gap on some datasets but not others, but it does not yet offer a causal or feature-level account of why certain tasks benefit more from SITE than others, which could help the readers to better understand the presented result. (Table 1)

- **Benchmark coverage:** The evaluation emphasizes ICL-style, often short-form tasks, which are limited in scope and difficulty considering the common benchmarks where the evaluated pre-trained models are usually evaluated. It is unclear how SITE performs on harder reasoning, multi-step math, program synthesis, tool use, or long-context tasks where demonstrations might also help.

- **Mechanistic interpretation:** The evidence for task-relevant heads is mostly empirical (weight concentration, transfer across similar tasks). Stronger causal tests (targeted ablations guided by the learned weights, interchange interventions) would bolster the "mechanistic" claim.

**Questions:**

I may have misunderstood certain parts of the paper, so if you could address the points raised in the weaknesses section, it would be of great help to resolve any potential misunderstandings.

and in addition, could you please comment on the following points, which are the major concerns I currently hold?

**Positioning and novelty compared to FV/MTV:**
Beyond the continuous per-head mixing parameterization, what aspects of SITE are genuinely new? A concise side-by-side comparison of the methods, including inputs, training objectives, injection points, parameters trained, computational cost, and evaluation protocol, would help clarify what SITE contributes beyond FV and MTV.

**Why ICL now, and does SITE scale to harder tasks?**
In practice, we often have instruction-tuned backbones or access to fine-tuning datasets, so for relatively simple benchmarks it is not immediately clear when in-context learning, and by extension SITE (which still involves a form of training), is the preferable choice. Could you clarify the concrete scenarios where SITE provides distinct advantages, such as cases involving data governance constraints, rapid per-task adaptation, or strict latency and compute limits? (I can see how ICL might be useful when high-quality training data is scarce, but the current evaluation tasks do not convincingly illustrate why ICL is necessary in these settings, especially given that modern instruction-tuned models already perform strongly on comparable tasks.)

In this context, do you expect SITE and task-vector methods more generally to scale to more complex reasoning or compositional tasks? A small stress-test slice, such as GSM8K for mathematical reasoning, ARC-Challenge for commonsense reasoning, MBPP or HumanEval for code generation, or a long-context task, would strengthen the argument. If running such experiments is not feasible, a discussion of how you expect SITE to behave and what failure modes might emerge on these harder benchmarks would still be valuable.

---

### Official Review · Reviewer_7GdG · 2025-11-04

**Soundness:** 2
**Presentation:** 3
**Contribution:** 2
**Rating:** 2
**Confidence:** 3

**Summary:**

This paper proposes SITE, a method that replaces the need for many in-context demonstrations in in-context learning (ICL) with "task embedding vectors" estimated from $M \cdot N$ labeled examples ($M=50,\ N=10$ in the main experiments), together with a small number (num_layers x num_heads) parameters that are used to combine the task vectors with the outputs of each attention head. The proposed approach outperforms 0- and 10-shot learning on a wide variety of tasks for a wide variety of models.

**Strengths:**

- The proposed method is simple and has very few learnable parameters (layers x num_heads)
- The experimental evaluation is very broad, comparing to ICL with 12 models of various architectures on 57 tasks
- The method shows convincing gains over ICL with 0 and 10 examples per prompt on this wide variety of tasks
- Several of the individual aspects of the method have corresponding ablation studies.
- The proposed approach [almost] doesn't affect the prefill stage, since it only applies to the last token in the prompt. This distinguishes it from several methods discussed below and could have interesting inference efficiency implications.

**Weaknesses:**

- "The task vectors are constructed using $M = 50$ prompts, each containing $N = 10$ input–output pairs." Depending on whether those $N=10$ pairs are shared across every prompt or not, this means SITE gets to access between 50 and 500 labeled pairs, whereas the ICL baselines get either 0 or 10. In the ablation in Table 4 that uses $M=1$, there is no comment on whether the numbers are averaged only over the tasks in each category, or also over multiple choices of the 10 examples for each prompt, and their order. ICL performance is very sensitive to the order of examples and which examples are selected (see, e.g., Zhao et al. "Calibrate Before Use") so it's plausible that the $M=1$ gains over Table 4 are not statistically significant.

- The main weakness of the paper is that there is no mention of the extensive literature on parameter-efficient fine-tuning, which,  like the proposed method, aims to replace ICL with a very small number of learned per-task parameters. Methods like soft prompt tuning, prefix tuning, LoRA, and $\mathrm{IA}^3$ dramatically outperform ICL given a small number of labeled examples, and with up to the ~500 labeled examples used to construct the task vectors in this work, can even match full fine-tuning on some tasks. Without an extensive comparison to these methods, which have the same goal, I don't think the paper can be accepted in its current form.

**Questions:**

- Given 500 labeled examples for a task, how does SITE empirically compare to LoRA, Soft Prompt Tuning, and/or $\mathrm{IA}^3$?
- Are the gains of the $M=1$ setting in Table 4 robust if you randomize which 10 examples get chosen for each prompt and report the average accuracy over several randomizations? What about if you give ICL a simple post-hoc correction like Calibrate Before Use?

---

### Official Review · Reviewer_7ApT · 2025-11-05

**Soundness:** 2
**Presentation:** 2
**Contribution:** 2
**Rating:** 2
**Confidence:** 3

**Summary:**

This paper proposes soft injection as an alternative strategy to ICL for adapting LLMs to execute a particular task. Soft injection is performed by first extracting "task embeddings" for each given task, by averaging the final token activation for each attention head over several few-shot ICL prompts, then learn soft head-selection parameters which are applied at inference time to inject the task information at the last token of the input.

**Strengths:**

- The paper has good empirical rigor, where the method's effectiveness is validated across a broad set of LLMs across 4 model families and parameter sizes of 4B-70B.
- The performance gain using soft injection over ICL is significant
- The interpretability insight of the learned alpha parameters is interesting, to localize certain task mechanisms to specific heads

**Weaknesses:**

My main reservation in recommending this paper for acceptance is that the paper frames soft injection as an alternative to prompt-based ICL, whereas in practice, the proposed method still requires employing ICL to extract the task activations, and it feels too strong to call that a separate method of its own. SITE could be useful for adapting models to known, repeated tasks, but due to how it requires a per-task optimization step (to extract embeddings and learn soft head parameters), it is not useful in practice to actually replace how ICL is currently used, for rapid adaptation to new tasks.
In addition, both the method and tasks used are very similar to the Todd et al. FV paper (cited in the submission). The paper mostly introduces a second step to learn soft-head selection parameters, which is also used in https://arxiv.org/abs/2505.05145 -- I can give this submission credit for this idea as well as the other work seems to be concurrent to the submission, however, I believe that with just this, the paper lacks novelty and practical contributions to justify acceptance.

**Questions:**

Could you also discuss how your work situates to other efficient parameter-light adaptation methods, like LoRA?

---

### Note · Authors · 2025-11-17

I have read and agree with the venue's withdrawal policy on behalf of myself and my co-authors.